# BIOLOGICALLY PLAUSIBLE LEARNING VIA BIDIRECTIONAL SPIKE-BASED DISTILLATION

**Yifei Wang**[1,2*]   **Yanxun Zhang**[1,2*]   **Changze Lv**[1,2*]   **Yiyang Lu**[1,2]   **Jingwen Xu**[1,2]
**Xiaohua Wang**[1,2]   **Di Yu**[3]   **Xin Du**[3]   **Xuanjing Huang**[1,2]   **Xiaoqing Zheng**[1,2]
[1] College of Computer Science and Artificial Intelligence, Fudan University
[2] Shanghai Key Laboratory of Intelligent Information Processing
[3] School of Software Technology, Zhejiang University
{wangyif22,yanxunzhang22,czlv24}@m.fudan.edu.cn
{zhengxq,xjhuang}@fudan.edu.cn

## ABSTRACT

Developing biologically plausible learning algorithms that can achieve performance comparable to error backpropagation remains a longstanding challenge. Existing approaches often compromise biological plausibility by entirely avoiding the use of spikes for error propagation or relying on both positive and negative learning signals, while the question of how spikes can represent negative values remains unresolved. To address these limitations, we introduce Bidirectional Spike-based Distillation (BSD), a novel learning algorithm that jointly trains a feedforward and a backward spiking network. We formulate learning as a transformation between two spiking representations (i.e., stimulus encoding and concept encoding) so that the feedforward network implements perception and decision-making by mapping stimuli to actions, while the backward network supports memory recall by reconstructing stimuli from concept representations. Extensive experiments on diverse benchmarks, including image recognition, image generation, and sequential regression, show that BSD achieves performance comparable to networks trained with classical error backpropagation. These findings represent a significant step toward biologically grounded, spike-driven learning in neural networks. Our code is available at `https://github.com/alden199/Bidirectional-Spike-Based-Distillation`.

## 1 INTRODUCTION

Human learning and cognition cannot be reduced to a simple unidirectional "perception-to-decision" pipeline. Rather, they emerge from bidirectional processes that integrate bottom-up sensory perception with top-down memory recall (Caucheteux et al., 2023; Bonetti et al., 2024). During visual perception, retinal signals are transformed through hierarchical neural pathways into high-level conceptual representations stored in memory. Conversely, during recall, the brain can reconstruct partial sensory features from these stored representations.

Kosslyn et al. (1993) showed with positron emission tomography that the same early visual cortical areas (i.e., V1 and V2) activated during perception are also engaged when subjects visualize objects with their eyes closed. Later studies demonstrated that stimulus identity can be decoded from early visual cortex activity during both working memory and mental imagery, with patterns resembling those elicited by actual stimulation (Albers et al., 2013). For instance, consider a learner distinguishing between different bird species. At the outset, early visual areas may encode only basic features such as edges or color patches, making two similar species appear nearly indistinguishable from each other. As categorical knowledge of the species is acquired, higher-level conceptual representations emerge and feed back to early visual regions. This feedback sharpens perceptual sensitivity to subtle diagnostic features, such as beak curvature or wing pattern, thereby refining low-level visual representations in accordance with learned category distinctions.

---

*Equal Contribution. Correspondence to Xiaoqing Zheng (zhengxq@fudan.edu.cn).

Inspired by the brain's bidirectional architecture of perception and recall, we introduce bidirectional spike-based distillation (BSD), a novel learning algorithm that frames learning as a transformation between two spiking representations: stimulus encoding and concept encoding. The feedforward pathway performs perception and decision-making by mapping sensory stimuli to conceptual representations, analogous to the brain's analytical mode, while the feedback pathway facilitates memory recall by reconstructing stimuli from semantic concept encodings, analogous to the brain's imaginative mode. These feedforward and feedback networks can be trained jointly by distilling feature representations from one another. By integrating perception and recall within a unified framework, BSD provides a biologically grounded alternative to conventional unidirectional learning paradigms.

We also show that the proposed bidirectional distillation (implemented via spike trains) yields a more biologically plausible learning algorithm. While backpropagation has achieved remarkable success in deep learning (LeCun et al., 2015; Rumelhart et al., 1986), its underlying mechanisms conflict with established neurobiological principles (Crick, 1989; Lillicrap et al., 2020). Key inconsistencies include the requirement for symmetric feedforward and feedback weights, reliance on global error signals instead of local synaptic plasticity, a two-stage learning process that clearly separates forward and backward passes, and the use of continuous activations rather than discrete spike-based communication. To address these limitations, and building on the three criteria proposed by Lv et al. (2025), we introduce two additional requirements: neurons should communicate using discrete binary spikes for both learning and inference, and learning should rely on unsigned spiking signals only (Hayden et al., 2011). The learning algorithm we present demonstrates that all five criteria can be satisfied, whereas existing approaches typically fall short on one or more of these criteria.

Through extensive experiments across a range of tasks, including image classification, text character prediction, time-series forecasting, and image generation, and using diverse network architectures such as multi-layer perceptrons, convolutional neural networks, recurrent neural networks, and autoencoders, we demonstrate that BSD achieves performance comparable to backpropagation while satisfying all five criteria for biological plausibility. Our results indicate that more adherence to biological fidelity does not necessarily compromise computational effectiveness.

The contributions of this study can be summarized as follows:

- Inspired by the brain's bidirectional architecture of perception and recall, we propose a novel learning framework in which the feedforward network (stimuli-to-decision) and the backward network (concept-to-stimuli) are jointly trained by mutually distilling spiking feature representations.
- We introduce two additional biological plausibility criteria to complement the three proposed by Lv et al. (2025), resulting in five key principles: asymmetric forward and backward weights, local error representation, non-two-stage learning, spiking neuron models, and unsigned error signals. The BSD algorithm satisfies all five criteria and demonstrates enhanced biological plausibility.
- We perform extensive experiments using the BSD algorithm across diverse network architectures and various tasks. The experimental results show that BSD achieves performance comparable to error backpropagation while maintaining greater adherence to biological fidelity.

## 2 RELATED WORK

Backpropagation (BP) (Rumelhart et al., 1986) has long been criticized for its limited biological plausibility, as it depends on weight symmetry (Stork, 1989), global error signals (Crick, 1989), and a strictly sequential forward–backward computation process (Guerguiev et al., 2017; Hinton, 2022).

To address these limitations, numerous alternative approaches have been proposed to improve biological plausibility (Schmidgall et al., 2024; Jiao et al., 2022; Li et al., 2024). Recent advances in spiking neural networks have introduced attention mechanisms (Yao et al., 2023b;a) and residual learning (Hu et al., 2024) to enhance representational capacity while maintaining spike-based communication. To eliminate the reliance on global error signals, local loss methods (Mostafa et al., 2018) and their variants (Belilovsky et al., 2019; Nøkland & Eidnes, 2019; Kaiser et al., 2020) have been introduced, which typically employ fixed or trainable auxiliary heads to align hidden layers directly with the target. Feedback alignment (Lillicrap et al., 2016) addresses the weight transport problem by replacing the backward weights with fixed, randomly initialized matrices, thereby breaking the symmetry between forward and backward weights. Target propagation (TP) (Bengio, 2014) introduces approximate inverse models to generate layer-wise targets, with weight updates obtained by minimizing the mis-

match between outputs and targets. Although TP employs local losses and avoids weight symmetry, it requires each layer to transmit two distinct types of signals at different times. Moreover, TP often suffers from convergence instability. Extensions such as Difference Target Propagation (DTP) (Lee et al., 2015) and SDTP (Bartunov et al., 2018) improved stability and performance but offered little progress towards greater biological plausibility, and Biologically-plausible Reward Propagation (BRP) (Zhang et al., 2021), which replaces floating-point interlayer signals in TP with spike-based signals, has not demonstrated reliable generalization across tasks. Predictive coding (Rao & Ballard, 1999) offers another influential framework, postulating that the brain continually generates top-down predictions to minimize sensory prediction errors. Motivated by this framework, Error-driven Local Representation Alignment (LRA-E) (Ororbia & Mali, 2019) introduces a mechanism where error signals are projected backward to generate local target representations for hidden layers. The network then learns by minimizing the local discrepancy between actual neuronal activities and these generated targets, thereby enabling training without global error backpropagation. Other methods, such as Decoupled Neural Interfaces (DNI) (Jaderberg et al., 2017), train an auxiliary head at each hidden layer to approximate layer-wise gradients yet fail to break weight symmetry. Alternatively, Dendritic Localized Learning (DLL) (Lv et al., 2025) employs local error signals for weight updates, yet still relies on transmitting signed floating-point values and generally performs poorly across benchmarks. Counter-Current Learning (CCL) (Kao & Hariharan, 2024) faces the same limitation of floating-point communication and further lacks demonstrated effectiveness in sequential regression tasks.

Other biologically inspired mechanisms, like Hebbian learning (Donald, 1949) and spike-timing-dependent plasticity (STDP) (Song et al., 2000), adjust synaptic strength according to correlations in neuronal activity, while burst-dependent synaptic plasticity(Payeur et al., 2021) regulates synaptic plasticity by high-frequency bursts of spikes. Although fully consistent with biological observations, these rules struggle to integrate supervised learning signals, and STDP additionally requires precise temporal resolution. Energy-based learning approaches (LeCun et al., 2006), e.g., Boltzmann machines (Ackley et al., 1985), Hopfield networks (Hopfield, 1984), and contrastive learning frameworks (Hinton, 2002), instead optimize an energy function. However, minimizing energy does not always correspond to reducing task-specific loss, limiting their utility for general supervised learning. E-prop (Bellec et al., 2020) approximates BPTT for SRNNs using eligibility traces; while supporting local, online updates, it remains limited by the use of signed error signals.

In contrast, our approach, which is inspired by the brain's bidirectional interplay between perception and recall, exhibits improved biological plausibility while achieving stable convergence and superior performance on various benchmarks and tasks.

## 3    PRELIMINARY

### 3.1    SPIKING NEURONS

We adopt the leaky integrate-and-fire (LIF) neuron model (Maass, 1997) in our spiking neural networks (SNNs). The dynamics of the LIF neuron are formulated in discrete time as follows:

$$U[t] = H[t](1 - S[t]) + U_{\text{reset}}S[t], \tag{1}$$

$$H[t] = U[t-1] + \frac{1}{\tau}\big(I[t] - (U[t-1] - U_{\text{reset}})\big), \tag{2}$$

$$S[t] = \Theta(H[t] - U_{\text{thr}}), \tag{3}$$

where $I[t]$ represents the input current at time step $t$. Here, $H[t]$ and $U[t]$ denote the membrane potential before and after the trigger of a spike $S[t]$, respectively. The parameter $\tau$ is the membrane time constant, while $U_{\text{thr}}$ and $U_{\text{reset}}$ specify the firing threshold and reset potential. A spike is generated when the pre-spike potential $H[t]$ exceeds the threshold $U_{\text{thr}}$, in which case the neuron fires ($S[t] = 1$) and the membrane potential is reset to $U_{\text{reset}}$. Additional preliminaries are given in the appendix B.

### 3.2    BIOLOGICAL PLAUSIBILITY CRITERIA

In this work, we mainly build upon three biological plausibility criteria established by Lv et al. (2025): **C1.** Asymmetric synaptic weights for feedforward and feedback pathways; **C2.** Local synaptic plasticity based solely on locally available information without global error signals; **C3.** Non-dual-

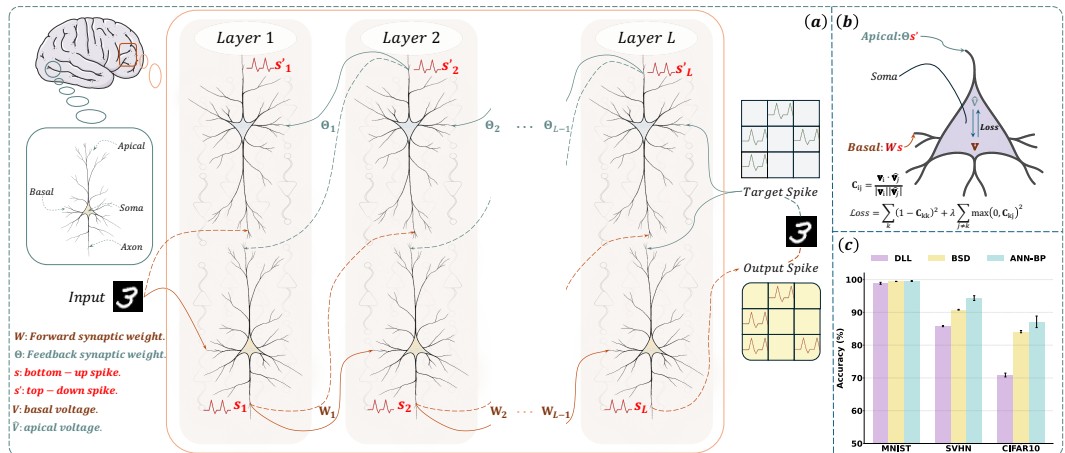

Figure 1: (a) Overview of the bidirectional spike-based distillation framework. (b) Illustration of feature alignment using the properties of pyramidal neurons, which receive feedforward and feedback signals through their basal and apical dendrites, respectively. (c) Performance comparison of neural networks trained with BSD, dendritic localized learning (DLL), a recently proposed biologically plausible learning algorithm, and backpropagation. The experimental results show that BSD achieves performance comparable to backpropagation.

phase training that eliminates sequential forward–backward dependencies. We will also introduce two additional criteria in Section 4, further extending the framework of biologically plausible learning.

# 4 METHOD

## 4.1 DESIGN PRINCIPLES

Inspired by the brain's bidirectional processes of perception and recall, we propose Bidirectional Spike-Based Distillation (BSD), which frames learning as a transformation between stimulus encoding and concept encoding through spiking representations. The feedforward pathway maps sensory inputs to concepts for perception and decision-making, while the backward pathway reconstructs inputs from concepts, thereby supporting feedforward learning during training.

Before introducing BSD in detail, we extend the three biological plausibility criteria of Lv et al. (2025) with two additional principles. Designed to satisfy all five criteria, BSD ensures a biologically grounded foundation for learning. The two additional criteria we propose are:

**C4. Model of Neurons.** Neurons in conventional artificial neural networks produce continuous activations, which are typically interpreted as approximations of average firing rates (Maass, 1997). This stands in contrast to the brain's actual processing, where biological neurons communicate using discrete action potentials (0-1 spikes) (Gerstner et al., 2014).

**C5. Unsigned error signaling.** The learning mechanism in most artificial neural networks relies on signed error signals to propagate directional gradient information for weight adjustments (Rumelhart et al., 1986). This stands in contrast with neurophysiological findings, which indicate that neurons encode unsigned reward prediction errors, firing in response to surprising outcomes irrespective of their positive or negative valence (Hayden et al., 2011).

To facilitate the understanding of the BSD algorithm, we illustrate both the network architecture and the neuronal learning mechanism in Figure 1, using an $L$-layer MLP configuration. The complete form of the global algorithm is detailed in Algorithm 1.

## 4.2 MODEL ARCHITECTURE

**Neuronal Dynamics.** To highlight the biological foundations of BSD, we first describe the neuron model employed in BSD. We follow Sacramento et al. (2018) and utilize pyramidal neurons with

a three-compartment structure: soma, apical dendrites (carrying backward learning signals), and basal dendrites (receiving feedforward inputs). To satisfy criterion **C4** (model of neurons), we use spiking neurons that emit discrete pulses instead of continuous activation values. For each neuron, the membrane potentials arriving at the soma from the basal and apical compartments are denoted $v$ and $\hat{v}$, respectively. The apical potential $\hat{v}$ acts as a supervisory signal guiding synaptic plasticity on the basal dendrites, while the basal potential $v$ drives spike generation through $s = \mathcal{SN}(v)$, where $\mathcal{SN}(\cdot)$ denotes the spiking operation that integrates input voltage and produces spike outputs. In the following sections, we use bold lowercase letters (e.g., $\mathbf{v}_i$, $\hat{\mathbf{v}}_i$) to denote layer-wise vectors, which are formed by aggregating the single-neuron scalar potentials ($v$ and $\hat{v}$, respectively) from all neurons in layer $i$.

**Network Architecture.**    The network comprises $L$ layers of pyramidal neurons, where each layer contains an equal number of two distinct types of neurons. Type 1 neurons receive synaptic inputs from lower-layer neurons and constitute the feedforward pathway, implementing a stimuli-to-decision process that transforms sensory inputs into conceptual representations. Conversely, Type 2 neurons receive inputs from higher-layer neurons and form the backward pathway, which attempts to reconstruct sensory features from conceptual encodings to assist feedforward learning. During training, the original input $\mathbf{x}$ is delivered to bottom-layer Type 1 neurons, while the learning target is first encoded into a spike train $\hat{\mathbf{s}}$ and provided to top-layer Type 2 neurons. The two pathways are jointly optimized through mutual distillation of their spiking feature representations, enabling bidirectional information flow that mirrors the brain's perception-recall architecture. To satisfy criterion **C1** (asymmetric synaptic weights), we employ independent synaptic weight matrices $\mathbf{W}$ and $\boldsymbol{\Theta}$ for the feedforward and backward pathways, respectively. The feedforward path is described by:

$$\mathbf{v}_1 = \mathbf{x}; \quad \mathbf{v}_i = \hat{\mathbf{v}}'_i = \mathbf{W}_{i-1}\mathbf{s}_{i-1}, \quad \mathbf{s}_i = \mathcal{SN}(\mathbf{v}_i), \quad i = 2, 3, \dots, L, \quad (4)$$

where $\mathbf{v}_i$ denotes the somatic membrane potential of Type 1 neurons in layer $i$ from basal dendritic integration, and $\hat{\mathbf{v}}'_i$ denotes the somatic membrane potential of Type 2 neurons in layer $i$ arising from apical dendritic integration. $\mathbf{W}_i$ is the synaptic weight for Type 1 neurons in layer $i$, and $\mathbf{s}_i$ denotes the spike train that Type 1 neurons in layer $i$ output.

The backward pathway is described by:

$$\mathbf{v}'_L = \hat{\mathbf{s}}; \quad \mathbf{v}'_i = \hat{\mathbf{v}}_i = \boldsymbol{\Theta}_i\mathbf{s}'_{i+1}, \quad \mathbf{s}'_i = \mathcal{SN}(\mathbf{v}'_i), \quad i = 1, 2, \dots, L-1, \quad (5)$$

where $\mathbf{v}'_i$ denotes the somatic membrane potential of Type 2 neurons in layer $i$ arising from basal dendritic integration, $\hat{\mathbf{v}}_i$ denotes the somatic membrane potential of Type 1 neurons in layer $i$ arising from apical dendritic integration. $\boldsymbol{\Theta}_i$ is the synaptic weight for Type 2 neurons in layer $i$, and $\mathbf{s}'_i$ denotes the spike train that Type 2 neurons in layer $i$ output.

## 4.3 Training Procedure

Motivated by the neuronal least-action principle (Senn et al., 2024), which postulates that pyramidal neurons minimize somato-dendritic mismatch errors through voltage dynamics, and to satisfy criterion **C2** (local error computation), our BSD algorithm introduces local loss functions for individual neurons to align basal-received voltage $v$ with apical-received voltage $\hat{v}$. Type 1 and Type 2 neurons respectively receive bottom-up sensory inputs $\mathbf{x}$ and top-down target signals $\hat{\mathbf{s}}$, which in classification tasks correspond to distinct modalities. Thus, the alignment between $v$ and $\hat{v}$ can be viewed as aligning cross-modal embeddings. Inspired by contrastive learning in multimodal representation learning and to satisfy **C5** (Unsigned Error Signal), we employ the Relaxed Contrastive (ReCo) loss (Lin et al., 2023), which is an unsigned loss function. In our design, error computation is localized within each neuron, and no error signals are propagated across the network. Specifically, for Type 1 neurons in a given layer $i$ (for $i = 2, \dots, L-1$), let $\mathbf{v}_{i,k}$ and $\hat{\mathbf{v}}_{i,k}$ denote the basal and apical membrane voltage for the $k$-th sample in a batch, respectively. By stacking these vectors across the batch dimension $B$, we construct matrices $\mathbf{V}_i \in \mathbb{R}^{B \times D_i}$ and $\hat{\mathbf{V}}_i \in \mathbb{R}^{B \times D_i}$, where $D_i$ is the number of neurons in layer $i$. The layer-specific affinity matrix $\mathbf{C}_i \in \mathbb{R}^{B \times B}$ is then defined by its elements:

$$[\mathbf{C}_i]_{kj} = \frac{\mathbf{v}_{i,k} \cdot \hat{\mathbf{v}}_{i,j}}{\|\mathbf{v}_{i,k}\|\|\hat{\mathbf{v}}_{i,j}\|}, \quad (6)$$

The local loss for Type 1 neurons in layer $i$ (for $i = 1, \dots, L-1$), denoted $\mathcal{L}_i$, is defined as:

$$\mathcal{L}_i = \sum_{k=1}^{B}(1 - [\mathbf{C}_i]_{kk})^2 + \lambda \sum_{k=1}^{B}\sum_{j \neq k}\left(\max(0, [\mathbf{C}_i]_{kj})\right)^2, \quad (7)$$

where $\lambda$ is a hyperparameter that controls the penalty strength for suppressing spurious correlations between non-corresponding voltage pairs. The local loss for Type 2 neurons, $\mathcal{L}_i'$, is defined analogously, with its formulation provided in Appendix D. Compared with InfoNCE loss commonly used in contrastive learning scenarios, ReCo loss avoids penalizing negatively correlated voltage pairs between $\mathbf{V}_i$ and $\hat{\mathbf{V}}_i$, thereby introducing enhanced flexibility and representational richness to learned embeddings while preserving alignment objectives. We provide a more detailed justification for adopting the ReCo loss in Appendix Q. Considering criterion **C2** (local synaptic plasticity without global error signals), we employ `detach()` operations to truncate the computational graph between layers, ensuring that each neuron's loss only propagates learning signals to synaptic weights connected to its dendrites. As a result, the gradient of the local loss $\mathcal{L}_i$ with respect to the feedforward weights $\mathbf{W}_{i-1}$ (for $i = 2, \ldots, L-1$) is given by:

$$
\frac{\partial \mathcal{L}_i}{\partial \mathbf{W}_{i-1}} = \sum_{k=1}^{B} \left[ -2(1 - [\mathbf{C}_i]_{kk}) \frac{1}{\|\mathbf{v}_{i,k}\|} \left( \frac{\hat{\mathbf{v}}_{i,k}}{\|\hat{\mathbf{v}}_{i,k}\|} - [\mathbf{C}_i]_{kk} \frac{\mathbf{v}_{i,k}}{\|\mathbf{v}_{i,k}\|} \right) \right.
$$
$$
\left. + \sum_{j \neq k} 2\lambda \max(0, [\mathbf{C}_i]_{kj}) \frac{1}{\|\mathbf{v}_{i,k}\|} \left( \frac{\hat{\mathbf{v}}_{i,j}}{\|\hat{\mathbf{v}}_{i,j}\|} - [\mathbf{C}_i]_{kj} \frac{\mathbf{v}_{i,k}}{\|\mathbf{v}_{i,k}\|} \right) \right] (\mathbf{s}_{i-1,k})^T,
$$
(8)

where $B$ denotes the batch size, $\mathbf{s}_{i-1,k}$ represents the spike output vector of Type 1 neurons from layer $i - 1$ for the $k$-th sample, and $\mathcal{L}_i$ denotes the local loss for neurons in layer $i$. This locally computed gradient is then used to update the feedforward synaptic weights via a standard gradient descent step for $i = 2, \ldots, L-1$:

$$
\mathbf{W}_{i-1}^{\text{new}} = \mathbf{W}_{i-1}^{\text{old}} + \eta_{\mathbf{W}} \frac{\partial \mathcal{L}_i}{\partial \mathbf{W}_{i-1}},
$$
(9)

where the superscripts 'new' and 'old' denote the weights after and before the update, and $\eta_{\mathbf{W}}$ is the learning rate for the feedforward weights. A symmetric update rule, also based on its corresponding local gradient, is applied to the backward weights $\mathbf{\Theta}$. The detailed gradient derivations for both $\mathbf{W}$ and $\mathbf{\Theta}$ are presented in Appendix D. The total loss for Type 1 neurons is defined as:

$$
\mathcal{L}_{\text{total}} = \sum_{i=1}^{L-1} \mathcal{L}_i + \mathcal{L}_{\text{top}},
$$
(10)

where the top-layer loss is defined as $\mathcal{L}_{\text{top}} = \sum_{k=1}^{B} \mathcal{L}_{\text{CE}}(\mathbf{v}_{L,k}, \hat{\mathbf{v}}_{L,k})$, where $\mathcal{L}_{\text{CE}}$ represents the cross-entropy loss. The total loss for Type 2 neurons, $\mathcal{L}'_{\text{total}}$, is defined symmetrically. As all information processing can be separated within a single neuronal compartment, the feedforward and backward distillation processes do not require strict temporal separation and can learn simultaneously, thereby satisfying **C3** (non-two-stage training). For classification tasks, during inference, the predicted label for input $\mathbf{x}$ is determined by computing cosine similarity between the ensemble of output spikes $\mathbf{s}_L$ and the spike trains corresponding to all candidate labels. The detailed training and inference procedures for RNN architectures are presented in Appendix C.

## 4.4 Learning for generation tasks

We employ an autoencoder architecture for generation tasks, where both bottom-up and top-down inputs are images $\mathbf{x}$. To capture fine-grained edge details while reducing noise artifacts, we apply Fast Fourier Transform (FFT) decomposition to the input images as well as to all basal and apical membrane voltages. This frequency-domain representation separates low- and high-frequency components, enabling adaptive loss computation. Within each layer, the regularization parameter $\lambda$ is adjusted according to frequency content: larger values are assigned to high-frequency components to suppress spurious correlations and preserve edge fidelity, whereas smaller values are used for low-frequency components to avoid noise amplification and maintain structural coherence. At the top layer, mean squared error is employed to compute the loss between basal and apical voltages. The detailed mechanism of the FFT decomposition is explained in Appendix M.

## 5 Experiments

In this section, we first present the experimental settings and implementation details. We then evaluate our proposed BSD algorithm on image classification tasks, comparing it against other biologically

Table 1: Comparison of various learning algorithms in terms of biological plausibility criteria and image classification performance. Our proposed BSD algorithm satisfies all five criteria of biological plausibility (**C1**–**C5**) while achieving performance comparable to backpropagation. "C1, C2, C3, C4, C5" refer to the criteria defined in Section 4.1. Results are averaged over four random seeds.

| Method | C1 | C2 | C3 | C4 | C5 | Model | MNIST | FashionMNIST | SVHN | CIFAR-10 | CIFAR100 | Avg. |
|---|---|---|---|---|---|---|---|---|---|---|---|---|
| Backpropagation on ANNs | ✗ | ✗ | ✗ | ✗ | ✗ | MLPs | **98.77**%$_{\pm0.33\%}$ | **89.59**%$_{\pm0.14\%}$ | **61.65**%$_{\pm0.42\%}$ | **57.65**%$_{\pm0.08\%}$ | **27.92**%$_{\pm0.17\%}$ | **67.12**% |
| | | | | | | CNNs | **99.56**%$_{\pm0.14\%}$ | **92.68**%$_{\pm0.42\%}$ | **94.36**%$_{\pm0.73\%}$ | **87.12**%$_{\pm1.76\%}$ | **57.75**%$_{\pm0.35\%}$ | **86.29**% |
| Backpropagation on SNNs | ✗ | ✗ | ✗ | ✓ | ✗ | MLPs | 98.57%$_{\pm0.24\%}$ | 88.87%$_{\pm0.65\%}$ | 61.28%$_{\pm0.52\%}$ | 49.95%$_{\pm0.53\%}$ | 23.32%$_{\pm0.87\%}$ | 64.40% |
| | | | | | | CNNs | 99.25%$_{\pm0.02\%}$ | 92.48%$_{\pm0.48\%}$ | 94.13%$_{\pm0.02\%}$ | 87.02%$_{\pm0.09\%}$ | 57.21%$_{\pm0.34\%}$ | 86.02% |
| Predictive Coding | ✗ | ✓ | ✗ | ✗ | ✗ | MLPs | 98.42%$_{\pm0.13\%}$ | 88.72%$_{\pm0.65\%}$ | 59.05%$_{\pm0.45\%}$ | 47.34%$_{\pm0.24\%}$ | 19.72%$_{\pm0.32\%}$ | 62.65% |
| | | | | | | CNNs | 99.41%$_{\pm0.40\%}$ | 92.03%$_{\pm0.70\%}$ | 94.53%$_{\pm1.54\%}$ | 72.94%$_{\pm0.32\%}$ | 53.08%$_{\pm0.43\%}$ | 82.40% |
| CCL | ✓ | ✓ | ✓ | ✗ | ✗ | MLPs | 98.13%$_{\pm0.10\%}$ | 88.58%$_{\pm0.29\%}$ | 60.98%$_{\pm0.23\%}$ | 52.73%$_{\pm0.59\%}$ | 21.76%$_{\pm0.22\%}$ | 64.44% |
| | | | | | | CNNs | 96.30%$_{\pm0.05\%}$ | 83.70%$_{\pm0.57\%}$ | 88.78%$_{\pm0.37\%}$ | 82.94%$_{\pm0.53\%}$ | 56.29%$_{\pm0.25\%}$ | 81.60% |
| DLL | ✓ | ✓ | ✓ | ✗ | ✗ | MLPs | 97.57%$_{\pm0.40\%}$ | 87.50%$_{\pm0.43\%}$ | 56.60%$_{\pm0.12\%}$ | 45.87%$_{\pm0.10\%}$ | 18.24%$_{\pm0.18\%}$ | 61.16% |
| | | | | | | CNNs | 98.87%$_{\pm0.30\%}$ | 90.88%$_{\pm0.40\%}$ | 85.81%$_{\pm0.17\%}$ | 70.89%$_{\pm0.58\%}$ | 38.60%$_{\pm0.21\%}$ | 77.01% |
| R-STDP | ✓ | ✓ | ✓ | ✓ | ✓ | MLPs | 77.18%$_{\pm0.17\%}$ | 70.03%$_{\pm0.28\%}$ | 41.76%$_{\pm0.46\%}$ | 22.68%$_{\pm0.30\%}$ | 1.33%$_{\pm0.15\%}$ | 42.58% |
| | | | | | | CNNs | 91.67%$_{\pm0.04\%}$ | 74.29%$_{\pm0.30\%}$ | 50.02%$_{\pm0.32\%}$ | 33.19%$_{\pm0.38\%}$ | 1.49%$_{\pm0.22\%}$ | 50.10% |
| **BSD (Ours)** | ✓ | ✓ | ✓ | ✓ | ✓ | MLPs | 95.62%$_{\pm0.09\%}$ | **86.39**%$_{\pm0.13\%}$ | 60.40%$_{\pm0.18\%}$ | 48.90%$_{\pm0.54\%}$ | **22.10**%$_{\pm0.25\%}$ | 62.68% |
| | | | | | | CNNs | **99.44**%$_{\pm0.03\%}$ | **91.05**%$_{\pm0.20\%}$ | **90.81**%$_{\pm0.11\%}$ | **84.13**%$_{\pm0.34\%}$ | **53.48**%$_{\pm0.22\%}$ | **83.78**% |

plausible learning algorithms. We assess BSD-trained RNNs on sequential regression tasks and evaluate BSD-trained autoencoders on image generation tasks. Finally, we conduct ablation studies and analyze the network's convergence properties and performance.

## 5.1 EXPERIMENTAL SETTINGS

**Image Classification.** We evaluate our BSD algorithm on widely used benchmarks including MNIST, FashionMNIST, SVHN, CIFAR-10, and CIFAR-100, using classification accuracy as the metric.

**Text Character Prediction.** We conduct next-character prediction experiments with BSD-trained RNNs on the Harry Potter text corpus (Plath et al., 2019), reporting prediction accuracy.

**Time-Series Forecasting.** BSD-trained RNNs are evaluated on three widely used real-world multivariate time-series forecasting datasets: Electricity (Lai et al., 2018), Metr-la (Li et al., 2018), and Pems-bay (Li et al., 2018), with performance primarily measured by mean squared error (MSE).

**Image Generation.** BSD-trained autoencoders are applied to generation tasks on MNIST, FashionMNIST, and CIFAR-10, with generation quality assessed using Fréchet Inception Distance (FID) (Heusel et al., 2017). Additional results and visualizations are provided in Appendix F.

## 5.2 IMPLEMENTATION DETAILS

To ensure fair comparison, identical network architectures are used across all learning algorithms for each task. For MLPs, CNNs, RNNs, and autoencoders, the same configurations are applied when comparing methods on a given dataset. Detailed dataset descriptions, architectural specifications, hyperparameter settings, and evaluation metrics are provided in Appendix E. A comprehensive analysis of the computational costs, including training and inference memory consumption, is provided in Appendix N.

## 5.3 IMAGE CLASSIFICATION

We benchmark the performance of backpropagation on Artificial Neural Networks (ANNs) and Spiking Neural Networks (SNNs), Predictive Coding, Reward-modulated Spike-Timing-Dependent Plasticity (R-STDP) (Izhikevich, 2007), Counter-Current Learning (CCL), Dendritic Localized Learning (DLL), and our BSD algorithm on image classification tasks. Comprehensive results are presented in Table 1, with additional implementation details provided in Appendix E.2.

**Spiking neuron outputs degrade performance compared to continuous activations.** When trained with backpropagation, Spiking Neural Networks (SNNs) consistently exhibit inferior performance compared to Artificial Neural Networks (ANNs) across all evaluated datasets and network architectures. This performance degradation highlights the disadvantage of using spiking neurons, which emit binary 0–1 spikes, as opposed to neurons that produce continuous-valued activations.

Table 2: Comparison of different learning algorithms for RNN training on text character prediction and time-series forecasting tasks. Our proposed BSD algorithm achieves performance comparable to backpropagation. ↑ (↓) denotes higher (lower) values indicate better performance. All results are averaged across 4 random seeds. The best results and the results of BSD are presented in **bold**.

| Method | Harry Potter | Electricity | | Metr-la | | Pems-bay | |
|---|---|---|---|---|---|---|---|
| | Pred. Acc. ↑ | MSE ↓ | MAE ↓ | MSE ↓ | MAE ↓ | MSE ↓ | MAE ↓ |
| Backpropagation on ANNs | $51.9\%_{\pm 1.0\%}$ | $0.175_{\pm 0.007}$ | $0.324_{\pm 0.007}$ | $0.131_{\pm 0.004}$ | $0.214_{\pm 0.005}$ | $\mathbf{0.164}_{\pm 0.001}$ | $\mathbf{0.190}_{\pm 0.002}$ |
| Backpropagation on SNNs | $27.8\%_{\pm 0.9\%}$ | $0.169_{\pm 0.018}$ | $0.316_{\pm 0.021}$ | $0.154_{\pm 0.014}$ | $0.243_{\pm 0.022}$ | $0.166_{\pm 0.005}$ | $0.201_{\pm 0.002}$ |
| Predictive Coding | $38.8\%_{\pm 1.8\%}$ | $\mathbf{0.162}_{\pm 0.019}$ | $\mathbf{0.312}_{\pm 0.018}$ | $0.141_{\pm 0.001}$ | $0.228_{\pm 0.005}$ | $0.178_{\pm 0.004}$ | $0.202_{\pm 0.003}$ |
| DLL | $33.7\%_{\pm 0.6\%}$ | $0.172_{\pm 0.018}$ | $0.321_{\pm 0.013}$ | $0.155_{\pm 0.005}$ | $0.264_{\pm 0.001}$ | $0.178_{\pm 0.005}$ | $0.224_{\pm 0.004}$ |
| BSD (Ours) | $\mathbf{41.8}\%_{\pm 0.1\%}$ | $\mathbf{0.165}_{\pm 0.018}$ | $\mathbf{0.314}_{\pm 0.018}$ | $\mathbf{0.125}_{\pm 0.005}$ | $\mathbf{0.197}_{\pm 0.005}$ | $\mathbf{0.174}_{\pm 0.007}$ | $\mathbf{0.206}_{\pm 0.013}$ |

**Our proposed BSD algorithm reconciles biological plausibility with competitive performance.** BSD satisfies all five criteria of biological plausibility (**C1–C5**) while achieving performance comparable to backpropagation, with stable convergence across diverse datasets. In particular, on challenging datasets such as SVHN, CIFAR-10, and CIFAR-100, and on more complex architectures including CNNs, BSD consistently delivers robust performance. Taken together, these findings underscore the feasibility of attaining biological plausibility without sacrificing task performance. To further assess the scalability of our approach, we conducted experiments on the more complex Tiny-ImageNet dataset, with the results presented in Appendix I. We also evaluate the robustness of our models against input noise in Appendix L.

### 5.4 SEQUENTIAL REGRESSION TASKS

We evaluate BSD on four representative sequential regression tasks, comparing its performance against backpropagation on ANNs and SNNs, Predictive Coding, and Dendritic Localized Learning (DLL). Comprehensive results are reported in Table 2.

For time-series forecasting, we utilize the Electricity, Metr-la, and Pems-bay datasets. Predictive Coding satisfies only the second biological plausibility criterion, whereas DLL meets the first three. In contrast, BSD satisfies all five criteria while achieving performance comparable to backpropagation, with RNNs trained under BSD converging reliably across all forecasting tasks.

For text character prediction, experiments are conducted on the Harry Potter corpus. As shown in Table 2, BSD again converges successfully, outperforming both Predictive Coding and DLL.

Together, these results demonstrate that BSD effectively captures temporal dependencies in sequential regression tasks while remaining consistent with biologically plausible learning principles.

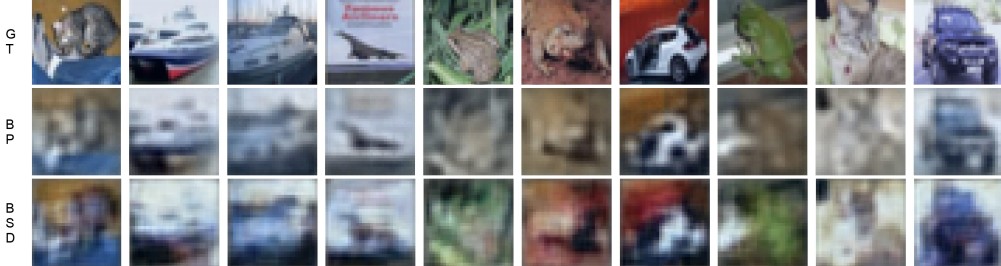

Figure 2: Results on image generation tasks using autoencoders trained with backpropagation and BSD. GT denotes the ground truth, BP indicates backpropagation, and BSD refers to our proposed method. The reconstruction quality demonstrates that BSD attains performance comparable to backpropagation on generation tasks, while preserving biological plausibility

### 5.5 IMAGE GENERATION

Figure 2 presents a comparison of image reconstruction on CIFAR-10 across three approaches: BP-trained autoencoders, the Fully Spiking Variational Autoencoder (FSVAE) (Kamata et al., 2022), and BSD-trained autoencoders. Generation quality is evaluated using the Fréchet Inception Distance (FID) (Heusel et al., 2017), a widely adopted metric for assessing the quality of generative models.

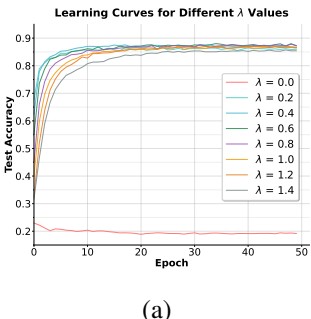 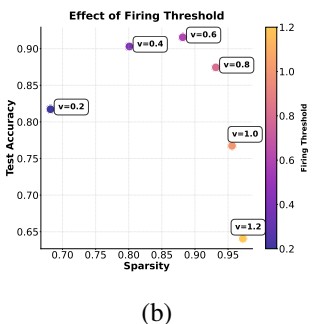 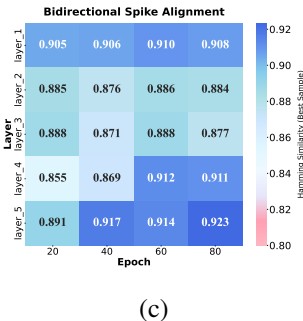

(a)          (b)          (c)

Figure 3: Analysis of BSD algorithm parameters and bidirectional spike dynamics. (a) Test accuracy curves of CNNs trained with BSD using different $\lambda$ values. (b) Impact of firing threshold on network performance. (c) Alignment of spikes emitted by Type 1 and Type 2 neurons within the same layer.

The visual results suggest that BSD adapts robustly to generative tasks and produces reconstruction quality closely comparable to that achieved with backpropagation. Table 3 reports FID scores on CIFAR-10 and MNIST, indicating that BSD attains competitive performance across both datasets and thereby further demonstrating its suitability for generative modeling while maintaining biological plausibility. Additional visualizations and results, including experiments on FashionMNIST, are presented in Appendix F.1.

Table 3: Performance comparison of different algorithms on image generation tasks. ↓ denotes lower values indicate better performance.

| Dataset | Model | FID ↓ |
|---------|-------|-------|
| CIFAR-10 | ANN-BP | **127.34** |
| | FSVAE | 175.5 |
| | **BSD (Ours)** | **168.12** |
| MNIST | ANN-BP | **49.56** |
| | FSVAE | 97.06 |
| | **BSD (Ours)** | **72.39** |

## 5.6 ABLATION STUDY

As mentioned in Section 4, inspired by contrastive learning, we adopt ReCo loss as our layer-wise loss function. Here, we delve deeper into the impact of layer-wise loss function selection, analyzing how BSD performs when using MSE or InfoNCE (van den Oord et al., 2018), a loss commonly used in contrastive learning scenarios, for aligning intraneuronal voltages. We term the method using MSE as layer-wise loss "BSD-MSE" and the method using InfoNCE as "BSD-InfoNCE," and conduct ablation experiments on both image classification and sequential regression tasks.

Table 4 compares the performance of BSD, BSD-MSE, and BSD-InfoNCE on image classification tasks. BSD-MSE yields markedly inferior results: MLPs attain only 19.11% accuracy on SVHN and fail to converge on other datasets, while CNNs

Table 4: Ablation study of loss functions on image classification tasks.

| Method | Model | MNIST | FashionMNIST | SVHN | CIFAR-10 | CIFAR-100 |
|--------|-------|-------|--------------|------|----------|-----------|
| BSD-MSE | MLPs | 12.51% | 13.72% | 19.11% | 11.49% | 1.39% |
| | CNNs | 21.10% | 29.31% | 19.46% | 16.93% | 1.58% |
| BSD-InfoNCE | MLPs | 94.56% | 85.71% | 57.33% | 43.77% | 19.25% |
| | CNNs | 98.77% | 88.97% | 83.27% | 72.38% | 38.06% |
| **BSD** | MLPs | **95.62%** | **86.39%** | **60.40%** | **48.90%** | **22.10%** |
| | CNNs | **99.44%** | **91.05%** | **90.81%** | **84.13%** | **53.48%** |

exhibit substantial performance gaps across all tasks and do not converge on CIFAR-100. BSD-InfoNCE converges reliably on all datasets, but BSD with ReCo loss consistently achieves higher accuracy. On complex benchmarks such as CIFAR-100, BSD surpasses BSD-InfoNCE by more than 15 accuracy points. Additional ablation experiments for text character prediction, time-series forecasting, and image generation tasks are provided in Appendix F.

To conclude, employing MSE loss for intraneuronal voltage alignment leads to convergence difficulties, indicating that it is ill-suited for BSD training. Moreover, compared to InfoNCE, ReCo loss offers a distinct advantage by not penalizing unpaired samples that are already orthogonal or negatively correlated, which allows for more flexible alignment and richer feature representations. We also investigate the impact of the number of timesteps on model performance in Appendix O. An ablation study demonstrating the importance of Batch Normalization is presented in Appendix P.

## 5.7 TRAINING ANALYSIS

In this section, we analyze the convergence behavior of BSD-trained models and the properties of Type 1 and Type 2 neurons during training. Since BSD applies ReCo loss to all layers except the top one and relies on spikes for inter-neuronal communication, both the penalty weight $\lambda$ in ReCo loss and the neuronal firing threshold are critical factors influencing its convergence. Additionally, we investigate the sensitivity of our framework to batch size in Appendix J, as contrastive learning methods often depend on sufficiently large batches. A detailed analysis of the energy efficiency of our BSD-trained models during inference is provided in Appendix K, highlighting the benefits of spike-based computation.

Figure 3(a) shows the effect of the penalty weight $\lambda$ on BSD-trained CNNs evaluated on the SVHN dataset. Performance drops markedly when $\lambda = 0$, underscoring the role of $\lambda$ in promoting separation among sample representations and thereby enlarging the representational space. We further observe that smaller values of $\lambda$ accelerate convergence, while the best final performance is achieved at $\lambda = 0.6$. Figure 3(b) illustrates the relationship among neuronal firing threshold, spike sparsity, and network performance on SVHN. The network attains its best performance at a firing threshold of 0.6, suggesting that thresholds that are either too low or too high impair effective learning. Figure 3(c) shows the Hamming similarity between the spike trains of Type 1 and Type 2 neurons within the same layer. The similarity rises quickly during training and exceeds 0.85 by epoch 20, demonstrating that the feedforward (stimuli-to-decision) and backward (concept-to-stimuli) pathways successfully achieve mutual alignment of their spiking feature representations through bidirectional distillation, thereby validating the effectiveness of our joint training framework.

For completeness, we provide $t$-SNE visualizations of representations on BSD-trained CNNs in Appendix H and examine the degree of alignment between weights $\mathbf{W}$ and $\mathbf{\Theta}$ in Appendix G.

## 6 CONCLUSION

Human learning and cognition emerge from bidirectional processes that integrate bottom-up sensory perception with top-down memory recall. In this framework, the feedforward network supports perception and decision-making by transforming sensory stimuli into conceptual representations, while the feedback network facilitates memory recall by reconstructing stimuli from semantic concepts. Inspired by this principle, we propose a novel bidirectional learning framework in which the feedforward and feedback networks are jointly trained by distilling their hidden feature representations from one another. This approach leverages the properties of pyramidal neurons, which receive feedforward (perception) and feedback (learning) signals through their basal and apical dendrites, respectively. Extensive experiments across diverse network architectures and tasks show that the resulting learning algorithm achieves performance comparable to error backpropagation while yielding stronger adherence to biological fidelity. Specifically, BSD achieves competitive performance across tasks: within 3% of backpropagation on MNIST (99.44% vs 99.56%) and CIFAR-10 (84.13% vs 87.12%), and superior MSE on time-series forecasting tasks like Electricity (0.165 vs 0.175), demonstrating the effectiveness of our approach. The limitations and future directions are discussed in Appendix R.

## ETHICS STATEMENT

This research presents a biologically plausible learning algorithm and does not involve human subjects or sensitive data. All experiments were conducted on publicly available datasets that are widely used in machine learning research. The proposed BSD algorithm is a general-purpose learning method without inherent bias, unethical practices, or discriminatory applications. The authors declare no conflicts of interest or competing financial interests related to this work. The research methodology adheres to well-established standard practices in machine learning and computational science, with no foreseeable harmful implications or misuse potential.

## REPRODUCIBILITY STATEMENT

The authors have made extensive efforts to ensure the reproducibility of the empirical results reported in this paper. First, detailed dataset characteristics are documented in Appendix E.1. Second,

comprehensive implementation details, covering network architectures, hyperparameter settings, training procedures, and evaluation metric explanations, are provided in Appendix E.2.

## ACKNOWLEDGMENTS

The authors would like to thank the anonymous reviewers for their valuable comments. This work was supported by the National Natural Science Foundation of China (No. 62076068).

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

## A  GLOBAL ALGORITHM OF BSD

---

**Algorithm 1** Spike Bidirectional Distillation Algorithm

---

1: **Input:** sequence of data $D$, layers $L$, learning rates $\eta_{\mathbf{W}}, \eta_{\mathbf{\Theta}}$
2: Initialize $\mathbf{W}_1, \ldots, \mathbf{W}_{L+1}$ and $\mathbf{\Theta}_1, \ldots, \mathbf{\Theta}_{L+1}$ randomly
3: **for** epoch $= 0, \ldots, $ max_epochs **do**
4:    **for** each batch $\mathbf{x}, \hat{\mathbf{y}} \in D$ **do**
5:       **Forward Path**(transforms the low-level signals into high-level representations):
6:       Assign the input values $\mathbf{x}$ to the voltage $\mathbf{v}_1$ of the input layer neurons.
7:       Inputs can be viewed as analog signals arising from low-level perceptual neurons.
8:       $\mathbf{v}_1 \leftarrow \mathbf{x}$ , $\mathbf{s}_1 \leftarrow \mathcal{SN}(\mathbf{v}_1)$,
9:       where $\mathrm{SN}(\cdot)$ denotes the leaky integrate-and-fire (LIF) spike generator, taking $\mathbf{v}_i$
          as input and yielding the spike train $\mathbf{s}_i$ for layer $i$.
10:      **for** $i = 2, \ldots, L$ **do**
11:         When spikes pass through the synaptic cleft, they are multiplied by $\mathbf{W}_i$, which is
            determined by the strength of the connection:
12:         $\hat{\mathbf{v}}_i' \leftarrow \mathbf{v}_i \leftarrow \mathbf{W}_{i-1}\mathbf{s}_{i-1}, \quad \mathbf{s}_i \leftarrow \mathcal{SN}(\mathbf{v}_i)$
13:      **end for**
14:
15:      **Backward Path**(transforms the high-level encoding to low-level stimuli):
16:      $\mathbf{v}_L' \leftarrow \hat{\mathbf{s}}$, where $\hat{\mathbf{s}}$ is the encoding of target $\hat{\mathbf{y}}$
17:      $\mathbf{s}_L' \leftarrow \mathcal{SN}(\mathbf{v}_L')$
18:      **for** $i = L - 1, \ldots, 1$ **do**
19:         $\hat{\mathbf{v}}_i \leftarrow \mathbf{v}_i' \leftarrow \mathbf{\Theta}_i \mathbf{s}_{i+1}', \quad \mathbf{s}_i' \leftarrow \mathcal{SN}(\mathbf{v}_i')$
20:      **end for**
21:
22:      **Loss Computation:**(layer-wise local loss)
23:      We present the loss defined for Type 1 neurons, with the loss for Type 2 neurons
         following symmetrically.
24:      **for** $i = 1, \ldots, L - 1$ **do**
25:         $\mathcal{L}_i \leftarrow \mathcal{L}_{\mathrm{ReCo}}(\mathbf{v}_i, \hat{\mathbf{v}}_i)$ ,
26:         where $\mathcal{L}_{\mathrm{ReCo}}$ denotes the Relaxed Contrastive (ReCo) loss described in Section 4.3.
27:      **end for**
28:      We calculate the loss of the last layer using cross-entropy: $\mathcal{L}_{\mathrm{top}} \leftarrow \mathcal{L}_{\mathrm{CE}}(\mathbf{v}_L, \hat{\mathbf{v}}_L)$
29:
30:      **Local Gradients:**
31:      **for** $i = 2, \ldots, L - 1$ **do**
32:         The calculation of the weight gradients only involves the local loss:
33:         $\nabla_{\mathbf{W}_{i-1}} \leftarrow \frac{\partial \mathcal{L}_i}{\partial \mathbf{W}_{i-1}}$ , $\nabla_{\mathbf{\Theta}_i} \leftarrow \frac{\partial \mathcal{L}_i}{\partial \mathbf{\Theta}_i}$
34:      **end for**
35:      $\nabla_{\mathbf{\Theta}_1} \leftarrow \frac{\partial \mathcal{L}_1}{\partial \mathbf{\Theta}_1}$
36:      $\nabla_{\mathbf{W}_{L-1}} \leftarrow \frac{\partial \mathcal{L}_{\mathrm{target}}}{\partial \mathbf{W}_{L-1}}$
37:
38:      **Parameter Updates:**
39:      **for** $i = 1, \ldots, L - 1$ **do**
40:         $\mathbf{W}_i \leftarrow \mathbf{W}_i - \eta_{\mathbf{W}} \cdot \nabla_{\mathbf{W}_i}$ , $\mathbf{\Theta}_i \leftarrow \mathbf{\Theta}_i - \eta_{\mathbf{\Theta}} \cdot \nabla_{\mathbf{\Theta}_i}$
41:      **end for**
42:   **end for**
43: **end for**

---

## B  ADDITIONAL PRELIMINARIES

Spiking neurons can be categorized into two types based on their signal propagation mechanisms. One category, referred to as spike-based neurons, conveys information through spike trains. Discrete Spiking Neural Networks (SNNs) integrate spike inputs at each time step and generate a binary spike output when the integrated value exceeds a threshold. Continuous analog input signals are typically

encoded (e.g., rate encoding, time encoding, or $\Delta$-encoding) into spike sequences. Representative models in this category include IF, PLIF(Fang et al., 2021), and LIF(Maass, 1997). Discrete SNNs are compatible with existing digital design processes, allowing the grouping and virtualization of large numbers of neurons on computational cores to achieve parallel architectures. These systems exhibit high scalability and efficient trainability, benefiting from sparse, event-driven computation with low power consumption during the inference phase. However, the discrete nature of spike signals limits their information capacity relative to traditional neuron models, which transmit continuous floating-point values directly(Pfeiffer & Pfeil, 2018).

The second type of spiking neuron architecture uses similar temporal equations as spiking neurons but propagates analog signals, referred to as analog-based neurons, such as LIAF(Wu et al., 2021). These signals can encode more information and exhibit lower response latency in certain tasks. However, hardware implementations of such neurons require analog circuits and because computations cannot be skipped during inference, the system exhibits higher power consumption (Hazan & Ezra Tsur, 2021).

In Section 3.1, we provide the time dynamics equation for LIF. The equation 1 and equation 3 for different types of spiking neurons are similar, while equation 2 exhibits multiple variations. For example, the equation for IF is defined as follows:

$$H[t] = U[t-1] + I[t] \tag{11}$$

The primary distinction between IF and LIF lies in the inclusion of a voltage leakage mechanism in LIF, which more closely resembles the behavior of biological neurons. Other variants, such as PLIF, consider the parameter $\tau$ in equation 2 to be learnable, while LIAF replaces $\Theta$ in equation 3 with ReLU, thereby enabling the transmission of continuous floating-point values.

## C  THE MODEL ARCHITECTURE OF RNNS

The network comprises one layer of auto-regressive pyramidal neurons, which contains a pair of two distinct types of neurons. Type 1 neurons receive synaptic inputs on their basal dendrites from axons of lower-layer neurons, while Type 2 neurons receive basal dendritic inputs from axons of higher-layer neurons. $\mathbf{x}_1, \mathbf{x}_2, \ldots, \mathbf{x}_N$ represent the input sequence where $N$ is the sequence length. The training objective is to align the output $\mathbf{o}_i$ of each timestep Type 1 neurons with $\hat{\mathbf{y}}_i$ at each timestep $i$. During learning, the original input $\mathbf{x}_i$ is delivered to the basal dendrites of Type 1 neurons at timestep $i$, while the target $\hat{\mathbf{y}}_i$ is provided to the basal dendrites of Type 2 neurons.

We employ synaptic weight matrices $\mathbf{W}_{ih}, \mathbf{W}_{hh}$ and $\mathbf{W}_{ho}$ in the feedforward pathway corresponding to the input to the hidden layer weight, the spike of the previous time to the hidden layer weight and the hidden layer to the output weight respectively. Similar to the feedforward pathway, $\mathbf{\Theta}_{oh}, \mathbf{\Theta}_{hh}$ and $\mathbf{\Theta}_{hi}$ in the backward pathways corresponding to the output target to the hidden layer weight, the spike of the previous time to the hidden layer weight and the hidden layer to the reconstructed input weight respectively.

The feedforward process is described by:

$$\mathbf{h}_i = \hat{\mathbf{h}}'_i = \mathbf{W}_{ih}\mathbf{x}_i + \mathbf{W}_{hh}\mathbf{s}_{i-1}, \quad \mathbf{o}_i = \mathbf{W}_{ho}\mathbf{h}_i, \quad \mathbf{s}_i = \mathcal{SN}(\mathbf{h}_i), \quad i = 2, 3, \ldots, N, \tag{12}$$

where $\mathbf{h}_i$ denotes the somatic membrane potential of Type 1 neurons at timestep $i$ arising from basal dendritic integration, $\hat{\mathbf{h}}'_i$ denotes the somatic membrane potential of Type 2 neurons at timestep $i$ arising from apical dendritic integration. $\mathbf{s}_i$ denotes the spike train that Type 1 neurons at timestep $i$ output and $\mathbf{s}_1 = 0$.

The backward process is governed by:

$$\mathbf{h}'_i = \hat{\mathbf{h}}_i = \mathbf{\Theta}_{oh}\hat{\mathbf{y}}_i + \mathbf{\Theta}_{hh}\mathbf{s}'_{i+1}, \quad \hat{\mathbf{x}}_i = \mathbf{\Theta}_{hi}\hat{\mathbf{h}}_i, \quad \mathbf{s}'_i = \mathcal{SN}(\hat{\mathbf{h}}_i), \quad i = 1, 2, \ldots, N-1, \tag{13}$$

where $\mathbf{h}'_i$ denotes the somatic membrane potential of Type 2 neurons at timestep $i$ arising from basal dendritic integration, $\hat{\mathbf{h}}_i$ denotes the somatic membrane potential of Type 1 neurons at timestep $i$ arising from apical dendritic integration. $\mathbf{\Theta}_i$ is the synaptic weight for Type 2 neurons in layer $i$, and $\mathbf{s}'_i$ denotes the spike train that Type 2 neurons at timestep $i$ output and $\mathbf{s}'_N = 0$.

## D  GRADIENTS DERIVATION FOR BSD

This section details the derivation of the gradients for the local loss functions with respect to the feedforward weights $\mathbf{W}$ and backward weights $\boldsymbol{\Theta}$.

### D.1  GRADIENT WITH RESPECT TO FEEDFORWARD WEIGHTS $\mathbf{W}_{i-1}$ (TYPE 1 NEURONS)

For Type 1 neurons in layer $i$ (for $i = 2, \ldots, L - 1$), the local loss $\mathcal{L}_i$ is defined as:

$$\mathcal{L}_i = \sum_{k=1}^{B} (1 - [\mathbf{C}_i]_{kk})^2 + \lambda \sum_{k=1}^{B} \sum_{j \neq k} \left( \max(0, [\mathbf{C}_i]_{kj}) \right)^2, \tag{14}$$

where $[\mathbf{C}_i]_{kj}$ is the cosine similarity between $\mathbf{v}_{i,k} = \mathbf{W}_{i-1}\mathbf{s}_{i-1,k}$ and $\hat{\mathbf{v}}_{i,j} = \mathbf{v}'_{i,j}$.

The gradient with respect to $\mathbf{W}_{i-1}$ is found via the chain rule. The resulting gradient is computed as a sum of outer products over the batch:

$$\frac{\partial \mathcal{L}_i}{\partial \mathbf{W}_{i-1}} = \sum_{k=1}^{B} \left( \frac{\partial \mathcal{L}_i}{\partial \mathbf{v}_{i,k}} \right) (\mathbf{s}_{i-1,k})^T \tag{15}$$

The gradient with respect to the basal voltage $\mathbf{v}_{i,k}$ is:

$$\frac{\partial \mathcal{L}_i}{\partial \mathbf{v}_{i,k}} = -2(1 - [\mathbf{C}_i]_{kk}) \frac{\partial [\mathbf{C}_i]_{kk}}{\partial \mathbf{v}_{i,k}} + \sum_{j \neq k} 2\lambda \max(0, [\mathbf{C}_i]_{kj}) \frac{\partial [\mathbf{C}_i]_{kj}}{\partial \mathbf{v}_{i,k}} \tag{16}$$

The derivative of the cosine similarity term is:

$$\frac{\partial [\mathbf{C}_i]_{kj}}{\partial \mathbf{v}_{i,k}} = \frac{1}{\|\mathbf{v}_{i,k}\|} \left( \frac{\hat{\mathbf{v}}_{i,j}}{\|\hat{\mathbf{v}}_{i,j}\|} - [\mathbf{C}_i]_{kj} \frac{\mathbf{v}_{i,k}}{\|\mathbf{v}_{i,k}\|} \right) \tag{17}$$

Combining these terms yields the final gradient for $\mathbf{W}_{i-1}$:

$$\begin{aligned}
\frac{\partial \mathcal{L}_i}{\partial \mathbf{W}_{i-1}} = \sum_{k=1}^{B} \Bigg[ &-2(1 - [\mathbf{C}_i]_{kk}) \frac{1}{\|\mathbf{v}_{i,k}\|} \left( \frac{\hat{\mathbf{v}}_{i,k}}{\|\hat{\mathbf{v}}_{i,k}\|} - [\mathbf{C}_i]_{kk} \frac{\mathbf{v}_{i,k}}{\|\mathbf{v}_{i,k}\|} \right) \\
&+ \sum_{j \neq k} 2\lambda \max(0, [\mathbf{C}_i]_{kj}) \frac{1}{\|\mathbf{v}_{i,k}\|} \left( \frac{\hat{\mathbf{v}}_{i,j}}{\|\hat{\mathbf{v}}_{i,j}\|} - [\mathbf{C}_i]_{kj} \frac{\mathbf{v}_{i,k}}{\|\mathbf{v}_{i,k}\|} \right) \Bigg] (\mathbf{s}_{i-1,k})^T
\end{aligned} \tag{18}$$

### D.2  GRADIENT WITH RESPECT TO BACKWARD WEIGHTS $\boldsymbol{\Theta}_i$ (TYPE 2 NEURONS)

For Type 2 neurons in layer $i$ (for $i = 2, \ldots, L - 1$), the local loss $\mathcal{L}'_i$ is defined symmetrically:

$$\mathcal{L}'_i = \sum_{k=1}^{B} (1 - [\mathbf{C}'_i]_{kk})^2 + \lambda \sum_{k=1}^{B} \sum_{j \neq k} \left( \max(0, [\mathbf{C}'_i]_{kj}) \right)^2, \tag{19}$$

where $[\mathbf{C}'_i]_{kj}$ is the cosine similarity between $\mathbf{v}'_{i,k} = \boldsymbol{\Theta}_i\mathbf{s}'_{i+1,k}$ and $\hat{\mathbf{v}}'_{i,j} = \mathbf{v}_{i,j}$.

The derivation for $\boldsymbol{\Theta}_i$ follows the same structure, yielding a sum of outer products over the batch:

$$\frac{\partial \mathcal{L}'_i}{\partial \boldsymbol{\Theta}_i} = \sum_{k=1}^{B} \left( \frac{\partial \mathcal{L}'_i}{\partial \mathbf{v}'_{i,k}} \right) (\mathbf{s}'_{i+1,k})^T \tag{20}$$

The gradient with respect to $\mathbf{v}'_{i,k}$ is:

$$\frac{\partial \mathcal{L}'_i}{\partial \mathbf{v}'_{i,k}} = -2(1 - [\mathbf{C}'_i]_{kk}) \frac{\partial [\mathbf{C}'_i]_{kk}}{\partial \mathbf{v}'_{i,k}} + \sum_{j \neq k} 2\lambda \max(0, [\mathbf{C}'_i]_{kj}) \frac{\partial [\mathbf{C}'_i]_{kj}}{\partial \mathbf{v}'_{i,k}}, \tag{21}$$

where the derivative of the cosine similarity term is:

$$\frac{\partial [\mathbf{C}'_i]_{kj}}{\partial \mathbf{v}'_{i,k}} = \frac{1}{\|\mathbf{v}'_{i,k}\|} \left( \frac{\hat{\mathbf{v}}'_{i,j}}{\|\hat{\mathbf{v}}'_{i,j}\|} - [\mathbf{C}'_i]_{kj} \frac{\mathbf{v}'_{i,k}}{\|\mathbf{v}'_{i,k}\|} \right) \tag{22}$$

This gives the final gradient for $\boldsymbol{\Theta}_i$:

$$
\begin{aligned}
\frac{\partial \mathcal{L}'_i}{\partial \boldsymbol{\Theta}_i} = \sum_{k=1}^{B} \Bigg[ & -2(1 - [\mathbf{C}'_i]_{kk}) \frac{1}{\|\mathbf{v}'_{i,k}\|} \left( \frac{\hat{\mathbf{v}}'_{i,k}}{\|\hat{\mathbf{v}}'_{i,k}\|} - [\mathbf{C}'_i]_{kk} \frac{\mathbf{v}'_{i,k}}{\|\mathbf{v}'_{i,k}\|} \right) \\
& + \sum_{j \neq k} 2\lambda \max(0, [\mathbf{C}'_i]_{kj}) \frac{1}{\|\mathbf{v}'_{i,k}\|} \left( \frac{\hat{\mathbf{v}}'_{i,j}}{\|\hat{\mathbf{v}}'_{i,j}\|} - [\mathbf{C}'_i]_{kj} \frac{\mathbf{v}'_{i,k}}{\|\mathbf{v}'_{i,k}\|} \right) \Bigg] (\mathbf{s}'_{i+1,k})^T
\end{aligned}
\tag{23}
$$

# E  EXPERIMENTAL SETTINGS

## E.1  STATISTICS OF DATASETS

We evaluate our proposed Bidirectional Spike-based Distillation (BSD) algorithm across a diverse range of tasks. The datasets employed in our experiments are detailed below, categorized by task.

**Image Classification.**  For the image classification tasks, we conducted experiments on five widely-used benchmarks to assess the model's performance in visual recognition.

- **MNIST.** The Modified National Institute of Standards and Technology (MNIST) dataset is a cornerstone benchmark comprising 60,000 training and 10,000 testing images of handwritten digits (0-9). Each image is a $28 \times 28$ pixel grayscale representation.

- **FashionMNIST.** As a more challenging drop-in replacement for MNIST, the FashionM-NIST dataset contains 60,000 training and 10,000 testing examples of clothing items and accessories across 10 classes, each being a $28 \times 28$ grayscale image.

- **Street View House Numbers (SVHN).** The SVHN dataset consists of $32 \times 32$ color images of house numbers from a real-world setting. We use the standard cropped version, which includes 73,257 digits for training and 26,032 for testing, categorized into 10 classes.

- **CIFAR-10.** The Canadian Institute for Advanced Research (CIFAR-10) dataset contains 50,000 training and 10,000 testing $32 \times 32$ color images across 10 mutually exclusive object classes (e.g., airplane, automobile, bird).

- **CIFAR-100.** The CIFAR-100 dataset is a collection of 60,000 $32 \times 32$ color images, designed for fine-grained object recognition tasks. It contains 100 distinct classes that are hierarchically grouped into 20 superclasses. For each class, there are 500 training images and 100 testing images.

**Image Generation.**  To evaluate the generative performance of BSD, we employed an autoencoder architecture for image reconstruction on the MNIST, FashionMNIST, and CIFAR-10 datasets. The quality of the generated images was quantitatively assessed using the Fréchet Inception Distance (FID) score, which measures the similarity between the distributions of the reconstructed and original images.

**Text Character Prediction.**  For the sequential prediction task, we utilized a natural language corpus to evaluate the model's ability to capture temporal dependencies at the character level.

- **Harry Potter Corpus.** This dataset is a text corpus derived from the Harry Potter book series (Plath et al., 2019), used for next-character prediction experiments.

**Time-Series Forecasting.**  For evaluating performance on multivariate time-series forecasting, we employed three standard real-world datasets from different domains.

- **Electricity.** The Electricity dataset (Lai et al., 2018) contains hourly electricity consumption data for 321 clients from 2012 to 2014, serving as a benchmark for long-term forecasting.

- **Metr-la.** This traffic forecasting dataset (Li et al., 2018) contains traffic speed data from 207 sensors on highways in Los Angeles County, aggregated every 5 minutes over a four-month period.

- **Pems-bay.** Sourced from the Caltrans Performance Measurement System (PeMS), this dataset (Li et al., 2018) comprises traffic speed information from 325 sensors in the San Francisco Bay Area over six months, providing another challenging forecasting benchmark.

## E.2 IMPLEMENTATION DETAILS

### E.2.1 BSD-MLPS

**Model Architecture.** For the feedforward path composed of Type 1 neurons, we design distinct Multi-Layer Perceptron (MLP) architectures tailored to each task, given the heterogeneity of input dimensions and class cardinalities across the five datasets. For each layer, the synaptic weight configurations of the backward path are designed to ensure that the number of neurons in both the Type 1 feedforward and Type 2 feedback paths remain consistent across all layers MNIST and FashionMNIST consist of single-channel $28 \times 28$ grayscale images, whereas SVHN, CIFAR-10, and CIFAR-100 comprise three-channel $32 \times 32$ color images. In terms of classification objectives, MNIST, FashionMNIST, SVHN, and CIFAR-10 are 10-class tasks, while CIFAR-100 involves 100 classes. To accommodate these differences and ensure optimal performance, we adopt dataset-specific MLP configurations. For MNIST and FashionMNIST, we employ a six-layer MLP with layer sizes [784, 1024, 1024, 512, 256, 10], where the input dimension of 784 corresponds to the flattened $28 \times 28 \times 1$ images and the output dimension of 10 reflects the number of classes. For SVHN and CIFAR-10, we adopt a wider six-layer MLP with [3072, 4096, 2048, 1024, 512, 10] neurons per layer, where the input dimension of 3072 is derived from flattening the $32 \times 32 \times 3$ images. For CIFAR-100, we maintain the same six-layer depth but adapt the final output to 100 classes, yielding layer sizes of [3072, 4096, 2048, 1024, 512, 100]. For the backward pathway, class labels are converted into spike-encoded targets: given a classification task with $C$ categories, the target for a sample of class $j$ is represented by a categorical code inspired by one-hot encoding, consisting of a binary vector of length $C$ with a single active entry at the $j$-th position and all others inactive. This vector is then repeated along the temporal dimension according to the number of time steps and delivered to the basal dendrites of top-layer Type 2 neurons. In all architectures, neuronal dynamics for membrane potential integration and spike generation are modeled using the Leaky Integrate-and-Fire (LIF) framework. The feedback path, composed of Type 2 neurons, follows a symmetric architecture to the feedforward path composed of Type 1 neurons.

**Training Hyperparameters.** All MLPs are optimized using AdamW with a cosine annealing learning-rate schedule. The simulation length is fixed at $T=4$ time steps. For LIF neurons, we employ the ATan surrogate function provided in the SpikingJelly framework. To ensure that spike sparsity remains within a regime conducive to effective learning, the firing thresholds are set to $0.2$ for Type 1 neurons and $0.1$ for Type 2 neurons. Across all datasets, we adopt a learning rate of $1 \times 10^{-4}$, a batch size of $128$, and apply RandAugment for data augmentation (Cubuk et al., 2020). To stabilize optimization, gradient clipping is applied with a threshold of $0.3$. Finally, for all non-top layers, the penalty weight in the ReCo loss is fixed at $\lambda=0.6$.

### E.2.2 BSD-CNNS

**Model Architecture.** For the feedforward path composed of Type 1 neurons, we adopt a unified CNN architecture for all tasks. The architecture consists of five convolutional layers followed by a fully connected output layer. Specifically, the first convolutional layer maps the input channels to 128 channels using $3 \times 3$ kernels, followed by $2 \times 2$ max-pooling. The second convolutional layer maintains 128 channels with the same $3 \times 3$ kernel and $2 \times 2$ max-pooling. The third and fourth convolutional layers expand the representation to 256 channels each, again using $3 \times 3$ kernels with $2 \times 2$ max-pooling. The fifth convolutional layer further increases the dimensionality to 512 channels, also with $3 \times 3$ kernels and $2 \times 2$ max-pooling. The feature maps are then flattened into a 512-dimensional vector, which is connected to the final output layer whose dimension is aligned with the spike-encoded target $\hat{s}$.

As in the MLP case, class labels are converted into spike-coded targets and delivered to the basal dendrites of top-layer Type 2 neurons. For a classification task with $C$ categories, the target of a sample from class $j$ is represented by a categorical spike code inspired by one-hot encoding: a binary

vector of length $C$ with a single active entry corresponding to class $j$, repeated along the temporal dimension to match the simulation time steps.

To stabilize neuronal spiking activity across layers, we insert a batch-normalization layer after each convolutional operation. This normalization constrains the membrane potentials to a comparable range across layers, thereby ensuring that spike sparsity remains consistent and that neuronal firing thresholds are properly regulated throughout the network.

Similarly to the MLP case, the feedback path, composed of Type 2 neurons, mirrors the structure of the feedforward path composed of Type 1 neurons. In the feedback path, we use upsampling as the reverse operation of the pooling used in the feedforward path, ensuring that the number of Type 1 and Type 2 neurons in feedforward and feedback paths remains equal across all layers.

**Training Hyperparameters.** For all tasks, we optimize our models using the AdamW optimizer and employ a cosine learning rate scheduler. To ensure that the neuronal firing sparsity remains within an ideal range, we set the firing thresholds of both Type 1 and Type 2 neurons to 1.0. The number of time steps is set to 4, with a warm-up period of 100 steps. For data augmentation, we use RandAugment with a magnitude of 4 for MNIST, FashionMNIST, CIFAR-10, and CIFAR-100, while on SVHN, we apply only RandomCrop. All images are normalized prior to processing. The batch size is set to 128 across all datasets. For MNIST, FashionMNIST, CIFAR-10, and CIFAR-100, the learning rates for both Type 1 and Type 2 neurons are set to $1 \times 10^{-3}$, whereas for SVHN, we reduce the learning rate to $1 \times 10^{-4}$. For the layer-wise ReCo loss applied to all layers except the top layer, we set the penalty weight $\lambda$ to 0.6, consistent with the configuration used in MLPs.

### E.2.3 BSD-RNNs

**Model Architecture.** For all sequential regression datasets, we employed the same RNN model architecture which is a single-layer RNN with a hidden size of 300, where the input and output dimensions corresponded to those of the respective datasets: Harry Potter(103,103), Electricity(320,1), Metr-la(206,1), Pems-bay(324,1), where the dimensions are presented in the form of (input dimensions, output dimensions).

**Training Hyperparameters.** Similarly to the classification tasks, we utilized the AdamW optimizer along with a cosine learning rate scheduler across all sequential regression datasets. To identify optimal performance in different datasets, we maintained identical activation thresholds for both Type 1 and Type 2 neurons and fixed the number of timesteps at 4 for all tasks. However, we varied the activation threshold according to each dataset: 1.0 for the Harry Potter and Metr-la datasets, 0.8 for Electricity, and 0.5 for Pems-bay. All weights were initialized with PyTorch's default weight initialization. The batch size was set to 128 for the Electricity dataset and 64 for the remaining sequential regression datasets, with the sequence length fixed to 32 across all datasets. The learning rate was set to 0.001 for the Harry Potter, Electricity and Metr-la datasets, and 0.00015 for Pems-bay. Due to the larger dataset size of Harry Potter, training was limited to 10 epochs, whereas the other datasets were trained for 100 epochs. To stabilize training, gradient clipping with an upper bound of 1 was employed for all sequential regression datasets, and batch normalization was applied to the hidden layer in Pems-bay to normalize each batch to zero mean and unit variance.

**Metric.** For time-series forecasting datasets (Electricity, Metr-la, and Pems-bay), the mean squared error (MSE) and the mean absolute error (MAE) were used as performance metrics. The formulas are as follows:

$$\text{MSE} = \frac{1}{B \times T \times D} \sum_{b=1}^{B} \sum_{t=1}^{T} \sum_{d=1}^{D} \left( \hat{y}_{b,t,d} - y_{b,t,d} \right)^2, \tag{24}$$

$$\text{MAE} = \frac{1}{B \times T \times D} \sum_{b=1}^{B} \sum_{t=1}^{T} \sum_{d=1}^{D} \left| \hat{y}_{b,t,d} - y_{b,t,d} \right|, \tag{25}$$

where $B, T, D$ represents the batch size, sequence length, and output dimensions, respectively. While $y_{b,t,d}$ is the value of the $d$-th feature of the output at the $t$-th time step for the $b$-th sample in the batch and $\hat{y}_{b,t,d}$ is its target value. We normalized each column of the raw data to zero mean and unit variance before calculating MSE and MAE, consistent with the preprocessing described in (Lv et al.,

2025). We selected the model with the lowest MSE as the final recorded performance metric. For the text character prediction dataset (Harry Potter), element-wise accuracy was used as the performance metric. In preprocessing, all whitespace characters in the text were replaced with a single space, and performance was evaluated on the processed dataset.

### E.2.4 BSD-AUTOENCODERS

**Model Architecture.** We address the task of image reconstruction on the MNIST, FashionMNIST, and CIFAR-10 datasets using a convolutional autoencoder. The architecture, which remains consistent across all datasets, is composed of a feedforward path (Type 1 neurons) and a symmetric feedback path (Type 2 neurons). To maintain an identical neuron count between the Type 1 and Type 2 populations at each corresponding layer, the architecture employs inverse spatial operations: where one path uses a strided convolution, the other utilizes an upsampling layer, and vice versa. To ensure stable and sparse spiking activity, each layer in both paths incorporates a batch normalization step after convolution and upsampling.

The encoder section systematically compresses the input into a latent representation through a cascade of three convolutional layers. The first layer applies a $3 \times 3$ convolution with a stride of 2, halving the input's spatial dimensions while expanding the feature maps to 128 channels. The second layer repeats this operation, again using a stride of 2 to halve the spatial resolution and increasing the channel depth from 128 to 256. The final encoding layer performs a third stride-2 convolution, mapping the features to a 512-channel bottleneck representation.

Symmetrically, the decoder reconstructs the image from this bottleneck representation. Each decoding layer first doubles the spatial dimensions of its input via a nearest-neighbor upsampling operation. Following this, a $3 \times 3$ convolution with a stride of 1 is applied to refine the features. This two-step process is repeated three times: the first block reduces the channel count from 512 to 256, the second from 256 to 128, and the final block restores the feature map to the original input resolution while matching the channel count of the source image.

The source image is provided as the basal input to both the bottom-layer Type 1 neurons and the top-layer Type 2 neurons. Thus, the top-layer Type 2 neurons function as a dedicated transducer for the backward path, converting the source image into a spike-based representation.

**Training Hyperparameters.** The model is optimized across all datasets using the AdamW optimizer, with a learning rate of $0.001$ managed by a cosine annealing schedule. The spiking threshold for both Type 1 and Type 2 neurons is uniformly set to $0.4$, and the network is simulated for a total of 8 timesteps. To apply differential penalties to the reconstruction error, we decompose the error in the frequency domain. This is achieved by creating a binary mask based on the image's Fourier transform. A circular region in the frequency spectrum, centered at the zero frequency, is defined as the low-frequency domain, while the area outside this region constitutes the high-frequency domain. The boundary is controlled by a `freq_cutoff_ratio` hyperparameter, representing the normalized radius of this low-frequency circle. This allows us to assign distinct $\lambda$ weights for the Reconstruction-on-Construction (ReCo) loss to the low-frequency and high-frequency components. For the MNIST and FashionMNIST datasets, the model is trained with a batch size of 32. The `freq_cutoff_ratio` is set to $0.6$, with the ReCo loss $\lambda$ configured to $0.005$ for low-frequency components and $0.01$ for high-frequency components. For the more complex CIFAR-10 dataset, the batch size is increased to 64. The `freq_cutoff_ratio` is adjusted to $0.7$, and the corresponding ReCo loss weights are set to $0.005$ for the low-frequency band and increased to $0.05$ for the high-frequency band, placing a greater penalty on the reconstruction of fine-grained details. To improve model generalization on this dataset, we also apply RandAugment for data augmentation during training.

**Evaluation Metrics** For autoencoder-based image generation tasks, we employ Fréchet Inception Distance (FID) (Heusel et al., 2017) as our evaluation metric. FID measures the distributional similarity between real and generated images by computing the distance between their feature representations in the Inception-v3 network's activation space. Specifically, the metric calculates the Fréchet distance between two multivariate Gaussian distributions fitted to the feature vectors of real and synthetic images extracted from the final pooling layer of Inception-v3. Lower FID scores indicate higher similarity between generated and real image distributions, with a perfect score

of 0 representing identical distributions. This metric provides a reliable quantitative assessment of generation quality that correlates well with human perceptual judgment, making it particularly suitable for evaluating the generative capabilities of our BSD-trained autoencoders.

# F    ADDITIONAL EXPERIMENTS

## F.1    SUPPLEMENTARY RESULTS FOR IMAGE GENERATION

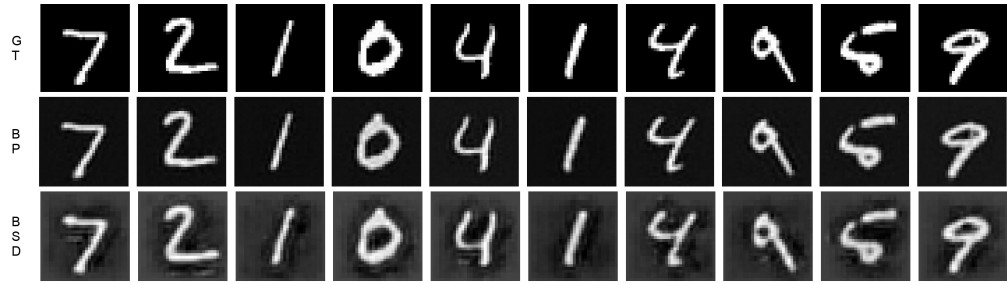

Figure 4:  Comparison of image reconstruction on the MNIST dataset by autoencoders trained with Backpropagation (BP) and our proposed Bidirectional Spike-based Distillation (BSD).

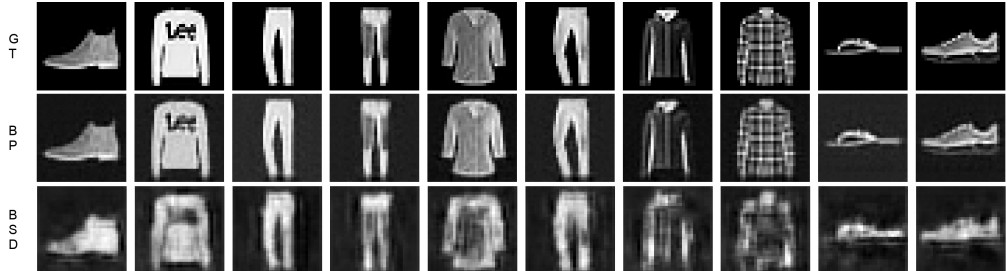

Figure 5:  Visual comparison of FashionMNIST reconstructions from autoencoders trained with Backpropagation (BP) and our Bidirectional Spike-based Distillation (BSD) method..

Figures 4 and 5 present a qualitative comparison of image reconstructions on the MNIST and Fashion-MNIST datasets, showcasing outputs from autoencoders trained with conventional Backpropagation (BP) alongside those trained with our proposed Bidirectional Spike-based Distillation (BSD). Table 5 provides a complementary quantitative analysis on the FashionMNIST dataset, benchmarking the performance of BSD against BP and the Fully Spiking Variational Autoencoder (FSVAE) using the Fréchet Inception Distance (FID) metric. While BSD-trained autoencoders demonstrate proficient generative capabilities by successfully capturing the salient global structure of the input images, compared to BP-trained ANNs, BSD-generated images tend to exhibit a loss of high-frequency detail. This suggests that while BSD establishes a robust framework for generative modeling, exploring further refinements or architectural adaptations could be beneficial for enhancing the preservation of fine-grained textures and improving the overall sharpness of the generated images.

Table 5: Quantitative comparison of image generation performance on the FashionMNIST dataset, evaluated using Fréchet Inception Distance (FID). ↓ denotes that lower scores indicate better performance.

| Dataset | Model | FID ↓ |
|---|---|---|
| | ANN-BP | **29.07** |
| FashionMNIST | FSVAE | 90.12 |
| | **BSD (Ours)** | **112.97** |

Table 6: Ablation study on text character prediction and time-series forecasting tasks.

| Method | Harry Potter | Electricity | | Metr-la | | Pems-bay | |
|---|---|---|---|---|---|---|---|
| | Pred. Acc. ↑ | MSE ↓ | MAE ↓ | MSE ↓ | MAE ↓ | MSE ↓ | MAE ↓ |
| BSD-MSE | 29.6% | 0.179 | 0.328 | 0.160 | 0.247 | 0.180 | 0.221 |
| BSD-InfoNCE | 29.9% | 0.172 | 0.323 | 0.159 | 0.257 | 0.190 | 0.230 |
| **BSD** | **41.8%** | **0.165** | **0.314** | **0.125** | **0.197** | **0.174** | **0.206** |

## F.2 ABLATION STUDY ON RNNs

Table 6 compares the performance of BSD-ReCo, BSD-MSE, and BSD-infoNCE on sequential regression tasks. In all tasks, the ReCo loss consistently demonstrated the highest performance. In contrast to the outcomes from the ablation studies on image classification tasks, all inter-layer loss functions achieved stable convergence on the sequential regression datasets, which may be attributed to the relatively shallow depth of the RNNs.

In conclusion, for sequential regression tasks, MSE, infoNCE, and ReCo all demonstrated stable convergence, but ReCo remains the most optimal inter-layer loss, with its performance advantage being particularly evident on text character prediction datasets.

## F.3 ABLATION STUDY ON AUTOENCODERS

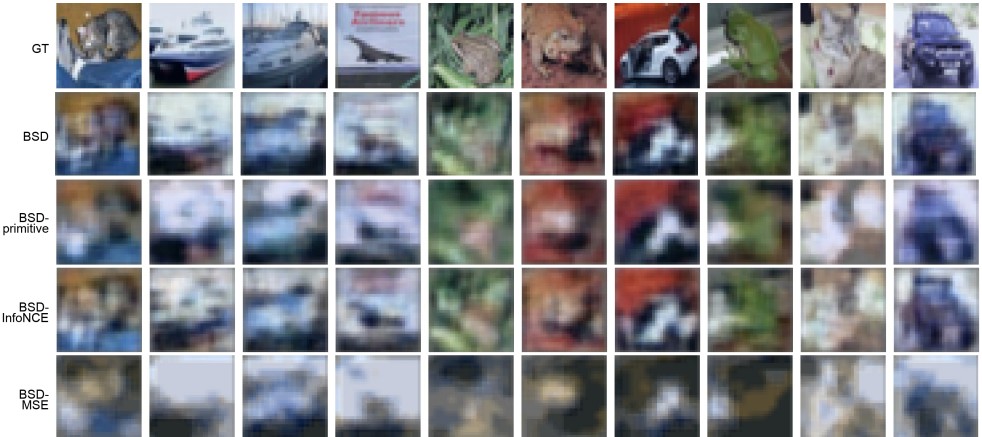

Figure 6: Visual examples of image reconstructions on the CIFAR-10 dataset, showcasing the performance of autoencoders trained with BSD using various layer-wise loss function configurations.

| Dataset | Model | FID ↓ |
|---|---|---|
| CIFAR-10 | **BSD** | **168.12** |
| | BSD-primitive | 185.33 |
| | BSD-InfoNCE | 190.95 |
| | BSD-MSE | 177.66 |

Table 7: Ablation study on CIFAR-10 image generation, evaluating the impact of different layer-wise loss functions and frequency decomposition strategies, with performance measured by Fréchet Inception Distance (FID). ↓ denotes that lower FID values indicate better performance. **BSD-primitive**: BSD without adaptive frequency domain decomposition. **BSD-InfoNCE**: BSD employing InfoNCE loss for intra-neuronal voltage alignment. **BSD-MSE**: BSD utilizing Mean Squared Error (MSE) loss for intra-neuronal voltage alignment.

To investigate the impact of different layer-wise loss functions and frequency decomposition strategies on image generation, we conduct an ablation study using BSD-trained autoencoders on the CIFAR-10 dataset. Figure 6 presents visual comparisons of image reconstructions, and Table 7 details the corresponding FID scores. Our ablation study includes the following configurations: **BSD**: our proposed method incorporating ReCo loss and adaptive frequency decomposition; **BSD-primitive**: BSD using ReCo loss but without adaptive frequency domain decomposition, applying a uniform $\lambda$ across all frequency components; **BSD-InfoNCE**: BSD that substitutes ReCo loss with InfoNCE loss for intra-neuronal voltage alignment; and **BSD-MSE**: BSD that replaces ReCo loss with Mean Squared Error (MSE) loss for intra-neuronal voltage alignment. From Table 7, it is evident that performing frequency domain segmentation using FFT and applying distinct $\lambda$ penalty weights for low and high-frequency regions significantly enhances the performance of BSD-trained autoencoders. As a result, BSD achieves the best FID score of $\mathbf{168.12}$, notably outperforming BSD-primitive (FID 185.33). Furthermore, for image generation tasks, BSD-InfoNCE (FID 190.95), which employs InfoNCE for intra-neuronal voltage alignment, yields inferior results compared to BSD (using ReCo Loss for intra-neuronal voltage alignment). Figure 6 shows that while BSD-MSE demonstrates an ability to capture salient structural features and approximate pixel locations of objects, it nonetheless exhibits obvious deficiencies in both color fidelity and detail retention. Compared to the reconstructions from BSD (which utilizes ReCo Loss and adaptive frequency decomposition), the images produced by BSD-MSE appear significantly blurred and desaturated, losing fine textures and distinct chromatic attributes present in the ground truth (GT) images. These results collectively affirm the effectiveness of our proposed design, which integrates ReCo Loss as the layer-wise loss function, performs frequency domain segmentation via FFT, and applies adaptive $\lambda$ penalty weights for distinct low and high-frequency regions.

## G  DYNAMICS OF SYNAPTIC WEIGHT ALIGNMENT

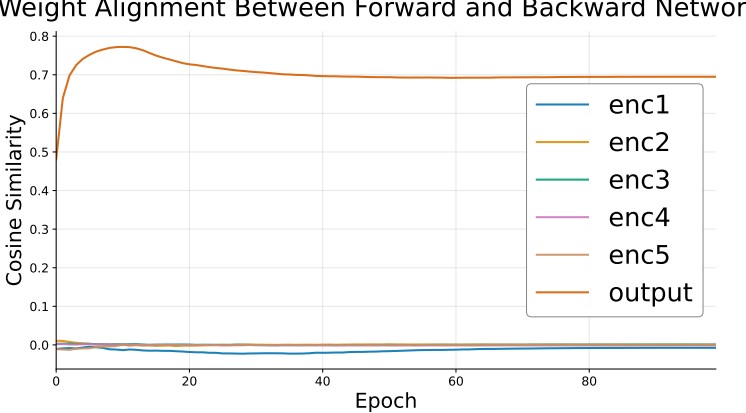

Figure 7: Alignment dynamics of synaptic weights ($\mathbf{W}$ and $\boldsymbol{\Theta}$) between the feedforward and backward pathways in a BSD-trained convolutional neural network for image classification on the CIFAR-10 dataset. The figure presents the cosine similarity of weights across network layers as a function of training epochs.

In this section, we examine the alignment dynamics between the feedforward synaptic weights ($\mathbf{W}$) and the backward synaptic weights ($\boldsymbol{\Theta}$) within a convolutional neural network trained using the BSD algorithm. The network was tasked with image classification on the CIFAR-10 dataset. We measured the cosine similarity between the weights connected to the basal dendrites of Type 1 and Type 2 neurons to quantify their alignment in each layer. In Figure 7, "enc1" through "enc5" denote the five feedforward convolutional layers and their corresponding backward upsampling layers, progressing from the shallowest to the deepest. The "output" label refers to the final fully connected layer. Figure 7 reveals that the weight alignment for the final fully connected layer undergoes a rapid initial increase, peaking within the first few epochs before stabilizing at a high cosine similarity value of approximately 0.7. In contrast, the weights of the convolutional and upsampling layers ("enc1" to "enc5") consistently exhibit negligible alignment, with their cosine similarity values remaining close to zero throughout the entire training process. Such alignment behavior demonstrates that

the feedforward and backward pathways do not converge toward symmetric weights under BSD. Thereby we confirm that the BSD algorithm adheres to criterion **C1**: Asymmetric synaptic weights for feedforward and feedback pathways.

## H   T-SNE VISUALIZATION OF LEARNED FEATURE REPRESENTATIONS IN BSD-TRAINED CNNS

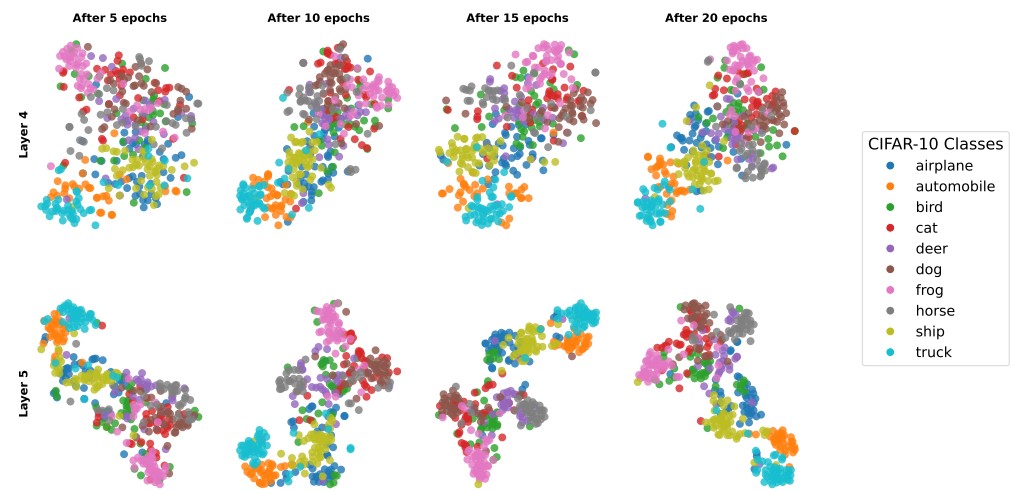

Figure 8:  t-SNE visualizations illustrating the evolution of learned feature representations from the final two convolutional layers of the feedforward path in BSD-trained CNNs, for an image classification task on the CIFAR-10 dataset across training epochs.

Figure 8 shows t-distributed stochastic neighbor embedding (t-SNE) projections of feature representations from the final two convolutional layers (Layers 4 and 5) of the feedforward path at epochs 5, 10, 15, and 20 during BSD training on CIFAR-10. As training proceeds, class-specific structure becomes increasingly separated; by epoch 20, Layer 5 exhibits distinct, compact clusters for most classes. This progression indicates that BSD progressively shapes intermediate features into more class-discriminative representations.

## I   EVALUATION ON TINY-IMAGENET

To evaluate the scalability of our proposed algorithm, we conducted experiments on the Tiny-ImageNet dataset. We compared our proposed BSD algorithm against both a standard backpropagation (BP) baseline and Dendritic Localized Learning (DLL). For a fair comparison, we used the same 5-layer CNN architecture across all three methods.

Table 8: Performance comparison on the Tiny-ImageNet dataset.

| Method | Accuracy (%) |
|---|---|
| BP | 41.07 |
| Dendritic Localized Learning (DLL) | 17.10 |
| **BSD (Ours)** | **35.34** |

The results are summarized in Table 8.  As shown, our BSD algorithm achieves an accuracy of 35.34%, substantially outperforming the other biologically plausible method, DLL.

## J   ABLATION STUDY ON BATCH SIZE

To investigate the sensitivity of our BSD framework to the batch size, we conducted an ablation study using the BSD-trained CNN across the SVHN, CIFAR-10, and CIFAR-100 datasets. All other

hyperparameters were kept consistent with the main experiments detailed in Appendix E.2. The results are summarized in Table 9.

Table 9: Impact of varying batch size on the performance of the BSD-trained CNN across multiple datasets.

| Batch Size | SVHN (%) | CIFAR-10 (%) | CIFAR-100 (%) |
|---|---|---|---|
| 16 | 71.55 | 68.96 | 34.99 |
| 32 | 78.18 | 78.07 | 42.53 |
| 64 | 90.97 | 81.54 | 49.15 |
| 128 | 90.81 | 84.53 | 53.11 |
| 256 | 88.97 | 83.85 | 53.49 |

The results demonstrate a clear trend: the performance of our BSD framework generally improves as the batch size increases. At smaller batch sizes, the model's performance is limited. This is an expected behavior for contrastive learning methods like ReCo, which require a sufficiently large and diverse set of negative samples within each batch to effectively shape the representation space. As the batch size increases, we observe substantial performance gains across all datasets, as larger batches provide a more robust estimate of the data distribution and a richer set of negative examples for the contrastive objective. We also observe that once the batch size reaches 128, the performance begins to stabilize. Further increasing the batch size to 256 yields minor fluctuations in accuracy. This suggests that a batch size of approximately 128 provides a sufficient number of negative samples for the model to achieve robust performance on these datasets. These findings confirm that our BSD framework scales favorably with batch size, which is consistent with the principles of contrastive learning.

## K  ENERGY CONSUMPTION ANALYSIS

To quantify the energy efficiency of our trained models, we conducted a theoretical analysis of energy consumption during inference. The primary energy advantage of SNNs is realized in this phase, particularly on specialized neuromorphic hardware. Our BSD algorithm is a training methodology; the resulting model used for inference is a standard SNN. Therefore, a model trained with BSD fully retains the inherent energy-saving characteristics of SNNs.

We estimate the theoretical energy consumption per sample by following the methodology proposed by Yao et al. (2023b). For a given SNN layer $l$, the energy consumption is estimated as:

$$\text{Energy}_{\text{SNN}}(l) = E_{\text{AC}} \times (T \times \gamma_l \times \text{FLOPs}(l)), \tag{26}$$

where $T$ is the number of timesteps, $\gamma_l$ is the layer's average firing rate, and $E_{\text{AC}}$ is the energy per accumulate operation. For an equivalent ANN layer $l$, the formula is:

$$\text{Energy}_{\text{ANN}}(l) = E_{\text{MAC}} \times \text{FLOPs}(l), \tag{27}$$

where $E_{\text{MAC}}$ is the energy per multiply-accumulate operation. The total energy for each model is the sum over all layers. We use the energy constants for a 45nm process from the literature ($E_{\text{AC}} = 0.9$ pJ, $E_{\text{MAC}} = 4.6$ pJ) for our analysis (Yao et al., 2023b). The analysis was performed on our BSD-trained 5-layer CNN and its equivalent ANN counterpart on the CIFAR-10 task. A layer-wise breakdown of the energy consumption for both models is provided in Table 10 and Table 11, with a final summary in Table 12.

Table 10: Layer-wise energy consumption analysis for the BSD-trained SNN during inference on a single sample from CIFAR-10.

| Layer Name | FLOPs | Avg Firing Rate ($\gamma$) | SOPs | Energy (J) |
|---|---|---|---|---|
| Layer 1 (Conv2d) | 3.54e+06 | 0.5006 | 7.09e+06 | 6.38e-06 |
| Layer 2 (Conv2d) | 3.77e+07 | 0.1519 | 2.29e+07 | 2.06e-05 |
| Layer 3 (Conv2d) | 1.89e+07 | 0.1815 | 1.37e+07 | 1.23e-05 |
| Layer 4 (Conv2d) | 9.44e+06 | 0.1943 | 7.34e+06 | 6.60e-06 |
| Layer 5 (Conv2d) | 4.72e+06 | 0.2037 | 3.85e+06 | 3.46e-06 |
| Layer 6 (Linear) | 5.12e+03 | 0.0803 | 1.65e+03 | 1.48e-09 |

Table 11: Layer-wise energy consumption analysis for the equivalent ANN-BP model during inference on a single sample from CIFAR-10.

| Layer Name | FLOPs | Energy (J) |
|---|---|---|
| Layer 1 (Conv2d) | 3.54e+06 | 1.63e-05 |
| Layer 2 (Conv2d) | 3.77e+07 | 1.74e-04 |
| Layer 3 (Conv2d) | 1.89e+07 | 8.68e-05 |
| Layer 4 (Conv2d) | 9.44e+06 | 4.34e-05 |
| Layer 5 (Conv2d) | 4.72e+06 | 2.17e-05 |
| Layer 6 (Linear) | 5.12e+03 | 2.36e-08 |

Table 12: Total energy consumption comparison per inference sample.

| Model | Total Operations per Sample | Energy per Sample (J) | Energy Reduction |
|---|---|---|---|
| ANN (BP) | 7.43e+07 (FLOPs) | 3.42e-04 | - |
| SNN (BSD) | 5.49e+07 (SOPs) | 4.94e-05 | 85.5% ↓ |

This quantitative analysis confirms the substantial energy advantage of the SNN model. By leveraging sparse, event-driven computation, the SNN trained with our BSD method is approximately 85.5% more energy-efficient during inference than its architecturally identical ANN counterpart. This highlights that our biologically plausible training method yields models that are not only performant but also highly efficient for deployment.

## L  ROBUSTNESS TO INPUT NOISE

To evaluate the robustness of our BSD framework against corrupted data, we conducted an experiment to assess the resilience of our algorithm to input noise. We used the model checkpoints pre-trained on the clean training sets of CIFAR-10 and SVHN. During the inference phase, we introduced corruption by adding random Gaussian noise (mean=0, std=0.05) to the input images. We then evaluated the performance of these clean-trained models on the noisy test data and compared it to a standard backpropagation (BP) baseline. The same 5-layer CNN architecture was used for both methods to ensure a fair comparison.

The results, summarized in Table 13, show the accuracy on both the original clean test set and the corrupted test set.

As expected, the performance of both methods degrades under noisy conditions. On both CIFAR-10 and SVHN, the drop in accuracy for BSD is comparable to that of BP. The results indicate that our approach learns robust and effective features.

We attribute this competitive robustness to the dual-objective nature of the BSD training process. The final task loss at the output layer pushes the network to extract discriminative information, while the layer-wise alignment objective simultaneously encourages the preservation of rich generative

Table 13: Comparison of model accuracy on clean and noisy test data for CIFAR-10 and SVHN. Models were trained only on clean data. "Noisy Accuracy" corresponds to Accuracy Under Attack (AUA).

| Dataset | Method | Clean Accuracy (%) | Noisy Accuracy (%) |
|---------|--------|--------------------|--------------------|
| CIFAR-10 | BP | 87.18 | 70.89 |
|          | BSD | 83.67 | 67.06 |
| SVHN | BP | 94.31 | 89.24 |
|      | BSD | 90.81 | 85.40 |

information from the input. This balance between two complementary goals acts as a form of regularization, guiding the model to learn more generalizable representations that are inherently more resilient to input perturbations, rather than learning brittle features that are only optimal for the clean data distribution.

## M  FFT DECOMPOSITION FOR ADAPTIVE LOSS

The FFT decomposition, used in our generation tasks, is a method to separate the low-frequency and high-frequency components of an image or a membrane voltage map, allowing us to apply an adaptive loss function. The process is as follows:

First, we apply a 2D Fast Fourier Transform to the spatial tensor (e.g., a batch of membrane voltage maps of shape $[B, C, H, W]$), converting it into its frequency-domain representation.

Second, to isolate these components, we construct frequency-domain masks. A low-pass filter is created by defining a circular region centered at the zero-frequency origin of the spectrum. Frequencies falling inside this radius are designated as low-frequency components, which represent the global structure and smooth areas of the image. Conversely, all frequencies outside this radius are designated as high-frequency components, which correspond to edges and fine-grained textures.

Finally, by element-wise multiplying the frequency-domain representation with these two masks, we separate it into two distinct tensors: one containing only low frequencies and another containing only high frequencies. This separation enables the adaptive loss computation mentioned in the main text. We calculate the ReCo loss independently on these two components but use a different penalty weight $\lambda$ for each. A smaller $\lambda$ is applied to the low-frequency components to maintain structural coherence, while a larger $\lambda$ is applied to the high-frequency components to more strongly enforce the preservation of sharp details and edge fidelity. This approach allows the model to better preserve fine details without sacrificing the image's structural coherence.

## N  COMPUTATIONAL COST ANALYSIS

To provide a comprehensive assessment of the computational costs associated with our proposed BSD algorithm, we conducted a comparative analysis against a standard Backpropagation (BP) baseline for SNNs. All experiments were performed on the CIFAR-10 dataset using a single NVIDIA GeForce RTX 2080 Ti GPU. We profiled memory consumption across different batch sizes to evaluate scalability. For both methods, we employed the 5-layer convolutional network architecture detailed in Appendix E.2 to ensure a fair comparison. The results for inference and training are presented separately below.

**Inference Performance.**   During inference for classification tasks, the BSD framework utilizes only its feedforward pathway (weights $\mathbf{W}$). Since this pathway is architecturally identical to a BP-trained SNN of the same configuration, the computational graph and parameters used during a forward pass are the same. As empirically confirmed in Table 14, this results in identical memory consumption across various batch sizes. This demonstrates that our biologically plausible training method introduces no computational overhead during the inference phase.

Table 14: Comparison of memory consumption during inference on CIFAR-10 across varying batch sizes.

| Batch Size | Method | Inference Memory Consumption (MB) |
|:---:|:---:|:---:|
| 32 | BP | 344 |
|  | BSD | 344 |
| 64 | BP | 944 |
|  | BSD | 944 |
| 128 | BP | 1826 |
|  | BSD | 1826 |

**Training Performance.** During training, BSD must maintain two sets of weights ($\mathbf{W}$ and $\mathbf{\Theta}$) and store the activations for both the feedforward and backward pathways. Furthermore, the ReCo loss constructs a $B \times B$ affinity matrix, which introduces an additional memory usage component that scales quadratically with the batch size ($O(B^2)$). However, the primary driver of BSD's increased memory consumption is the necessity of storing the parameters and full activation maps for the second (backward) pathway, a characteristic inherent to our dual-network design. As detailed in Table 15, this architectural requirement leads to higher memory consumption compared to BP, which only stores one set of weights and the activations required for its backward pass.

Table 15: Comparison of memory consumption during training on CIFAR-10 across varying batch sizes.

| Batch Size | Method | Training Memory Consumption (MB) |
|:---:|:---:|:---:|
| 32 | BP | 950 |
|  | BSD | 1154 |
| 64 | BP | 1844 |
|  | BSD | 2852 |
| 128 | BP | 3324 |
|  | BSD | 5034 |

**Time Complexity Analysis.** The empirical results are consistent with the theoretical time complexity of the algorithms. For a network with $L$ layers, $T$ timesteps, a batch size of $B$, and an average layer width of $D$, the complexities are as follows:

- **BPTT:** The complexity is $O(L \cdot T \cdot B \cdot D^2)$, derived from the forward pass and a symmetric backward pass through the unrolled computation graph.
- **BSD:** The complexity is $O(L \cdot T \cdot B \cdot D^2 + L \cdot B^2 \cdot D)$. The first term, $O(L \cdot T \cdot B \cdot D^2)$, accounts for propagation through the two pathways and is comparable to BPTT. The second term, $O(L \cdot B^2 \cdot D)$, arises from computing the $B \times B$ affinity matrix for the layer-wise ReCo loss, making our training complexity more sensitive to batch size.

## O  ABLATION STUDY ON THE NUMBER OF TIMESTEPS

To more comprehensively investigate the influence of the number of timesteps ($T$) on our model's performance, we expanded our ablation study across multiple datasets for both our CNN and RNN architectures. All other hyperparameters were kept consistent with those described in our main experiments. The results are summarized in Table 16 for image classification tasks and Table 17 for sequential regression tasks.

For both CNN and RNN models, performance generally improves when increasing the number of timesteps from $T = 2$ to $T = 4$. For timesteps greater than four ($T \geq 4$), performance tends to stabilize, exhibiting only minor fluctuations across most datasets and metrics. This suggests that four timesteps provide a robust and effective balance, allowing neurons to integrate sufficient information

Table 16: Impact of varying timesteps ($T$) on the performance of BSD-trained CNNs across image classification datasets.

| Timesteps ($T$) | CIFAR-100 (%) | SVHN (%) | CIFAR-10 (%) |
|---|---|---|---|
| 2 | 47.12 | 82.54 | 78.09 |
| 4 | 53.48 | 90.81 | 84.53 |
| 6 | 53.11 | 90.35 | 82.28 |
| 8 | 53.24 | 90.94 | 81.51 |

Table 17: Impact of varying timesteps ($T$) on the performance of BSD-trained RNNs across sequential regression tasks.

| Timesteps ($T$) | Harry Potter (acc ↑) | Metr-la (mse ↓) |
|---|---|---|
| 2 | 0.4185 | 0.1280 |
| 4 | 0.4169 | 0.1245 |
| 6 | 0.4168 | 0.1248 |
| 8 | 0.4212 | 0.1238 |

to form rich representations without introducing excessive temporal complexity. These expanded findings empirically justify our choice of $T = 4$ as a robust and efficient setting across a variety of tasks and architectures.

## P    ABLATION STUDY ON BATCH NORMALIZATION

To more broadly investigate the specific impact of batch normalization (BN), we expanded our ablation study to include the SVHN, CIFAR-10, and CIFAR-100 datasets. The experiments were conducted on our BSD-trained CNN, and the results are presented in Table 18.

Table 18: Impact of Batch Normalization (BN) on the performance of the BSD-trained CNN across multiple datasets.

| Method | SVHN (%) | CIFAR-10 (%) | CIFAR-100 (%) |
|---|---|---|---|
| BSD without BN | 80.68 | 80.75 | 53.31 |
| BSD with BN | 90.81 | 84.53 | 53.48 |

The results show that incorporating BN improves performance across all tested datasets. This benefit is attributed to the ability of BN to stabilize the distribution of membrane potentials across layers. The normalization ensures that spike sparsity remains consistent, leading to more stable and effective learning.

## Q    JUSTIFICATION FOR THE CHOICE OF RECO LOSS

Here, we provide a more detailed justification for our choice of the Relaxed Contrastive (ReCo) loss over other contrastive alternatives like InfoNCE.

Our ablation study in Section 5.6 provides the empirical evidence, showing that BSD with ReCo loss consistently and significantly outperforms BSD-InfoNCE. This is particularly evident on complex datasets like CIFAR-100, where the performance margin is over 15 percentage points. The theoretical rationale for this superiority, especially for aligning the voltage signals in our BSD algorithm, is as follows.

The core objective at each hidden layer is to align the pre-spike membrane potential from the feedforward path ($\mathbf{v}_{i,k}$) with its corresponding supervisory voltage from the backward path ($\hat{\mathbf{v}}_{i,k}$). Simultaneously, the network must distinguish this pair from all negative pairs, where the voltage $\mathbf{v}_{i,k}$ is paired with supervisory signals from other samples in the batch ($\hat{\mathbf{v}}_{i,j}$ for $j \neq k$).

A standard InfoNCE loss enforces this distinction by applying a uniform repulsive force to all negative pairs. This compels the membrane voltage pattern generated for one sample to be strictly anti-correlated with the supervisory voltage patterns of all other samples. This is an overly restrictive constraint for high-dimensional membrane potentials, which can make the learning task unnecessarily difficult and potentially warp the feature space.

In contrast, the ReCo loss adopts a more targeted and flexible objective. It applies a penalty only when the voltage for one sample is confusingly similar to the supervisory signal of another (i.e., has a positive cosine similarity). If a negative pair's voltage representations are already orthogonal or dissimilar, they incur no loss penalty. This allows the network to focus its learning capacity on separating the most confusable voltage signals, rather than expending effort pushing already distinct representations further apart. This fosters a richer and more flexible representational space for the membrane potentials, leading to the improved final performance that is empirically validated by our results.

## R    LIMITATIONS AND FUTURE DIRECTIONS

### R.1    LIMITATIONS

Despite its promising performance and significant contributions, the Bidirectional Spike-Based Distillation (BSD) framework also presents opportunities for further research. The current BSD framework has been evaluated on conventional neural network architectures such as Multi-Layer Perceptrons (MLPs), Convolutional Neural Networks (CNNs), Recurrent Neural Networks (RNNs), and autoencoders. Expanding the applicability of BSD to a broader range of network architectures, such as those that incorporate residual connections or attention mechanisms, can further enhance its capacity to address complicated tasks.

### R.2    FUTURE DIRECTIONS

Future research can explore extending the BSD framework to support a wider variety of network architectures, including those that employ residual connections and attention mechanisms, to handle increasingly complex datasets and challenging tasks.

## S    THE USE OF LARGE LANGUAGE MODELS

The authors acknowledge the use of a Large Language Model (LLM) in this work. Its role was strictly confined to polish the writing and correcting grammatical errors to enhance the overall clarity and readability of the paper.

