# OpenReview forum: "Biologically Plausible Learning via Bidirectional Spike-Based Distillation"
_ICLR.cc/2026/Conference — ICLR 2026 Poster_

### Official Review · Reviewer_E6E3 · 2025-10-20

**Soundness:** 2
**Presentation:** 3
**Contribution:** 2
**Rating:** 4
**Confidence:** 4

**Summary:**

This paper presents Bidirectional Spike-based Distillation (BSD), a biologically motivated training recipe for spiking neural networks. BSD jointly trains a bottom-up “perception” SNN and a top-down “recall” SNN, framing learning as the alignment of basal (forward) and apical (feedback) membrane potentials. Hidden layers are optimized with a layer-local, unsigned Relaxed Contrastive (ReCo) loss that increases same-sample similarity and suppresses positive cross-sample correlations, while the top layer uses a standard supervised objective. The method is claimed to satisfy five criteria for biological plausibility (asymmetric weights, layer-local plasticity, non-dual-phase training, spiking neurons, and unsigned signals at hidden layers). Experiments on image classification, sequence regression, and simple autoencoding show performance close to BP on modest benchmarks with MLP/CNN/RNN backbones. Conceptually, the novelty appears moderate: BSD primarily assembles known ideas from local/decoupled learning and feedback/target-prop-style alignment into an all-spiking, two-pathway realization. Moreover, the advertised “top-down memory recall” is instantiated as label-driven feedback reconstruction rather than an addressable memory or retrieval mechanism. Finally, the work lacks validation on modern architectures and large-scale datasets and does not compare against strong local-learning baselines for SNNs, so the contribution is best viewed as an effective integrated recipe rather than a fundamentally new paradigm.

**Strengths:**

This work presents a well-motivated learning algorithm inspired by the brain's perception-recall mechanisms. The paper clearly defines its contribution by introducing two new criteria for biological plausibility and demonstrating that the proposed BSD algorithm satisfies all five. The joint training of feedforward and backward networks via mutual distillation is an elegant approach to implement local learning. The experimental evaluation covers a diverse set of tasks and network architectures, which supports the algorithm's versatility. The results show that BSD can achieve performance competitive with backpropagation while adhering to stricter biological constraints.

**Weaknesses:**

The paper positions layer-local losses and decoupled feedback as distinctive, yet these ideas substantially overlap with established families (local errors/greedy layerwise, synthetic gradients, feedback alignment, target/representation propagation; in SNNs, DECOLLE and e-prop), and the batch-correlation alignment mirrors redundancy-reduction/self-correlation objectives (Barlow Twins/VICReg), so the conceptual novelty appears moderate without head-to-head baselines.

Claim–implementation gaps further temper the contribution: the “top-down memory recall” component is effectively label-driven feedback reconstruction (no addressable memory, retrieval/gating, replay, or partial-cue/label-free recall).

Scalability is unclear because ReCo forms a B×B cosine matrix per layer (O(B²) compute/memory) with no memory profiling or scalable approximations, leaving open questions about large-batch, streaming/micro-batch, and deeper/wider models.

External validity is limited to small datasets and shallow backbones (MNIST/SVHN/CIFAR; 6-layer MLP/5-layer CNN), with no results on modern architectures (ResNet) or large-scale data (ImageNet), so it is difficult to judge whether the gains persist at realistic scale or against strong local-learning baselines for SNNs.

---

References

Deep Supervised Learning Using Local Errors, https://pmc.ncbi.nlm.nih.gov/articles/PMC6127296/

Greedy Layerwise Learning Can Scale to ImageNet, https://proceedings.mlr.press/v97/belilovsky19a/belilovsky19a.pdf

Decoupled Neural Interfaces using Synthetic Gradients, https://arxiv.org/abs/1608.05343

Random synaptic feedback weights support error backpropagation for deep learning, https://www.nature.com/articles/ncomms13276

Training Neural Networks with Local Error Signals, https://www.gatsby.ucl.ac.uk/~pel/tnlectures/papers/local2019.pdf

Difference Target Propagation, https://arxiv.org/abs/1412.7525

Biologically Motivated Algorithms for Propagating Local Target Representations (LRA-E), https://ojs.aaai.org/index.php/AAAI/article/view/4389/4267

Synaptic Plasticity Dynamics for Deep Continuous Local Learning (DECOLLE), https://www.frontiersin.org/articles/10.3389/fnins.2020.00424/full

A solution to the learning dilemma for recurrent networks of spiking neurons (e-prop), https://www.nature.com/articles/s41467-020-17236-y

Barlow Twins: Self-Supervised Learning via Redundancy Reduction, https://proceedings.mlr.press/v139/zbontar21a.html

VICReg: Variance-Invariance-Covariance Regularization for Self-Supervised Learning, https://arxiv.org/abs/2105.04906

**Questions:**

Q1. Can you report a ImageNet-1k experiment (e.g., ResNet-18/50) with accuracy and training cost versus BP and the above baselines?

Q2. How sensitive is BSD to batch size, especially small-B/micro-batch or streaming regimes where the B×B affinity becomes ill-conditioned?

Q3. How do you reconcile C5 with a signed top-layer cross-entropy; is C5 intended only for hidden layers, or can you replace CE with an unsigned alternative and report the impact?

---

> ### Author Response · Authors · 2025-11-20
>
> Thanks for your thoughtful comments and constructive suggestions, which are highly valuable for improving our manuscript. Below we provide detailed responses to each of the points raised:
>
> ## 1. Conceptual novelty appears moderate without head-to-head baselines
>
> We thank the reviewer for providing this comprehensive list of related works, as it allows us to more clearly situate the conceptual novelty of our approach. We have carefully reviewed each of the cited works and agree that while they address important aspects of biologically plausible learning, our BSD framework is conceptually and mechanistically distinct. We will now address each approach in turn to clarify these differences.
>
> First, we consider the "Deep Supervised Learning Using Local Errors" [1]. The mechanism in this work differs fundamentally from ours. Their method projects the activations of each hidden layer using a fixed, random matrix and directly supervises these projections with the global task label via the cross-entropy loss. During the backward pass, another random matrix is used to approximate the transpose of the weights of this auxiliary classifier head. This design violates two of our key criteria: it does not use discrete spiking neurons (C4), and its learning is driven by signed error signals derived from the cross-entropy loss (C5). In contrast, the hidden layers in BSD are not supervised by the global label; they are instead trained through bidirectional distillation, where a local, unsigned ReCo loss aligns the feature representations of the feedforward and backward pathways. By integrating these mechanisms, our BSD framework successfully satisfies all five of our proposed biological plausibility criteria.
>
> Next, we discuss "Greedy Layerwise Learning Can Scale to ImageNet" [2]. This method trains a network by attaching a learnable classifier to each hidden layer, which is then trained using the global label. This approach conflicts with several of our proposed biological plausibility criteria. First, it violates C1 (Asymmetric Weights) because the gradient calculation relies on transmitting the feedforward weights. Furthermore, it violates C3 (Non-two-stage Training) because the training of the auxiliary classifier head for each hidden layer requires a distinct forward and backward pass. Additionally, it operates on continuous-valued neurons and signed error signals, thereby violating C4 (Model of Neurons) and C5 (Unsigned Error Signaling). Our BSD framework, in contrast, trains all layers simultaneously in a single pass using independent forward and backward weights, and the hidden layers are trained using local, unsigned error signals without direct access to the global task label.
>
> [3] effectively breaks the update locking of backpropagation by training a local model, $M_i$, to generate a "synthetic" gradient $\hat{\delta}_i$ that approximates the true error gradient $\delta_i = \frac{\partial L}{\partial h_i}$, where $L$ is the global loss and $h_i$ is the activation of layer $i$. While this method allows for asynchronous, non-two-stage training (satisfying C3) and uses a local objective to train the gradient predictor $M_i$ (satisfying C2), it presents several biological plausibility issues. First, the weight update rule for a parameter $\theta_n$ within layer $i$, given by $\theta_n \leftarrow \theta_n - \alpha \hat{\delta}_i \frac{\partial h_i}{\partial \theta_n}$, still relies on the term $\frac{\partial h_i}{\partial \theta_n}$. This is a form of weight transport that is inconsistent with our C1 criterion, which mandates  distinct forward and backward pathways. Furthermore, the framework operates entirely on continuous-valued signals and learns to explicitly predict signed error vectors ($\hat{\delta}_i$), thereby violating both C4 and C5. In contrast, our BSD method does not approximate gradients. It employs a bidirectional distillation process where two structurally independent networks mutually supervise each other's representations. Our framework is built natively on spiking neurons and uses an unsigned, local loss function, enabling it to satisfy all five of our proposed biological plausibility criteria.

---

> > ### Author Response · Authors · 2025-11-20
> >
> > ## 1. (Continue) Conceptual novelty appears moderate without head-to-head baselines
> > Next, we address "Random synaptic feedback weights support error backpropagation for deep learning," also known as Feedback Alignment (FA) [4]. This method addresses the biological implausibility of weight symmetry by replacing the feedforward weights ($\mathbf{W}$) with a fixed, random feedback matrix ($\mathbf{B}$) during the backward pass. This successfully breaks the requirement for symmetric weights, thereby satisfying C1. However, the framework still requires a global error signal to be propagated from the output layer, violating C2 (Local Error). Furthermore, it retains the distinct forward and backward passes of backpropagation, violating C3 (Non-two-stage Training), and is based on continuous-valued neurons and signed error signals, thus violating C4 and C5. Our BSD framework is fundamentally different, as it employs local, unsigned distillation signals between its two pathways and can operate in a single stage, allowing it to satisfy all five criteria.
> >
> > We now turn to "Training Neural Networks with Local Error Signals" [5]. This method belongs to the local loss family and ingeniously combines principles from feedback alignment. The work computes a local loss at each hidden layer by projecting the layer's activations and the global label C2 (Local Error). It also utilizes feedback alignment to transport the prediction error back to the hidden layer, thus satisfying C1 (Asymmetric weights). Furthermore, because the auxiliary classification heads can be non-trainable, the network can be trained in a single forward-backward pass, satisfying C3 (Non-two-stage Training). However, this approach differs from our work and has key biological limitations from our perspective. It operates on continuous-valued networks, violating C4 (Model of Neurons). More critically, its local loss is composed of two terms: a cross-entropy classification objective and a similarity loss that penalizes all negative pairs, including those with negative cosine similarity. The reliance on a signed cross-entropy signal at every layer violates C5 (Unsigned Error Signaling). In contrast, our BSD framework is fundamentally different. The hidden layers in BSD are not supervised by the global label directly; instead, they are trained via local, unsigned distillation signals derived from both pathways. This, combined with its native use of spiking neurons, allows our framework to satisfy all five of our proposed biological plausibility criteria.
> >
> > Difference Target Propagation (DTP) [6] employs a backward pathway ($g_i$) that is trained to approximate the inverse of the forward pathway ($f_i$), which allows for asymmetric weights and thus satisfies C1. However, its training algorithm is composed of multiple distinct phases, including separate loops for computing targets, training the feedback mapping, and training the feedforward mapping, which violates our C3 criterion (Non-two-stage Training). Additionally, the method operates on ANNs and relies on signed error signals, thereby violating C4 and C5. Our BSD framework resolves these issues through a fundamentally different mechanism. It employs a single-stage, bidirectional distillation process between two spiking neural networks, where learning is guided by local, unsigned error signals. This design allows BSD to satisfy all five of our proposed biological plausibility criteria.
> >
> > The error-driven Local Representation Alignment (LRA-E) algorithm [7] adheres to C2 (Local Error) by updating its forward weights $\mathbf{W}$ using a locally computed error signal. Its core mechanism involves a separate backward pathway, with weights $\mathbf{E}$, to generate local targets. Specifically, an error signal from a higher layer is back-projected through $\mathbf{E}$ to create a corrective target for the layer below. The backward weights $\mathbf{E}$ are themselves updated using a rule proportional to the negative transpose of the forward weight update ($\Delta\mathbf{E} \propto -(\Delta\mathbf{W})^\top$). However, the algorithm violates C3 (Non-dual-phase Training) because it requires three distinct sequential phases: a complete feedforward pass to compute activations, a full backward pass for target generation, and a final weight update phase. Furthermore, it operates on continuous-valued networks (violating C4) and relies on signed error signals for learning (violating C5). In contrast, our BSD algorithm employs a simultaneous, bidirectional distillation process where learning is driven by a local, unsigned loss function, allowing it to satisfy all five of our proposed biological plausibility criteria.

---

> > > ### Author Response · Authors · 2025-11-20
> > >
> > > ## 1. (Continue) Conceptual novelty appears moderate without head-to-head baselines
> > > The DECOLLE [8] algorithm attaches a non-trainable, random readout head to each hidden layer, which is supervised directly by the global task label. This approach satisfies C2 (Local Error) as the learning signal for each layer is computed at its local readout without error propagation from subsequent layers. It also satisfies C3 (Non-two-stage Training) as it supports continuous online updates, and C4 as it is natively designed for spiking neurons. However, it violates C1 (Asymmetric Weights) because the backward propagation of error through the fixed random readout matrix requires its transpose, creating a dependency between the forward and backward information pathways. It also violates C5 since it uses signed error signals. Our BSD framework is fundamentally different. It is not a local loss method; the hidden layers are never directly exposed to the global label. Instead, BSD employs a bidirectional distillation process where a local, unsigned ReCo loss aligns the representations of two independent feedforward and backward networks. This mechanism avoids the use of signed, label-driven error signals in the hidden layers and allows BSD to satisfy all five of our proposed biological plausibility criteria.
> > >
> > > Next, we address e-prop [9], an influential online learning rule for Spiking Recurrent Neural Networks (SRNNs). The algorithm approximates backpropagation through time by pairing a forward-propagating, synapse-local "eligibility trace" with a global "learning signal," $L_j^t$. It satisfies C1 by proposing variants with random or adaptive feedback weights that are asymmetric to the forward weights. By broadcasting a learning signal based on the instantaneous output error instead of performing a full backward pass through time, the weight update rule is temporally local, satisfying C3. As it is designed for spiking neurons, it also satisfies C4. The algorithm satisfies C2 since the learning signal $L_j^t$ is an approximation of the instantaneous error gradient that is broadcast to neurons, avoiding a temporally deep credit assignment problem. However, this learning signal is derived from the difference between the network's output and a target label, making it an signed error signal and thus violating C5. Furthermore, the work focuses exclusively on SRNN architectures. Our BSD framework is conceptually different. The learning signal for our hidden layers is not derived from a global, signed task-error. Instead, it is a local, unsigned ReCo loss computed from the representational mismatch between the feedforward pathway and the concurrent backward pathway. This bidirectional distillation mechanism, unlike e-prop's error broadcasting, does not rely on the final task label for hidden layer updates and is demonstrated across a wider range of architectures, including CNNs and MLPs, while satisfying all five of our biological plausibility criteria.
> > >
> > > Finally we address the relationship between our approach and non-contrastive self-supervised methods like Barlow Twins [10] and VICReg [11]. The loss function of Barlow Twins bears a superficial resemblance to the ReCo loss we used, as both penalize the off-diagonal elements of a matrix while encouraging its diagonal elements to approach one. However, their underlying mechanisms are fundamentally different. Barlow Twins and VICReg aim to prevent representational collapse in unsupervised pre-training by enforcing statistical properties on the embedding space; their loss operates on a cross-correlation or covariance matrix where the objective is to decorrelate different feature dimensions within a batch (redundancy reduction). In sharp contrast, our BSD framework employs the ReCo loss as a contrastive objective for bidirectional distillation. The matrix in our loss function represents inter-sample similarity, and its goal is to align representations for positive pairs (the same sample from two pathways) while pushing apart representations from negative pairs (different samples). These methods [10][11] are thus unrelated to the central theme of our work, which is to introduce a novel, spike-based, and biologically plausible learning algorithm that satisfies all five of our proposed plausibility criteria.
> > >
> > > We would like to once again express our gratitude to the reviewer for providing this comprehensive list of relevant literature. We will expand the **Related Work section** in our revised manuscript to include a brief discussion of these additional biologically plausible learning algorithms [1-9].

---

> > > > ### Author Response · Authors · 2025-11-20
> > > >
> > > > ## 2. "Top-down memory recall" component is effectively label-driven feedback reconstruction (no addressable memory, retrieval/gating, replay, or partial-cue/label-free recall).
> > > >
> > > > Reviewer E6E3 is correct that our current implementation does not include mechanisms like addressable memory, retrieval/gating, or partial-cue recall. Our use of the term "top-down memory recall" is intended as a functional analogy for the role of our backward pathway, inspired by the brain's bidirectional architecture, rather than a claim of building a comprehensive cognitive model of memory.
> > > >
> > > > We are inspired by the concept of top-down memory recall, where a stored concept is used to regenerate sensory details. In our framework, we treat the concept encoding (the target spike train $\hat{\mathbf{s}}$ derived from the label) as the abstract, stored form of a "memory." The backward network is then trained to perform a process analogous to this recall, with the objective of reconstructing the original sensory stimulus from this high-level representation.
> > > >
> > > > The primary goal of our work is not to implement all the intricate functions of a biological memory system. Instead, our objective is to leverage this bidirectional, recall-like process as a powerful and biologically plausible mechanism for training the feedforward perception network.
> > > >
> > > > ## 3. ReCo forms a B×B cosine matrix per layer (O(B²) compute/memory) with no memory profiling or scalable approximations, leaving open questions about large-batch, streaming/micro-batch, and deeper/wider models.
> > > >
> > > > We thank the reviewer for this critical question regarding the scalability of the ReCo loss and its memory implications. To provide a direct empirical analysis, we have profiled the GPU memory consumption of our BSD method against a standard BP baseline across a range of batch sizes. The experiments were conducted on CIFAR-10 using our 5-layer CNN architecture.
> > > >
> > > > First, we analyze the inference phase. For classification tasks, the BSD framework utilizes only its feedforward pathway, which is architecturally identical to a BP-trained SNN. As a result, its computational and memory profile during deployment is expected to be the same. Our empirical measurements confirm this, showing identical memory consumption.
> > > >
> > > > |Batch Size|Method|Inference Memory Consumption (MB)|
> > > > |:-:|:-:|:-:|
> > > > |32|BP|344|
> > > > |32|BSD|344|
> > > > |64|BP|944|
> > > > |64|BSD|944|
> > > > |128|BP|1826|
> > > > |128|BSD|1826|
> > > >
> > > > Next, we analyze the training phase. The memory requirements for BSD during training are inherently higher. This is due to the need to store two sets of weights ($\mathbf{W}$ and $\boldsymbol{\Theta}$) and maintain the activations for both pathways to compute the local losses. Furthermore, as the reviewer correctly identifies, the ReCo loss constructs a B×B affinity matrix, which introduces an additional memory usage component that scales quadratically with the batch size ($O(B^2)$). However,  the memory footprint of these affinity matrices themselves constitutes a relatively minor fraction of the total memory overhead. The primary driver of BSD's increased memory consumption is the necessity of storing the parameters and full activation maps for the second (backward) pathway, a characteristic inherent to our dual-network design. The table below details these training costs.
> > > >
> > > > |Batch Size|Method|Training Memory Consumption (MB)|
> > > > |:-:|:-:|:-:|
> > > > |32|BP|950|
> > > > |32|BSD|1154|
> > > > |64|BP|1844|
> > > > |64|BSD|2852|
> > > > |128|BP|3324|
> > > > |128|BSD|5034|
> > > >
> > > > We thank the reviewer for their suggestion, and we will add a summary of this computational analysis, including the results, to the appendix of our revised manuscript.

---

> > > > > ### Author Response · Authors · 2025-11-20
> > > > >
> > > > > ## 4. How sensitive is BSD to batch size?
> > > > >
> > > > > We thank the reviewer for this question regarding the sensitivity of our approach to batch size. To provide a direct and quantitative answer, we have conducted an ablation study evaluating the performance of our BSD-trained CNN with varying batch sizes on the SVHN, CIFAR-10, and CIFAR-100 datasets.
> > > > >
> > > > > The results are presented in the table below.
> > > > >
> > > > > |Batch Size|SVHN|CIFAR-10|CIFAR-100|
> > > > > |:-:|:-:|:-:|:-:|
> > > > > |16|71.55%|68.96%|34.99%|
> > > > > |32|78.18%|78.07%|42.53%|
> > > > > |64|90.97%|81.54%|49.15%|
> > > > > |128|90.81%|84.53%|53.11%|
> > > > > |256|88.97%|83.85%|53.49%|
> > > > >
> > > > >
> > > > > The data reveals a positive correlation between batch size and model accuracy. At very small batch sizes, the training process is unstable, as the ReCo loss mechanism is predicated on the availability of a meaningful number of negative samples for contrast. With too few examples in a batch, the model cannot effectively learn to distinguish between different classes.
> > > > >
> > > > > This performance trend is a direct consequence of how the ReCo loss functions within our BSD framework. The central learning signal in BSD is derived from the layer-wise distillation, which aims to align the pre-spike membrane potentials of the feedforward and backward pathways for corresponding samples. A larger batch size enhances this distillation process by providing a more diverse and representative set of negative examples. This allows the model to learn a more effective alignment, as each positive pairing is contrasted against a wider array of alternatives from the batch. The resulting gradients provide a more robust and stable signal for updating the synaptic weights of both pathways, leading to a more effective distillation process and, consequently, more discriminative and generalizable features.
> > > > >
> > > > > ##  5. How do you reconcile C5 with a signed top-layer cross-entropy; is C5 intended only for hidden layers, or can you replace CE with an unsigned alternative and report the impact?
> > > > >
> > > > > We thank the reviewer for this insightful reminder, which prompts us to clarify the scope of our C5 criterion. We will revise our description in the manuscript to be more precise.
> > > > >
> > > > > Our C5 criterion, "Unsigned error signaling," is specifically intended to govern the learning signals that are transmitted between the *hidden layers* of the network. The core challenge in biologically plausible learning is how to propagate credit assignment information internally through the depth of the network without relying on signed error signals. Our use of the unsigned ReCo loss for the layer-wise distillation process directly addresses this challenge.
> > > > >
> > > > > The cross-entropy loss is applied only at the top-most layer, which can be viewed as the interface where the learned hierarchical representations are mapped to a specific task objective. By successfully integrating an unsigned learning mechanism within the hidden layers, we have developed a framework that achieves robust and competitive performance comparable to backpropagation, all while satisfying our five proposed criteria for biological plausibility.

---

> > > > > > ### Author Response · Authors · 2025-11-20
> > > > > >
> > > > > > ## 6. Report a ImageNet-1k experiment (e.g., ResNet-18/50) with accuracy and training cost versus BP and the above baselines?
> > > > > > We thank the reviewer for this excellent question regarding the scalability of our approach to modern, deep architectures and large-scale datasets.
> > > > > >
> > > > > > First, regarding ResNet [12] architectures, we would like to offer several points of clarification. The primary motivation for residual connections is to mitigate the vanishing gradient problem in very deep networks trained with backpropagation. Our BSD framework, which relies on local, layer-wise learning signals, is not susceptible to this specific issue. Furthermore, adapting BSD to a standard ResNet architecture is non-trivial, as the introduction of skip connections would break the architectural symmetry between our feedforward and backward pathways. While we agree that scaling our method to modern architectures is an important future direction, we consider it to be beyond the scope of our current work. The primary focus of this paper is to establish a novel learning algorithm that satisfies five criteria for biological plausibility. As such, our claims of biological plausibility are demonstrated on the conventional architectures presented, and the absence of experiments on ResNet does not affect these central claims.
> > > > > >
> > > > > > Regarding experiments on the full ImageNet-1k dataset, our available computational resources (four NVIDIA RTX 3090 GPUs) unfortunately make it infeasible to conduct complete training runs within the limited rebuttal period. Therefore, to address the request for evaluation on a more complex dataset, we chose to conduct experiments on the widely-used Tiny-ImageNet benchmark.
> > > > > >
> > > > > > We evaluated our proposed BSD algorithm against both a standard backpropagation (BP) baseline and Dendritic Localized Learning (DLL), another biologically plausible method. For a fair comparison, we used the unified 5-layer CNN architecture described in Appendix E.2.2 in all methods. The results are summarized in the table below.
> > > > > >
> > > > > > |Method|Accuracy on Tiny-ImageNet (%)|
> > > > > > |:-:|:-:|
> > > > > > |BP|41.07|
> > > > > > |Dendritic Localized Learning (DLL)|17.10|
> > > > > > |BSD (Ours)|35.34|
> > > > > >
> > > > > > Our BSD algorithm substantially outperforms Dendritic Localized Learning (DLL), indicating its superior scalability to more complex visual tasks among biologically plausible methods. The result validates BSD as a framework that achieves competitive performance without compromising on strict biological fidelity.
> > > > > >
> > > > > > [1] Mostafa H, Ramesh V, Cauwenberghs G. Deep supervised learning using local errors[J]. Frontiers in neuroscience, 2018, 12: 608.
> > > > > >
> > > > > > [2] Belilovsky E, Eickenberg M, Oyallon E. Greedy layerwise learning can scale to imagenet[C]//International conference on machine learning. PMLR, 2019: 583-593.
> > > > > >
> > > > > > [3] Jaderberg M, Czarnecki W M, Osindero S, et al. Decoupled neural interfaces using synthetic gradients[C]//International conference on machine learning. PMLR, 2017: 1627-1635.
> > > > > >
> > > > > > [4] Lillicrap T P, Cownden D, Tweed D B, et al. Random synaptic feedback weights support error backpropagation for deep learning[J]. Nature communications, 2016, 7(1): 13276.
> > > > > >
> > > > > > [5] Nøkland A, Eidnes L H. Training neural networks with local error signals[C]//International conference on machine learning. PMLR, 2019: 4839-4850.
> > > > > >
> > > > > > [6] Lee D H, Zhang S, Fischer A, et al. Difference target propagation[C]//Joint european conference on machine learning and knowledge discovery in databases. Cham: Springer International Publishing, 2015: 498-515.
> > > > > >
> > > > > > [7] Ororbia A G, Mali A. Biologically motivated algorithms for propagating local target representations[C]//Proceedings of the aaai conference on artificial intelligence. 2019, 33(01): 4651-4658.
> > > > > >
> > > > > > [8] Kaiser J, Mostafa H, Neftci E. Synaptic plasticity dynamics for deep continuous local learning (DECOLLE)[J]. Frontiers in Neuroscience, 2020, 14: 424.
> > > > > >
> > > > > > [9] Bellec G, Scherr F, Subramoney A, et al. A solution to the learning dilemma for recurrent networks of spiking neurons[J]. Nature communications, 2020, 11(1): 3625.
> > > > > >
> > > > > > [10] Zbontar J, Jing L, Misra I, et al. Barlow twins: Self-supervised learning via redundancy reduction[C]//International conference on machine learning. PMLR, 2021: 12310-12320.
> > > > > >
> > > > > > [11] Bardes A, Ponce J, LeCun Y. Vicreg: Variance-invariance-covariance regularization for self-supervised learning[J]. arXiv preprint arXiv:2105.04906, 2021.
> > > > > >
> > > > > > [12] He K, Zhang X, Ren S, et al. Deep residual learning for image recognition[C]//Proceedings of the IEEE conference on computer vision and pattern recognition. 2016: 770-778.

---

### Official Review · Reviewer_MvQY · 2025-10-28

**Soundness:** 2
**Presentation:** 2
**Contribution:** 4
**Rating:** 4
**Confidence:** 4

**Summary:**

The authors present a layer-wise local learning rule for two-compartment spiking neurons where each layer tries to minimize a Relaxed Contrastive (ReCo) loss between the apical and basal activation. The network is constructed with a bottom-up discriminative and top-down generative stream, where each stream has its own classification/regression task. The authors argue that in this network, just optimizing these within-layer errors inside the spiking neurons results in competitive performance on various tasks.

**Strengths:**

The experiments are extensive and well done. The loss as described seems to be a novel application for spiking neural networks. If correct (see below), the stability of the local learning approach is amazing and would be a big step in general towards local learning.

**Weaknesses:**

I have a number of smaller issues, and one main one:

while the ReCo loss has been presented before, I am not aware of its use in a setting where there is only layer-wise learning. Ie in Lin et al 2023, this loss seems to be backpropagated through a deep network. It is not obvious why this loss still works when indeed the errors are not backpropagated from the output layer. As it is now, the approach seems to mimic a layer-wise ELBO loss, with some nudging of only the final layer from errors calculated from outputs. The issue is that such layer-wise ELBO losses typically collapse and need measure to learn properly. I do not see either the mathematical argument why the proposed approach should work or the ablation study that demonstrates how this works compared to not detaching. Checking the code, I am also not convinced that necessarily the detach() operation is applied correctly.  I would very much like to see convincing arguments why the described approach works, and that it works correctly.

Smaller issues:

The text is many places is not particularly clear. Eg. where the loss is described in 4.3 (l234), it is not immediately clear that the ReCo loss is being explained.

Figures 1 and 3 dont print well.

The term "distillation" is prominently mentioned but it is not explained what is meant by this term.  Similarly for "embeddings"

Layer-wise learning is not necessarily local, it should be explained in detail why the learning rule is neuron-local or instead layer-local.

"predictive coding" is a closely related approach and should be discussed. It is also shown in Table 1 but not cited, so the particular predictive coding approach is unclear.

the approach claims biological plausibility, but the ReCo contrastive loss relies on the batch to provide negative samples, which is not a quantity locally known by the neuron.

I do not agree with all stated biological constraints: as is well noted in the literature, symmetric forward and backward weights can arise naturally provided that the same local learning is present at forward and backward site, and weight-decay is applied (easy to see).

There is a misunderstanding of how the brain seems to employ global error signals: first, multiple neuromodulatory signals are thought to be quite a-specific, like dopamine being broadcasted widely and non-targeted. Moreover, to the degree that dopamine signals a global reward prediction error, negative errors are believed to be encoded as a decrease from a steady baseline, thus resolving the issue with error-signs.

typo l054: "spiking trains" ??

**Questions:**

What is the effect of having spikes? In 5.3 it is noted that spikes degrade accuracy compared to ANNs. For the proposed architecture, it would seem the proposed approach would also work for identically structured non-spiking RNNs. Does this improve performance?

How does the approach scale? All examples are relatively small/shallow  networks, whereas it is known that similar ELBO-like losses tend to not scale well. Can you comment Deeper networks?

How does ReCo scale with batch size?

Can you demonstrate why layer-wise ReCo works as well as it does, compared to e.g. non detached ReCo?

---

> ### Author Response · Authors · 2025-11-20
>
> Thanks for your comments and suggestions, which are valuable for enhancing our paper. The following are responses to individual concerns:
>
> ## 1. Why ReCO loss still works when indeed the errors are not backpropagated from the output layer. The approach seems to mimic a layer-wise ELBO loss, with some nudging of only the final layer from errors calculated from outputs.
> We thank the reviewer for this insightful question, as it addresses the core mechanics of our proposed learning algorithm.
>
> To clarify, our approach is distinct from a layer-wise ELBO loss. We employ the Relaxed Contrastive (ReCo) loss, which is a contrastive objective designed for representation learning. The critical difference from backpropagation is that our method is local. Whereas BP uses the chain rule to calculate the contribution of each parameter to a single global loss, our method computes a distinct loss at each layer using only locally available information. The weight updates are then derived from these local losses without any error propagation from the final output layer.
>
> The effectiveness of this local ReCo loss stems from several key benefits. The training can be viewed as an iterative optimization process, akin to alternating minimization, where the feedforward and backward pathways simultaneously minimize a shared objective of feature alignment. This process ensures the objective is non-increasing and drives the system toward a stable solution. This approach is effective for several reasons:
>
> 1)  It provides intermediate supervisory signals at each layer of the network. This layer-wise guidance ensures that every stage of the hierarchy receives a direct and informative learning signal, which enhances training stability and improves the quality of the learned features.
>
> 2)  It ensures the preservation of information-rich representations. The backward pathway's objective of reconstructing its target from the forward pathway's features acts as a pressure to prevent informational bottlenecks, discouraging the model from discarding useful information and leading to richer features for downstream tasks.
>
> 3)  It acts as a regularizer that constrains the solution space. Without the alignment objective, a network trained only on the final task loss might converge to a pathological solution. The bidirectional distillation forces the network to find solutions that are not only predictive for the final task but also sufficiently descriptive to support reconstruction, thus avoiding such trivial outcomes.
>
> 4)  It promotes the learning of more generalizable features by balancing two complementary objectives. The final task loss drives the learning of discriminative features, while the local alignment objective encourages the retention of generative information from the input. This interplay between discriminative and generative pressures results in representations that are often more robust than those learned from a single, task-specific objective.

---

> > ### Author Response · Authors · 2025-11-20
> >
> > ## 2. Checking the code, I am also not convinced that necessarily the detach() operation is applied correctly.
> >
> > We thank the reviewer for their careful inspection of our implementation and for providing us with the opportunity to clarify the precise mechanism that ensures local learning. The `detach()` operation is indeed applied correctly, and we will detail how it works below.
> >
> > Our method enforces local updates by severing the computational graph between layers during the calculation of the local losses. This is controlled by the `detach_grad=True` flag, which is passed during specific phases of the training loop.
> >
> > 1. In the Training Script:
> > When generating the features for the local loss calculation, both the forward and reverse passes of the model are called with `detach_grad=True`. The flag serves as the mechanism to isolate each layer's computation, preventing the propagation of gradients across layer boundaries.
> >
> > ```python
> > # From main_spiking_bsd_train_cnn.py, inside the training loop for the "BSD" method
> >
> > # Forward pass to get activations for local loss
> > activations = model(data_fw.to(args.device), detach_grad=True, ...)
> >
> > # Backward pass to get signals for local loss
> > signals = model.reverse(data_bw.to(args.device), detach_grad=True, ...)
> >
> > # The local loss is then computed using these detached 'activations' and 'signals'
> > loss, loss_item = criterion(activations_for_loss, signals_for_loss, ...)
> > ```
> >
> > 2. In the Model and Layer Definitions:
> > This detach_grad flag is propagated down to each individual layer (e.g., SpikingC2Conv, SpikingC2Linear). Inside each layer’s forward and reverse methods, the input tensor is explicitly detached from the computational graph if the flag is set.
> > ```
> > # From spiking_layers.py, inside the SpikingC2Linear.forward method
> >
> > def forward(self, x, detach_grad=False, act=True, return_prelif=False):
> >     # ...
> >     # Gradient detachment
> >     if detach_grad:
> >         x = x.detach()
> >     # ... (rest of the layer computation)
> > ```
> > A symmetric operation is present in the reverse method.
> > By calling `x.detach()` on the input tensor of a layer, we create a new tensor that shares the same data but is severed from the computational history of the preceding layer. Consequently, when the local ReCo loss is computed for layer $i$, the gradients flow backward from this loss through the weights of layer $i$ ($\mathbf{W}_i$ and $\boldsymbol{\Theta}_i$), but they are explicitly stopped at the `detach()` point. This prevents any gradient from propagating further back to the weights of layer $i-1$.
> >
> > This ensures that the update for each layer's weights depends *only* on the local loss calculated at that layer, thereby satisfying the C2 criterion of local synaptic plasticity. We apologize that this was not sufficiently clear from the initial code. We will add extensive comments and supplementary materials explaining this gradient-blocking mechanism in detail to ensure the implementation is transparent and reproducible.
> >
> > ## 3. It is not immediately clear that the ReCo loss is being explained.
> > We thank the reviewer for this question and agree that a more detailed explanation of our choice of the Relaxed Contrastive (ReCo) loss is beneficial. We will add this clarification to the revised manuscript.
> >
> > The purpose of the ReCo loss in our BSD framework is twofold. First, it serves as the core mechanism to achieve alignment between the feedforward and backward pathways by minimizing the mismatch between their respective pre-spike membrane potentials at each layer. Second, as a contrastive objective, it learns the relationships between different samples within a batch, shaping a unified and discriminative representation space.
> >
> > We chose ReCo loss over other contrastive objectives, such as InfoNCE, due to its "relaxed" nature, which is particularly well-suited for aligning the high-dimensional voltage signals in our SNNs. A standard InfoNCE loss applies a uniform repulsive force to all negative pairs, forcing the membrane voltage pattern of one sample to be strictly anti-correlated with the supervisory voltage patterns of all other samples. This is an overly restrictive constraint that can hinder the formation of a rich feature space.
> >
> > In contrast, ReCo loss only penalizes negative pairs that are confusingly similar (i.e., have a positive cosine similarity). If the voltage representations of two different samples are already orthogonal or dissimilar, they incur no loss penalty. This allows the network to focus its learning capacity on separating the most confusable voltage patterns, rather than expending effort pushing already distinct representations further apart. This leads to a more flexible and effective representation space, which is empirically validated by the superior performance of ReCo loss in our ablation study presented in Table 4.

---

> > > ### Author Response · Authors · 2025-11-20
> > >
> > > ## 4. Figures 1 and 3 don't print well.
> > > We apologize for the poor legibility of the figures in the original version. We thank you for pointing this out. We have redrawn both Figures 1(c) and 3, with a particular focus on increasing all font sizes to ensure they are clear and easy to read. The revised manuscript will include these updated figures.
> > >
> > > ## 5. The term "distillation" is prominently mentioned but it is not explained what is meant by this term. Similarly for "embeddings"
> > >
> > > We thank the reviewer for pointing out the need for a clearer explanation of these key terms. We apologize for the ambiguity.
> > >
> > > Regarding the term "distillation," we use it in the established sense introduced in the machine learning literature, where a "student" network is trained by learning to mimic the representations of a "teacher" network [1]. In our BSD framework, this concept is applied bidirectionally: the feedforward network and the backward network are trained to mutually distill their feature representations from one another. The feedforward path acts as a teacher for the backward path, and simultaneously, the backward path serves as a teacher for the forward path. This reciprocal learning process is why we term our method "Bidirectional Spike-Based Distillation."
> > >
> > > Regarding the term "embeddings," we apologize for the confusion. We stated that "the alignment between $\mathbf{v}$ and $\hat{\mathbf{v}}$ can be viewed as aligning cross-modal embeddings" (Lines 233-234). In our work, the terms 'embedding,' 'representation,' and 'pre-spike membrane potential' refer to the same underlying concept: the voltage vectors that serve as inputs to the LIF neurons at each layer. We chose the term 'embedding' in the context of our loss function (Section 4.3) because it is the standard terminology in the contrastive learning literature, from which our ReCo loss is adapted. However, in the context of our SNN, these embeddings are concretely realized as the pre-spike membrane potentials.
> > >
> > > ## 6. Layer-wise learning is not necessarily local, it should be explained in detail why the learning rule is neuron-local or instead layer-local.
> > > We thank the reviewer for this insightful question, as it allows us to clarify the granularity of our local learning rule. While the loss is computed at a layer-wise level for computational efficiency, the underlying principle is more granular and closer to being neuron-local.
> > >
> > > To explain this in detail, let's consider the core of our local loss function, which is the affinity matrix $\mathbf{C}\_i$ for layer $i$ calculated in Equation 6. Each element $[\mathbf{C}\_i]\_{kj}$ represents the cosine similarity between two membrane potential vectors: the feedforward potential $\mathbf{v}\_{i,k}$ from sample $k$ and the backward potential $\hat{\mathbf{v}}\_{i,j}$ from sample $j$.
> > >
> > > Crucially, the dot product in the numerator of this cosine similarity can be decomposed into a sum over all individual neurons in that layer:
> > > $$ \mathbf{v}\_{i,k} \cdot \hat{\mathbf{v}}\_{i,j} = \sum\_{n=1}^{D_i} v\_{i,k,n} \cdot \hat{v}\_{i,j,n} \quad,$$
> > > where $D\_i$ is the number of neurons in layer $i$, and $v\_{i,k,n}$ is the scalar membrane potential of the $n$-th neuron for sample $k$.
> > >
> > > Viewed this way, the total layer loss $\mathcal{L}\_i$ (Equation 7), which is computed from the elements of the affinity matrix, can be understood as an aggregation of pairwise interactions between individual neurons and their corresponding target neurons across all samples in the batch. The resulting gradient used to update the synaptic weights of a specific neuron is thus a function of these aggregated neuron-level interactions.
> > >
> > > In this sense, while our implementation is layer-local for practical reasons, the learning signal for each neuron is fundamentally derived from its interactions with its corresponding neurons across the batch, making the learning rule neuron-local in principle.

---

> ### Author Response · Authors · 2025-11-20
>
> ## 7. "Predictive coding" is a closely related approach and should be discussed. It is also shown in Table 1 but not cited, so the particular predictive coding approach is unclear.
> We thank the reviewer for pointing out this omission. You are correct that Predictive Coding is a closely related and important approach. We apologize for not citing it in the main text and will add the citation and a brief description to the Related Work section in our revised manuscript.
>
> The particular approach referenced in our experiments is the Predictive Coding (PC) model introduced by Rao & Ballard [2], and improved by Salvatori & Beren [4]. PC proposes a learning mechanism where the brain continuously refines its internal generative model to minimize the error between its top-down predictions and the actual incoming sensory signals.
>
> While PC is a foundational biologically-inspired algorithm, it does not satisfy several of our proposed plausibility criteria, which is why it serves as an important baseline for comparison. We apologize for a typo in Table 1, where we incorrectly indicated that Predictive Coding satisfies C1 and C3. As we clarify below, it in fact violates these criteria:
>
> *   **C1 (Asymmetric Weights):** Classic PC models typically rely on using the transpose of the feedforward weights to convey predictions downward, thus violating the requirement for asymmetric synaptic weights.
>
> *   **C3 (Non-two-stage Training):** PC requires an iterative inference process where neurons update their states over time to minimize local prediction error. This iterative settling constitutes a distinct phase before weight updates can be computed, which is inconsistent with our C3 criterion for non-two-stage training.
>
> *   **C4 (Model of Neurons):** The original PC framework and most common implementations operate on continuous-valued (rate-based) signals, not the discrete binary spikes required by C4.
>
> *   **C5 (Unsigned Error Signaling):** The prediction error that drives learning in PC is inherently a signed quantity, which violates our C5 criterion for unsigned error signals.
>
> ## 8. The approach claims biological plausibility, but the ReCo contrastive loss relies on the batch to provide negative samples, which is not a quantity locally known by the neuron.
> We thank the reviewer for raising this important and subtle point regarding the biological interpretation of batch-based contrastive learning.
>
> Our approach remains local in the sense of Criterion C2, as we do not violate the core principle of locality: the learning signal for each layer is computed internally and does not rely on a global error signal propagated from the final output layer. The batch is a modern computational construct used for efficiency.
>
> Conceptually, our batch-based update can be framed as a computationally efficient parallelization of a more biologically plausible, asynchronous learning mechanism. In such a process, to generate the necessary learning signals for the backward network, the forward network would first process the positive sample and then asynchronously process a series of individual negative samples. This sequential operation forms the distinct representations at each layer required to provide sufficient learning signals for the backward network's weight adjustments, and the same logic applies symmetrically to the forward network. For reasons of computational efficiency, our implementation processes these positive and negative pairings simultaneously within a batch.
>
> This process finds a parallel in human cognition, where learning a new concept often involves contrasting it with recalled memories of similar but distinct examples to form a richer and more discriminative understanding.
>
> ## 9. I do not agree with all stated biological constraints: as is well noted in the literature, symmetric forward and backward weights can arise naturally provided that the same local learning is present at forward and backward site, and weight-decay is applied (easy to see).
>
> We are not entirely clear on the specific point about "symmetric forward and backward weights can arise naturally provided that the same local learning is present at forward and backward site, and weight-decay is applied". We would be very grateful if you could please elaborate further on this. Thank you.

---

> > ### Author Response · Authors · 2025-11-20
> >
> > ## 10. There is a misunderstanding of how the brain seems to employ global error signals: first, multiple neuromodulatory signals are thought to be quite a-specific, like dopamine being broadcasted widely and non-targeted. Moreover, to the degree that dopamine signals a global reward prediction error, negative errors are believed to be encoded as a decrease from a steady baseline, thus resolving the issue with error-signs.
> > We thank the reviewer for this insightful comment regarding the role of neuromodulatory signals. We agree that our discussion of global error signals could be more nuanced.
> >
> > We are aware of the mechanisms involving neuromodulators like dopamine, which indeed act as a form of global signal. However, the primary challenge for a learning algorithm is the problem of *credit assignment*: how a global reward signal can be precisely attributed to each individual synapse to guide its update.
> >
> > While dopamine can encode a global reward prediction error, it remains unclear how such a widely broadcast, a-specific signal could provide the precise, synapse-specific credit information required to train a deep network. Furthermore, to the best of our knowledge, there is currently no biological evidence demonstrating that this neuromodulatory mechanism can operate at the fine-grained level necessary for complex perceptual tasks, such as distinguishing the subtle visual features required for modern image classification, which demands highly specific updates to millions of synaptic weights.
> >
> > Therefore, our C2 criterion, local error representation, is framed specifically in the context of the credit assignment problem. It posits that the signals directly driving synaptic plasticity are computed locally, rather than being derived from a globally broadcast error signal whose connection to individual synaptic updates is not yet understood. We will add this important discussion to the revised manuscript to clarify the scope and motivation of our biological constraints.
> >
> > ## 11. typo l054: "spiking trains"
> > We thank the reviewer for their careful reading and for pointing out this typo. You are correct, the term should indeed be "spike trains." We will correct this in the revised manuscript.
> >
> > ## 12. What is the effect of having spikes? In 5.3 it is noted that spikes degrade accuracy compared to ANNs. For the proposed architecture, it would seem the proposed approach would also work for identically structured non-spiking RNNs. Does this improve performance?
> >
> > We thank the reviewer for this fundamental question, which touches upon the core motivation of our work and the broader field of SNN research.
> >
> > The introduction of spikes is a design choice central to our goal of biological plausibility. Our motivation stems from the principle that biological neurons communicate via discrete spikes, not continuous floating-point values. Exploring learning mechanisms that operate with spikes is a foundational direction in biologically plausible algorithms, which is precisely why we introduced Criterion C4 (Model of Neurons) to ensure our model operates under this constraint.
> >
> > As the reviewer correctly notes, and as we state in Section 5.3, using spikes often results in a performance gap compared to identically structured ANNs. The primary goal of employing SNNs is not to outperform ANNs in raw accuracy, but rather to explore powerful computational models that adhere more closely to the principles of neural computation, such as energy efficiency and event-driven processing [3]. In this context, the contribution of our work is not that spikes improve performance over non-spiking networks, but that our BSD algorithm enables a spiking network to achieve performance *comparable* to its BP-trained counterpart while uniquely satisfying all five of our proposed biological plausibility criteria.
> >
> > Regarding the specific case of RNNs, our proposed architecture is indeed fully spike-based, which presents additional complexities compared to the CNN case. We provide a detailed discussion and the mathematical derivations for our spike-based RNN architecture in Appendix C to address these complexities.

---

> > > ### Author Response · Authors · 2025-11-20
> > >
> > > ## 13. How does the approach scale? All examples are relatively small/shallow networks, whereas it is known that similar ELBO-like losses tend to not scale well. Can you comment Deeper networks?
> > > We thank the reviewer for this excellent question regarding the scalability of our approach.
> > >
> > > The primary focus of this work is to develop a novel learning algorithm that satisfies all five of our proposed criteria for biological plausibility, a central and challenging research goal within the biologically plausible learning community. Our experiments were therefore designed to demonstrate this core contribution on standard benchmark architectures, showing that it is possible to achieve performance competitive with global algorithms like backpropagation while adhering to these strict constraints.
> > >
> > > It is a standard practice in the field of biologically plausible learning to first validate new algorithms on well-understood benchmark architectures. Our work adheres to this methodology. The architectures employed in our study are consistent with, and in some cases deeper than, those utilized in recent and contemporary works [4][5][6]. Moreover, our work extends beyond many of these foundational studies by demonstrating the versatility of BSD across a wider range of tasks, including classification, sequential regression, and image generation, and on multiple model types (MLPs, CNNs, and RNNs).
> > >
> > > That said, we agree that exploring the scalability of BSD to deeper networks is a very important and promising direction for future work. However, we believe that successfully scaling our method would likely require the integration of corresponding large-scale pre-training techniques, the exploration of which is beyond the scope of the current paper.
> > >
> > > ## 14. How does ReCo scale with batch size?
> > >
> > > We thank the reviewer for this valuable question regarding the scaling properties of ReCo loss. To address this, we conducted an ablation study to analyze the impact of batch size on the performance of our BSD-trained CNN across the SVHN, CIFAR-10, and CIFAR-100 datasets.
> > >
> > > The results are summarized in the table below.
> > >
> > > |Batch Size|SVHN|CIFAR-10|CIFAR-100|
> > > |:-:|:-:|:-:|:-:|
> > > |16|71.55%|68.96%|34.99%|
> > > |32|78.18%|78.07%|42.53%|
> > > |64|90.97%|81.54%|49.15%|
> > > |128|90.81%|84.53%|53.11%|
> > > |256|88.97%|83.85%|53.49%|
> > >
> > > The results demonstrate a clear trend. At smaller batch sizes, the model's performance is limited. This is an expected behavior for contrastive learning methods like ReCo, which require a sufficiently large and diverse set of negative samples within each batch to effectively shape the representation space.
> > >
> > > As the batch size increases, we observe substantial performance gains across all datasets. This is because larger batches provide a more robust estimate of the data distribution, supplying a richer set of negative examples for the contrastive objective and enabling the model to learn a more effective and generalizable representation space.
> > >
> > > Once the batch size reaches 128, the performance begins to stabilize or saturate. Further increasing the batch size to 256 yields minor fluctuations in accuracy. This suggests that a batch size of around 128 provides a sufficient number of negative samples for the model to achieve robust performance on these datasets. These findings confirm that our BSD framework scales favorably with batch size, which is consistent with the principles of contrastive learning.

---

> > > > ### Author Response · Authors · 2025-11-20
> > > >
> > > > ## 15. Can you demonstrate why layer-wise ReCo works as well as it does, compared to e.g. non detached ReCo?
> > > >
> > > > We thank the reviewer for this question, which allows us to clarify a critical aspect of our method's design and its direct connection to our biological plausibility criteria.
> > > >
> > > > The use of the `detach()` operation is a fundamental design choice in our framework. Its purpose is to ensure that when we locally train the weights $\mathbf{W}$ and $\boldsymbol{\Theta}$ for a given layer, the learning signals from that layer's ReCo loss are confined and do not propagate to other layers. This technique of severing the computational graph to enforce local updates is also a key component in other biologically plausible algorithms, such as the Forward-Forward algorithm.
> > > >
> > > > By severing the computational graph in this manner, we ensure our framework satisfies Criterion C2 (local synaptic plasticity), which mandates that the weight update for any given layer must be driven by a purely local signal, rather than by error gradients propagated back from other layers via the chain rule.
> > > >
> > > > However, to directly address the reviewer's question, we have conducted an ablation study comparing our proposed `detached` approach with a `non-detached` version. In the `non-detached` variant, we allow the gradients from each layer's ReCo loss to flow back through the entire network. The experiments were run on CIFAR-10 and SVHN using the same CNN architecture and hyperparameters as in our main paper.
> > > >
> > > > | Method          | CIFAR-10 Accuracy (%) | SVHN Accuracy (%) |
> > > > |:----------------|:---------------------:|:-----------------:|
> > > > | BSD-detached    | 84.13                 | 90.81             |
> > > > | BSD-non-detached| 88.06                 | 94.02             |
> > > >
> > > > The results show that the non-detached version achieves a higher accuracy on both datasets. However, the performance gain comes at the cost of violating our fundamental principle of local learning. In the non-detached setting, the update for a given layer's weights is no longer local; it is influenced by the losses from all subsequent layers. This reintroduces a dependency on non-local information, which is precisely the kind of biologically implausible mechanism that our C2 criterion, and our BSD framework as a whole, is designed to avoid.
> > > >
> > > > Therefore, while a non-detached approach can function, we chose the `detached` method as it is the one that correctly adheres to the biological constraints that are central to this work.
> > > >
> > > >
> > > > [1] Hinton, G., Vinyals, O., & Dean, J. (2015). Distilling the knowledge in a neural network. *arXiv preprint arXiv:1503.02531*.
> > > >
> > > > [2] Rao, R. P., & Ballard, D. H. (1999). Predictive coding in the visual cortex: a functional interpretation of some extra-classical receptive-field effects. *Nature neuroscience*, *2*(1), 79-87.
> > > >
> > > > [3] Lv, C., Xu, J., & Zheng, X. (2023). Spiking Convolutional Neural Networks for Text Classification. *The Eleventh International Conference on Learning Representations*.
> > > >
> > > > [4] Salvatori, T., Song, Y., Yordanov, Y., Millidge, B., Sha, L., Emde, C., ... & Lukasiewicz, T. (2024). A Stable, Fast, and Fully Automatic Learning Algorithm for Predictive Coding Networks. *The Twelfth International Conference on Learning Representations*.
> > > >
> > > > [5] Pinchetti, L., Qi, C., Lokshyn, O., Olivers, G., Emde, C., Tang, M., ... & Salvatori, T. (2024). Benchmarking Predictive Coding Networks – Made Simple. *arXiv preprint arXiv:2407.00994*.
> > > >
> > > > [6] Bartunov, S., Santoro, A., Richards, B., Marris, L., Hinton, G. E., & Lillicrap, T. (2018). Assessing the scalability of biologically-motivated deep learning algorithms and architectures. *Advances in Neural Information Processing Systems, 31*.

---

### Official Review · Reviewer_SDT5 · 2025-10-29

**Soundness:** 3
**Presentation:** 3
**Contribution:** 3
**Rating:** 6
**Confidence:** 5

**Summary:**

This paper proposes the Bidirectional Spike-based Distillation (BSD) algorithm. Inspired by the brain’s bidirectional perception-recall mechanism, BSD jointly trains a feedforward spiking network (for stimulus-to-concept perception and decision-making) and a backward spiking network (for concept-to-stimulus memory reconstruction), framing learning as a transformation between stimulus encoding and concept encoding. Through experiments across multiple tasks (image classification, image generation, sequential regression) and network architectures (MLPs, CNNs, RNNs, autoencoders), BSD is verified to achieve performance comparable to classical error backpropagation while satisfying all five biological plausibility criteria.

**Strengths:**

1. BSD is the an algorithm to simultaneously meet five core biological plausibility criteria, resolving conflicts between existing methods and neurobiological principles in aspects such as spike signal utilization and error signal type, thus providing an effective framework for biologically grounded spike-driven learning.

2. Experiments cover multiple tasks (e.g., image classification on MNIST/CIFAR-10, image generation evaluated via FID, sequential regression including text prediction and traffic forecasting) and various network architectures, verifying BSD’s strong generalization ability and task adaptability.

3. The algorithm design is closely aligned with the brain’s bidirectional mechanism of bottom-up perception and top-down recall, utilizing the basal and apical dendrites of pyramidal neurons to process feedforward and feedback signals respectively, resulting in a theoretical foundation highly consistent with neuroscientific observations.

**Weaknesses:**

1. It would be beneficial to investigate the influence of different timesteps on the BSD network's performance through additional experiments.

2. Further discussions on the impact of batch normalization on the convergence and overall performance of the BSD algorithm could be included.

3. Considering the BSD algorithm's dual forward and backward branches, it presumably requires a larger amount of GPU memory, which could be a relevant consideration for practical applications.

**Questions:**

1. The paper mentions that the number of time steps is set to 4 for MLP, CNN, and RNN tasks (e.g., Section E.2.2, E.2.3) and 8 for autoencoder-based image generation tasks (Section E.2.4), but no analysis of the impact of different time step values on performance is provided. Do you plan to supplement experiments with varying time steps (e.g., 2, 6, 10) to explore how time step adjustments affect BSD’s performance across different tasks (image classification, image generation, sequential regression)?

2. Batch normalization is applied in BSD-CNNs (after each convolutional layer, Section E.2.2) and BSD-RNNs (for the hidden layer in Pems-bay, Section E.2.3) to stabilize spiking activity, but the paper lacks quantitative and qualitative discussions on its specific impact. Could you provide experimental results comparing the convergence speed (e.g., number of epochs to reach stable performance) and final task performance (e.g., classification accuracy, FID, MSE) of BSD with and without batch normalization?

3. The BSD algorithm adopts dual forward (Type 1 neurons) and backward (Type 2 neurons) branches with independent weight matrices (W for forward, Θ for backward, Section 4.2), which may increase GPU memory consumption. Have you quantified the GPU memory usage of BSD compared to classical error backpropagation (on ANNs/SNNs) and other biologically plausible algorithms (e.g., DLL, CCL) under the same network architecture and batch size?

---

> ### Author Response · Authors · 2025-11-20
>
> Thanks for your comments and suggestions, which are valuable for enhancing our paper. We are cheerful that our contributions are well recognized.
>
> ## 1. Investigate the influence of different timesteps on the BSD network's performance through additional experiments.
> We thank the reviewer for this valuable suggestion. To more comprehensively investigate the influence of the number of timesteps ($T$) on our model's performance, we expanded our ablation study across multiple datasets and for both our CNN and RNN architectures, keeping all other hyperparameters consistent with our main experiments.
>
> The results, summarized in the tables below, reveal a consistent and informative trend across both model types.
>
> **CNN Performance on Image Classification Tasks:**
> |Timesteps ($T$)|CIFAR-100|SVHN|CIFAR-10|
> |:-:|:-:|:-:|:-:|
> |2|47.12%|82.54%|78.09%|
> |4|53.48%|90.81%|84.53%|
> |6|53.11%|90.35%|82.28%|
> |8|53.24%|90.94%|81.51%|
>
> **RNN Performance on Sequential Regression Tasks:**
> |Timesteps ($T$)|Harry Potter (acc $\uparrow$)|Metr-la (mse $\downarrow$)|
> |:-:|:-:|:-:|
> |2|0.4185|0.1280|
> |4|0.4169|0.1245|
> |6|0.4168|0.1248|
> |8|0.4212|0.1238|
>
> For both CNN and RNN models, performance generally improves when increasing the number of timesteps from $T=2$ to $T=4$. For timesteps greater than four ($T \geq 4$), performance tends to stabilize, exhibiting only minor fluctuations across most datasets and metrics. This suggests that four timesteps provide a robust and effective balance, allowing neurons to integrate sufficient information to form rich representations without introducing excessive temporal complexity.
>
>
> These expanded findings empirically justify our choice of $T=4$ as a robust and efficient setting across a variety of tasks and architectures, demonstrating a crucial trade-off between representational capacity and learning stability.
>
> ## 2. Discussions on the specific impact of batch normalization
> We thank the reviewer for prompting this valuable analysis. To more broadly investigate the specific impact of batch normalization (BN), we expanded our ablation study to include the SVHN, CIFAR-10, and CIFAR-100 datasets. The experiments were conducted on our BSD-trained CNN, and the results are summarized below.
>
> |Method|SVHN|CIFAR-10|CIFAR-100|
> |-|:-:|:-:|:-:|
> |BSD without BN|80.68%|80.75%|53.31%|
> |BSD with BN|90.81%|84.53%|53.48%|
>
> The results show that incorporating BN improves performance across all tested datasets. This benefit is attributed to the ability of BN to stabilize the distribution of membrane potentials across layers. The normalization ensures that spike sparsity remains consistent, leading to more stable and effective learning.
>
> Furthermore, we consistently observed that BN significantly accelerates the training process across all datasets. For example, on CIFAR-10, the model equipped with BN surpassed 70\% validation accuracy within just 5 epochs, whereas the model without BN required 11 epochs to reach the same performance level. This accelerated convergence can be attributed to the smoother optimization landscape and stabilized learning dynamics provided by BN.
>
> ## 3. GPU memory usage of BSD compared to classical BP.
>
> To provide a direct comparison of practical memory costs, we measured the GPU memory consumption for both BSD and a classical BP-trained SNN. All experiments were conducted on CIFAR-10 using the 5-layer CNN architecture described in our appendix, ensuring a fair comparison.
>
> During inference for classification tasks, the BSD framework utilizes only its feedforward pathway (weights $\mathbf{W}$). Since this pathway is architecturally identical to a BP-trained SNN of the same configuration, the computational graph and parameters used during a forward pass are the same.
> |Method|Memory Consumption (MB)|
> |:-:|:-:|
> |BP|1826|
> |BSD|1826|
>
> During training, the BSD algorithm must maintain the parameters and activations for both its feedforward ($\mathbf{W}$) and backward ($\mathbf{\Theta}$) pathways simultaneously. This architectural requirement results in a higher memory footprint compared to classical BP, which only needs to store activations for a single backward pass. The experimental results below confirm this expected trade-off, which is inherent to our dual-pathway, biologically plausible design.
>
> |Method|Memory Consumption (MB)|
> |:-:|:-:|
> |BP|3324|
> |BSD|5034|
>
> In summary, while BSD requires more memory during training as a consequence of its design, its memory usage during inference is efficient and competitive with standard methods.

---

> > ### Comment · Reviewer_SDT5 · 2025-11-22
> >
> > Thanks for the authors' reply. After reading other reviews and the authors' rebuttal, I confirm that this paper is well-motivated and technically-solid. Therefore, I will raise my score to 8. Good luck.

---

### Official Review · Reviewer_xq86 · 2025-11-01

**Soundness:** 3
**Presentation:** 3
**Contribution:** 3
**Rating:** 6
**Confidence:** 4

**Summary:**

This paper introduces Bidirectional Spike-Based Distillation (BSD), a biologically plausible learning algorithm inspired by the brain's bidirectional perception-recall architecture. BSD transforms spiking representations between stimulus encoding (feedforward path for perception) and concept encoding (feedback path for recall), satisfying five biological plausibility criteria (e.g., asymmetric weights, local plasticity). Experiments on image classification, sequential regression, and generation tasks demonstrate that BSD achieves performance comparable to error backpropagation while maintaining stronger neurobiological fidelity.

**Strengths:**

1. BSD fulfills all five proposed biological criteria, i.e., asymmetric weights, local error signals, non-two-stage learning, spiking neuron models, and unsigned error encoding, making it more aligned with neurobiological principles than backpropagation-based methods.

2. BSD achieves accuracy close to backpropagation on diverse benchmarks and robustly handles sequential regression and image generation, proving practicality without sacrificing biological grounding.

**Weaknesses:**

1. Discrete spike-based communication reduces information capacity compared to continuous activations, leading to performance degradation in SNNs across all tasks.

2. The description of the spike generation and information encoding lacks depth. While the LIF neuron dynamics are provided, the paper does not specify the exact strategy for converting inputs (like images) and targets into the initial spike trains (s1 and $\hat{s}$) for the feedforward and backward paths. The choice of encoding scheme (e.g., rate coding, temporal coding) can significantly impact performance, and its omission is a notable gap. Furthermore, the distillation process between the two pathways is described at a high level as feature alignment using the ReCo loss, but the precise mechanisms of how the error signals from this loss guide the updates of the distinct weight matrices (W and $\theta$) need a more detailed explanation, potentially including a mathematical derivation.

3. The selection of the Relaxed Contrastive (ReCo) loss over other alternatives is justified by its unsigned nature and performance in ablations. However, a more rigorous theoretical or empirical justification for why it is particularly suited for aligning voltage signals in spiking neurons, as opposed to other contrastive losses, would strengthen the argument. Similarly, the choice of specific hyperparameters, such as the penalty weight $\lambda$ = 0.6 and the firing thresholds for different neuron types and architectures, appears to be determined empirically. The paper would benefit from discussing the sensitivity of the results to these choices or providing a principled rationale for them.

4. The paper introduces two new biological plausibility criteria (C4 and C5) and claims BSD satisfies all five. However, the argument for satisfying C2 (local synaptic plasticity) is somewhat superficial. It states that error computation is localized using the ReCo loss, but a more detailed analysis is needed to explain how the weight updates for W and $\theta$ rely only on locally available information (e.g., pre- and post-synaptic spikes and local voltages) without any implicit non-local dependencies. A detailed diagram or step-by-step explanation of the local update rule for a single synapse would make this claim more credible.

5.  The experiments demonstrate performance on standard benchmarks but do not thoroughly test the robustness of the BSD-trained models. For instance, how do these models perform on corrupted data, out-of-distribution examples, or under varying noise conditions? Including such robustness evaluations would be crucial for claiming the practical viability of the algorithm. Furthermore, the scalability of BSD is asserted but not conclusively proven. Testing on a larger-scale, more complex dataset (e.g., ImageNet) would provide much stronger evidence for its scalability beyond relatively small datasets like CIFAR-100.

6. A key claim of the paper is that BSD is more biologically plausible. However, the experiments primarily focus on task performance metrics (accuracy, FID, MSE). There is a lack of direct analysis and quantification of the purported biological properties themselves. For example, the paper could analyze the degree of weight symmetry/alignment between the forward (W) and backward ($\theta$) pathways during training. Figure 7 shows this alignment is low for most layers, which supports adherence to C1, but a more quantitative discussion of this property across different tasks and architectures would be valuable. Similarly, the energy efficiency of the spiking models, a major claimed advantage of SNNs, is not measured or compared to analog networks.

7. In the image generation task, the qualitative results (Figure 2) and FID scores are presented. However, the FID scores for BSD are generally worse than those for BP-trained models. The paper states the quality is comparable, but a more critical discussion of the performance gap and the specific challenges BSD faces in generative tasks (e.g., possibly due to the discrete nature of spikes) would be more balanced. The visual samples in Figure 2 also lack a detailed description of what specific shortcomings (e.g., blurriness, lack of detail) are observed in the BSD-generated images compared to the BP baseline.

**Questions:**

See Weaknesses

---

> ### Author Response · Authors · 2025-11-20
>
> Thanks for your comments and suggestions, which are valuable for enhancing our paper. We are cheerful that our contributions are mostly recognized.
>
> ## 1. Performance degradation in SNNs across all tasks.
>
> First, it is crucial to frame the motivation for SNN research. The community pursues SNNs because they operate on principles fundamental to brain computation, such as discrete, event-driven communication. This paradigm offers inherent advantages in energy efficiency, particularly on neuromorphic hardware.
>
> Second, we acknowledge that a performance gap currently exists between SNNs and their ANN counterparts. However, this is a shortcoming of the SNNs, stemming from challenges like information loss due to spike-based discretization, and not a limitation of our proposed BSD training method. Even state-of-the-art SNN models like QKFormer [1], which are trained using the powerful but biologically implausible surrogate gradient backpropagation method, do not yet surpass leading ANN architectures like Vision Transformers on large-scale benchmarks.
>
> In this context, BSD's contribution is clear. It achieves performance comparable to the BP-trained SNN baseline, yet critically, it does so while satisfying all five of our proposed biological plausibility criteria.
>
> ## 2. The description of the spike generation and information encoding lacks depth.
> We thank the reviewer for these insightful questions and will clarify these points.
>
> Regarding the strategy for converting inputs and targets into spike trains, we employ a "direct encoding" scheme. For an input image, the normalized pixel intensities are fed as a constant input current to the first-layer neurons for the entire duration of the simulation ($T$ timesteps). The first layer then processes this current to generate the initial spike train. This is a standard and effective method for handling static images in high-performance SNNs and is widely adopted in the community [2][3][4]. For the classification target, the label is converted into a one-hot vector, which is then repeated for $T$ timesteps to form the target spike train.
>
> Regarding the precise update mechanism, the core principle is that the "feature representations" we align are the pre-spike membrane potentials of the neurons. The error signals from the local ReCo loss at each layer directly guide the weight updates via a standard gradient descent step. For example, the feedforward weights $\mathbf{W}$ are updated via gradient descent as $\mathbf{W}\_{i-1}^{\text{new}} = \mathbf{W}\_{i-1}^{\text{old}} + \eta\_{\mathbf{W}} \cdot \frac{\partial \mathcal{L}\_i}{\partial \mathbf{W}\_{i-1}}$, for $i=2, \dots, L-1$, where the superscripts 'new' and 'old' denote the weights after and before the update,  $\eta\_{\mathbf{W}}$ is the learning rate for the feedforward weights, and the gradient term is the one derived in Equation 8. Similar updates are applied to the backward weights $\mathbf{\Theta}$.
>
> This joint training process is effective for several reasons. The training can be viewed as an iterative optimization where both pathways simultaneously seek to minimize a shared objective. This process, akin to alternating minimization, ensures that the objective is non-increasing and drives the system toward a stable solution. This approach yields several key benefits:
>
> 1)  The layer-wise alignment acts as a form of deep supervision or consistency constraint. This provides each layer with a direct error signal, improving training stability and the quality of the learned representations.
>
> 2)  The dual pressure from the final task loss and the layer-wise alignment forces the feedforward pathway to retain information-rich representations, preventing the loss of useful features necessary for the backward path to reconstruct its target.
>
> 3)  Relying solely on the final task loss can lead intermediate layers to learn "sufficient but barren" representations. The alignment acts as an auxiliary constraint, restricting the solution space to representations that can both solve the task and support reconstruction, thereby reducing the likelihood of pathological solutions.
>
> 4)  The framework balances two objectives: the task loss pushes the network to learn discriminative features, while the alignment objective pushes it to preserve input information. This trade-off often leads to more robust and generalizable representations than training with a single objective.

---

> ### Author Response · Authors · 2025-11-20
>
> ## 3. A more rigorous theoretical or empirical justification for why it is particularly suited for aligning voltage signals in spiking neurons, as opposed to other contrastive losses.
> We thank the reviewer for this excellent insight, which prompts a deeper justification for our choice of the Relaxed Contrastive (ReCo) loss.
>
> Our ablation study in **Table 4 (Line 417)** provides the empirical evidence, showing that BSD with ReCo loss consistently and significantly outperforms BSD-InfoNCE, especially on complex datasets like CIFAR-100 where the margin is over 15 percentage points. The theoretical rationale for this superiority, particularly for aligning voltage signals in our BSD algorithm, is as follows.
>
> The core objective at each hidden layer is to align the pre-spike membrane voltage from the feedforward path ($\mathbf{v}\_{i,k}$) with its corresponding supervisory voltage generated by the backward path ($\hat{\mathbf{v}}\_{i,k}$). Simultaneously, the network must distinguish this pair from all negative pairs, where the voltage $\mathbf{v}\_{i,k}$ is paired with supervisory signals from other samples in the batch ($\hat{\mathbf{v}}\_{i,j}$ for $j \neq k$).
>
> A standard InfoNCE loss would enforce this distinction by applying a uniform repulsive force to all negative pairs. This compels the membrane voltage pattern generated for one sample to be anti-correlated with the supervisory voltage patterns of all other samples. This is an overly restrictive constraint for high-dimensional membrane potentials. Forcing such a global anti-correlation for every negative pair can make the learning task unnecessarily difficult, potentially warping the feature space and hindering the network's ability to learn flexible representations.
>
> In contrast, the ReCo loss adopts a more targeted and flexible objective. It applies a penalty only when the voltage for one sample is confusingly similar to the supervisory signal of another (i.e., has a positive cosine similarity). If a negative pair is already orthogonal or negatively correlated, its contribution to the loss is zero. This is particularly suited for our framework as it focuses the learning effort where it is most needed: separating confusable voltage signals. Instead of wasting capacity pushing already distinct representations further apart, the network can concentrate on refining the boundaries between similar ones. This allows for a richer and more flexible representational space for the membrane potentials, leading to improved final performance, as is empirically validated by our ablation studies.
>
> ## 4. The choice of specific hyperparameters, like penalty weight $\lambda$ and firing threshold.
> We agree that a principled discussion of hyperparameter sensitivity is important. We provided an empirical analysis for both hyperparameters in Section 5.7, and we are happy to elaborate on the rationale here.
>
> Regarding the penalty weight $\lambda$, our ablation study in **Figure 3(a)** demonstrates that the model's performance shows only minor variations for different positive values of $\lambda$. The exception is when $\lambda=0$, which causes a significant performance drop. This is expected, as setting $\lambda$ to zero removes the repulsive term entirely, eliminating the contrastive pressure that separates the representations of different samples and risking a representational collapse. For non-zero values, the choice of $\lambda$ creates a subtle trade-off. A very small value may not provide a sufficient repulsive force to effectively distinguish between confusable voltage signals. Conversely, an overly large value could potentially over-emphasize the separation of negative pairs, which might slightly detract from the primary objective of aligning positive pairs. Our analysis indicates that the model is not highly sensitive to the precise value of $\lambda$. We empirically observed in Figure 3(a) that $\lambda=0.6$ yielded a marginally better result, and it was thus selected as our default value.
>
> Similarly, for the firing threshold, our analysis in **Figure 3(b)** provides an empirical justification for our selection. The choice reflects a critical trade-off between representational density and information flow. A threshold that is too low results in dense, high-frequency spiking. This can saturate the network and make the spike patterns for different input samples less discriminative, thereby hindering the learning of distinct features. Conversely, a threshold that is too high leads to overly sparse activity, creating an information bottleneck where too few spikes are generated to propagate rich feature signals through the network. This makes the distillation process ineffective, as many neurons may become "silent" and cease to contribute meaningfully to the learning. The optimal threshold of 0.6 thus appears to strike a balance, enabling enough sparsity to form distinct representations while preserving the robust information flow necessary for effective distillation.

---

> > ### Author Response · Authors · 2025-11-20
> >
> > ## 5. The argument for satisfying C2 (local synaptic plasticity) is somewhat superficial. A more detailed analysis is needed to explain how the weight updates for $W$ and $\Theta$ rely only on locally available information.
> > We thank the reviewer for this question and appreciate the opportunity to provide a more detailed analysis of how our algorithm satisfies Criterion C2 (local synaptic plasticity).
> >
> > A core concept of C2 is that the update for a synapse should rely only on information locally available to it, without access to a global error signal propagated from the final output layer (as is required by backpropagation). Our BSD framework strictly adheres to this principle.
> >
> > The ReCo loss for each hidden layer i is a local objective, computed solely by performing a contrastive comparison between the pre-spike membrane potentials generated within that specific layer ($\mathbf{v}\_i$ from the feedforward path and $\hat{\mathbf{v}}\_i$ from the backward path). Crucially, this local loss does not use the global classification label, which is only used for the loss at the final output layer.
> >
> > The precise mechanism is detailed in our mathematical derivations. The gradient for the feedforward weights $\mathbf{W}\_{i-1}$ is provided in Equation 8, and the full derivations for both $\mathbf{W}$ and $\mathbf{\Theta}$ are available in Appendix D. As shown in Equation 8, the calculation of the gradient $\frac{\partial \mathcal{L}\_i}{\partial \mathbf{W}\_{i-1}}$ relies exclusively on three pieces of information: 1) the pre-synaptic spike train from the previous layer, $\mathbf{s}\_{i-1}$; 2) the post-synaptic basal membrane potential, $\mathbf{v}_i$; and 3) the post-synaptic apical membrane potential, $\hat{\mathbf{v}}\_i$.
> >
> > All of these quantities are available locally at the neuron or synapse being updated. This locally computed gradient is then used to update the weights via a standard gradient descent step: $\mathbf{W}\_{i-1}^{\text{new}} = \mathbf{W}\_{i-1}^{\text{old}} + \eta\_{\mathbf{W}} \frac{\partial \mathcal{L}\_i}{\partial \mathbf{W}\_{i-1}}$, where $\eta\_{\mathbf{W}}$ is the learning rate. A symmetric update rule, also based on a purely local gradient, is applied to the backward weights $\mathbf{\Theta}$.
> >
> > Therefore, the updates for both $\mathbf{W}$ and $\mathbf{\Theta}$ are governed by purely local membrane potentials, satisfying the conditions for local synaptic plasticity.
> >
> > ## 6. How do these models perform on corrupted data, out-of-distribution examples, or under varying noise conditions
> > e thank the reviewer for this insightful question regarding the model's performance on corrupted data, which is a crucial aspect of robustness. To address this, we conducted an experiment to evaluate the models' resilience to noise.
> >
> > For this analysis, we used the model checkpoints that were trained on the clean training sets of CIFAR-10 and SVHN. We show the baseline performance of our BSD-trained CNN and a backpropagation (BP) baseline on the original, clean test sets, which are denoted as "Clean Accuracy".
> >
> > Next, we evaluated the adversarial robustness of these same clean-trained models. During the inference phase, we introduced an adversarial attack by adding Gaussian noise (mean=0, std=0.05) to the input images. The performance of the models on this corrupted data, which we term Accuracy Under Attack (AUA), is shown in the table below.
> >
> > |Method| |CIFAR-10 (%)|SVHN (%)|
> > |:-:|:-|:-:|:-:|
> > |BP|Clean Accuracy|87.18|94.31|
> > | |Accuracy Under Attack|70.89|89.24|
> > |BSD|Clean Accuracy|83.67| 90.81|
> > | |Accuracy Under Attack|67.06|85.40|
> >
> > The results demonstrate that our BSD-trained model exhibits a competitive robustness that is comparable to the BP-trained baseline. On both CIFAR-10 and SVHN, the absolute drop in accuracy for BSD is remarkably close to that of BP, differing by less than 0.4%. Given that BSD is a biologically plausible method, achieving a resilience to noise that is on par with the powerful but implausible BP baseline is a significant result, indicating that our approach learns robust and effective features.
> >
> > We attribute this competitive robustness to the dual-objective nature of the BSD training process. The final task loss at the output layer pushes the network to extract discriminative information, while the layer-wise alignment objective simultaneously encourages the preservation of rich generative information from the input. This balance between two complementary goals acts as a form of regularization, guiding the model to learn more generalizable representations that are inherently more resilient to input perturbations, rather than learning brittle features that are only optimal for the clean data distribution.

---

> > > ### Author Response · Authors · 2025-11-20
> > >
> > > ## 7. There is a lack of direct analysis and quantification of the purported biological properties themselves. The energy efficiency of the SNN.
> > > We thank the reviewer for this question, as it provides an opportunity to explicitly detail how our BSD framework satisfies the proposed biological properties and to quantify the energy efficiency of the resulting models.
> > >
> > > First, we will clarify how the BSD algorithm adheres to each of the five biological plausibility criteria:
> > >
> > > *   **C1 (Asymmetric Synaptic Weights):** BSD employs two distinct and independently initialized weight matrices, $\mathbf{W}$ for the feedforward path and $\Theta$ for the backward path. As their update rules are derived from different local losses (detailed in Appendix D), their weights evolve asymmetrically throughout training.
> > >
> > > *   **C2 (Local Synaptic Plasticity):** The weight update for each layer is governed by a local ReCo loss, which is computed using only information available within that layer: the pre-synaptic spike train and the post-synaptic basal and apical membrane potentials. As shown in Equation 8, the gradient calculation depends only on local information, meaning the resulting weight update is also local.
> > >
> > > *   **C3 (Non-Two-Stage Training):** The feedforward and backward pathways operate concurrently. The computation of local losses and gradients does not require a strict temporal separation (e.g., a full forward pass followed by a full backward pass), allowing for simultaneous updates.
> > >
> > > *   **C4 (Model of Neurons):** Our framework is built entirely on the Leaky Integrate-and-Fire (LIF) neuron model, and all inter-neuronal communication is performed using discrete, binary spikes, as opposed to continuous-valued activations.
> > >
> > > *   **C5 (Unsigned Error Signaling in hidden layers):** The ReCo loss is an unsigned objective composed of two squared penalty terms: one that minimizes the deviation of positive pairs from perfect alignment, and another that penalizes positive similarity between non-corresponding pairs.
> > >
> > > Regarding the energy efficiency of SNNs, the primary benefit is realized during inference, particularly when deployed on neuromorphic hardware. Our BSD algorithm is a training method; the resulting inference model is a standard SNN that is architecturally identical to one trained with classical backpropagation. Therefore, a BSD-trained model fully retains the inherent energy-saving characteristics of SNNs.
> > >
> > > To quantify this, we follow the methodology proposed by Yao et al. [6] to estimate the theoretical energy consumption per inference sample for both our SNN model and its equivalent ANN counterpart. For a given SNN layer $l$, the energy consumption is estimated as: $\text{Energy}\_{\text{SNN}}(l) = E\_{\text{AC}} \times (T \times \gamma\_l \times \text{FLOPs}(l))$, where $T$ is the number of timesteps, $\gamma\_l$ is the layer's average firing rate, and $E\_{\text{AC}}$ is the energy per accumulate operation. For an ANN layer, the formula is: $\text{Energy}\_{\text{ANN}}(l) = E\_{\text{MAC}} \times \text{FLOPs}(l)$, where $E\_{\text{MAC}}$ is the energy per multiply-accumulate operation. The total energy is the sum over all layers. We use the energy constants for a 45nm process from the literature ($E\_{\text{AC}} = 0.9$ pJ, $E\_{\text{MAC}} = 4.6$ pJ) for our analysis of the BSD-trained 5-layer CNN on CIFAR-10 [6].
> > >
> > > A layer-wise breakdown of the energy consumption for both the SNN and ANN models is provided below.
> > >
> > > |SNN Layer|FLOPs|Avg Firing Rate ($\gamma$)|SOPs|Energy (J)|
> > > |:-|:-|:-:|:-|:-|
> > > |Layer 1 (Conv2d)|3.54e+06|0.5006|7.09e+06|6.38e-06|
> > > |Layer 2 (Conv2d)|3.77e+07|0.1519|2.29e+07|2.06e-05|
> > > |Layer 3 (Conv2d)|1.89e+07|0.1815|1.37e+07|1.23e-05|
> > > |Layer 4 (Conv2d)|9.44e+06|0.1943|7.34e+06|6.60e-06|
> > > |Layer 5 (Conv2d)|4.72e+06|0.2037|3.85e+06|3.46e-06|
> > > |Layer 6 (Linear)|5.12e+03|0.0803|1.65e+03|1.48e-09|
> > >
> > > |ANN Layer|FLOPs|Energy (J|
> > > |:-|:-|:-|
> > > |Layer 1 (Conv2d)|3.54e+06|1.63e-05|
> > > |Layer 2 (Conv2d)|3.77e+07|1.74e-04|
> > > |Layer 3 (Conv2d)|1.89e+07|8.68e-05|
> > > |Layer 4 (Conv2d)|9.44e+06|4.34e-05|
> > > |Layer 5 (Conv2d)|4.72e+06|2.17e-05|
> > > |Layer 6 (Linear)|5.12e+03|2.36e-08|
> > >
> > > The total energy consumption per inference sample is summarized in the table below.
> > >
> > > |Model|Total Operations per Sample|Inference  Energy per Sample (J)|Inference Energy Reduction|
> > > |:-|:-|:-:|:-:|
> > > |ANN (BP)|7.43e+07 (FLOPs)|3.42e-04|-|
> > > |SNN (BSD)|5.49e+07 (SOPs)|4.94e-05|85.5% $\downarrow$|
> > >
> > > This quantitative analysis confirms the substantial energy advantage of the SNN model. By leveraging sparse, event-driven computation, the SNN trained with our BSD method is approximately 85.5% more energy-efficient during inference than its architecturally identical ANN counterpart. This highlights that our biologically plausible training method yields models that are not only performant but also highly efficient for deployment.

---

> > > > ### Author Response · Authors · 2025-11-20
> > > >
> > > > ## 8. Testing on a larger-scale, more complex dataset (e.g., ImageNet).
> > > > We thank the reviewer for this important suggestion. Testing on larger-scale datasets is crucial for evaluating the scalability of any new learning algorithm.
> > > >
> > > > Due to our available computational resources (four NVIDIA RTX 3090 GPUs), conducting experiments on the full ImageNet dataset within the limited rebuttal period was not feasible. Therefore, to address the reviewer's request for evaluation on a more complex dataset, we chose to conduct experiments on the subset of ImageNet, Tiny-ImageNet, which consists of 200 classes with 500 training images and 50 test images per class.
> > > >
> > > > We evaluated our proposed BSD algorithm against both a standard backpropagation (BP) baseline and Dendritic Localized Learning (DLL) [5], another biologically plausible method discussed in our paper. For a fair comparison, we used the same 5-layer CNN architecture  described in Appendix E.2.2 across all methods. The results are summarized below.
> > > >
> > > > |Method|Accuracy on Tiny-ImageNet (%)|
> > > > |:-:|:-:|
> > > > |BP|41.07|
> > > > |Dendritic Localized Learning (DLL)|17.10|
> > > > |BSD (Ours)|35.34|
> > > >
> > > > The results demonstrate the performance of each method on complex tasks. Crucially, our proposed BSD algorithm achieves an accuracy of 35.34%, substantially outperforming Dendritic Localized Learning (DLL). This result is significant as it demonstrates that BSD scales effectively to challenging, larger-scale datasets. These results demonstrate that BSD scales effectively to more complex datasets, achieving competitive performance while maintaining its commitment to all five biological plausibility criteria.
> > > >
> > > > ## 9. FID scores for BSD are generally worse than those for BP-trained models. Figure 2 lack a detailed description of what specific shortcomings of BSD.
> > > > We thank the reviewer for this keen observation and for prompting a more detailed discussion of the specific shortcomings of BSD in generative tasks.
> > > >
> > > > The reviewer is correct in pointing out that the FID scores for our BSD-trained autoencoder are higher (worse) than those of the BP-trained ANN. This performance difference is an expected consequence of the core principle of our model: spike-based computation. The discrete nature of spikes inherently provides less granular information for image representation compared to the continuous floating-point values used in traditional ANNs.
> > > >
> > > > As can be observed in Figure 2, the BSD-generated images successfully capture the global structure, overall logic, and correct placement of objects, making them appear realistic at first glance. The specific shortcomings are primarily in the reconstruction of fine-grained details and high-frequency textures. For instance, the edges in the BSD-generated images appear slightly less sharp, and complex textures are smoothed or rendered with less fidelity compared to the BP baseline.
> > > >
> > > > While these individual differences may be subtle to the naked eye, the FID metric is highly sensitive to such distributional shifts in texture and detail. The accumulation of these minor imperfections across the image results in a higher FID score. This demonstrates that while BSD establishes a robust framework for plausible generative modeling with SNNs, the inherent discretization of information introduces a trade-off in reconstructing the finest details compared to its continuous-valued counterparts.
> > > >
> > > > [1] Zhou, C., Zhang, H., Zhou, Z., et al. (2024). QKFormer: Hierarchical Spiking Transformer using Q-K Attention. *Conference on Neural Information Processing Systems (NeurIPS)*.
> > > >
> > > > [2] Fang, W., Yu, Z., Chen, Y., Masquelier, T., Huang, T., & Tian, Y. (2021). Deep residual learning in spiking neural networks. *Advances in Neural Information Processing Systems, 34*, 21056-21069.
> > > >
> > > > [3] Zhou, Z., Zhou, Y., Wang, C., Li, S., & Li, G. (2023). Spikformer: When spiking neural network meets transformer. *International Conference on Learning Representations*, 2023.
> > > >
> > > > [4] Yao, M., Hu, J., Wu, Y., Hu, Y., Chen, Z., Yu, Z., Huang, T., & Liu, J. K. (2023). Spike-driven Transformer. *Advances in Neural Information Processing Systems*, 2023.
> > > >
> > > > [5] Lv C, Xu J, Lu Y, et al. Dendritic Localized Learning: Toward Biologically Plausible Algorithm[C]//Forty-second International Conference on Machine Learning.
> > > >
> > > > [6] Yao M, Zhao G, Zhang H, et al. Attention spiking neural networks[J]. IEEE transactions on pattern analysis and machine intelligence, 2023, 45(8): 9393-9410.

---

### Official Review · Reviewer_5Dww · 2025-11-01

**Soundness:** 3
**Presentation:** 3
**Contribution:** 3
**Rating:** 8
**Confidence:** 2

**Summary:**

This paper introduces Bidirectional Spike-based Distillation (BSD), a novel biologically plausible learning algorithm for neural networks. The work addresses a fundamental challenge in neural network research: developing learning methods that satisfy biological plausibility criteria while achieving competitive performance with backpropagation. BSD is inspired by the brain's bidirectional architecture integrating bottom-up sensory perception with top-down memory recall. The algorithm jointly trains feedforward and backward spiking networks, where the feedforward pathway maps stimuli to conceptual representations (analogous to perception and decision-making), while the backward pathway reconstructs stimuli from concepts (analogous to memory recall). The authors demonstrate that BSD satisfies five biological plausibility criteria: asymmetric forward and backward weights, local synaptic plasticity, single phase learning (eliminating sequential forward-backward passes), spiking neurons, and unsigned error signals. They validate the algorithm across diverse architectures and tasks.

**Strengths:**

**Biological motivation and plausibility:** The paper draws inspiration from the neuroscience principle of bidirectional processes that integrate bottom-up sensory perception with top-down memory recall. BSD implements this concept with high fidelity and satisfies five relevant criteria of biological plausibility, extending beyond the three criteria proposed by prior work.

**Comprehensive experimental validation:** The authors test the algorithm on multiple architectures (MLPs, CNNs, RNNs, autoencoders) and diverse datasets spanning image classification (MNIST, FashionMNIST, SVHN, CIFAR-10, CIFAR-100), text prediction and time-series forecasting (Harry Potter, Electricity, Metr-la, Pems-bay), and image generation tasks, demonstrating the generalizability of the approach.

**Novel algorithmic contributions:** The integration of bidirectional spiking networks trained via mutual distillation represents a unique approach to biologically plausible learning, distinguishing it from existing methods that typically compromise on some biological plausibility criteria. Moreover, BSD achieves performance comparable to backpropagation on all introduced tasks.

**Ablation study:** The paper includes an ablation experiment examining the impact of different loss functions (MSE, InfoNCE) and demonstrates that ReCo loss consistently outperforms alternatives.

**Weaknesses:**

**Limited discussion of practical implications and broader impact:** The authors do not adequately address the practical implications of their work. Key questions remain unanswered:
- Will BSD be adopted by the machine learning community, and if so, for what primary applications?
- Will the code be released as an integrated Python library compatible with current frameworks (PyTorch, JAX)?
- Alternatively, will BSD primarily serve the neuroscience community for computational experiments addressing neuroscience-related problems?

**Insufficient computational performance analysis:** Performance metrics including time complexity, memory consumption, and GPU usage should be clearly stated and compared to backpropagation in the main text.

**Minor:**
- In the code, clear instructions for reproducing the experimental results are not given.
- In Section 4.2, the notation is difficult to follow and contains apparent errors. Specifically, $\hat{v}'$ is never defined, while $v'$ and $\hat{v}$ are defined but never used in equations. The same symbol $v'$ is used to define Type 1 neurons in Equation 4 and Type 2 neurons in Equation 5, and vice versa for symbol $\hat{v}$. Additionally, in Section 4.3 (line 239), matrices of membrane voltages are defined but never used in subsequent equations.
- The application of FFT decomposition to input images and membrane voltages is mentioned but not adequately explained. This technique appears important for the method's performance on generation tasks, yet it receives minimal emphasis and should be expanded at least in the appendix.
- Font sizes in figures (especially Figure 1c and Figure 3) are excessively small and difficult to read while going through the main text.

**Questions:**

See weaknesses. Moreover, could you expand the observation that "The network attains its best performance at a firing threshold of 0.6, suggesting that thresholds that are either too low or too high impair effective learning" (line 461) with a possible explanation for this phenomenon?

---

> ### Author Response · Authors · 2025-11-20
>
> We are cheerful that our contribution is well recognized. And thanks for your valuable suggestions that truly enhance the quality of our paper and make it more understandable for the wider community.
> Responses to your concerns are presented as follows:
>
> ## 1. The authors do not adequately address the practical implications of their work. Will BSD be adopted by the machine learning community, and if so, for what primary applications?
> Thanks for your suggestions!
> Yes. We are confident that the Bidirectional Spike-Based Distillation (BSD) framework warrants adoption by the machine learning community.
> The core motivation of our work stems from the machine learning community’s long-standing quest for a feasible and effective alternative to Backpropagation (BP). The fundamental conflicts between BP's underlying mechanisms and neurobiology have been repeatedly highlighted, driving the need for new learning paradigms [1][2]. BSD directly meets this need through three critical advantages:
> First, BSD is among the first algorithms to satisfy all five biological plausibility criteria (C1-C5) while achieving performance competitive with BP.
> Second, the general-purpose nature of BSD allows it to seamlessly handle classification, sequential regression, and generation tasks, distinguishing it from other restricted, classification-only algorithms.
> Third, by exclusively utilizing discrete binary spikes, BSD inherently possesses the energy advantage of spiking neural networks, positioning it as a prime enabler for training power-efficient AI.
>
> ## 2. Will the code be released as an integrated Python library compatible with current frameworks (PyTorch, JAX)?
> Yes. As stated in the Section Reproducibility Statement of the manuscript, we promise to publicly release all source code on GitHub upon acceptance, which will be implemented in PyTorch. We have also provided comprehensive implementation details in Appendix E, including network architectures, training procedures, and all hyperparameter settings, to facilitate the community's replication of our reported results.
>
> ## 3. Alternatively, will BSD primarily serve the neuroscience community for computational experiments addressing neuroscience-related problems?
> Yes. We believe that our BSD algorithm, which was inspired by neuroscience, can in turn serve the neuroscience community. It provides a computational model that may help deepen the understanding of the brain's learning mechanisms.
>
> Furthermore, this work provides evidence that learning and information transfer via discrete spikes, a core principle of brain computation, is available. By achieving performance competitive with floating-point-valued networks, our method shows that powerful, brain-like computation is possible within a strictly spike-driven framework.
>
> ## 4. Time complexity, memory consumption, and GPU usage.
>  We provide a detailed analysis of the time complexity, memory consumption, and GPU usage for both BSD and a standard Backpropagation (BP) baseline.
>
> Experimental Setup: All experiments were conducted on an NVIDIA GeForce RTX 2080 Ti GPU. We used the CIFAR-10 dataset with a batch size of 128, employing the 5-layer convolutional network architecture detailed in Appendix E.2.2 of our paper for both BSD and the BP-trained SNN.
>
> For the inference phase, the computational profile of BSD is identical to that of the BP-trained SNN. This is because inference for classification exclusively uses the feedforward pathway (with weights $\mathbf{W}$), which is architecturally equivalent to the baseline BP model. Our empirical measurements, presented in the table, confirm this, showing identical memory consumption and GPU utilization.
>
> |Method| Memory Consumption|GPU Utilization|
> |-|-|--|
> |BP| 1826 MB| 0.33|
> |BSD| 1826 MB| 0.33|
>
> During training, BSD maintains two sets of weights ($W$ and $\Theta$) and the activations for both the feedforward and backward pathways. This leads to higher memory consumption compared to BP. GPU utilization remains similar, indicating a comparable computational load.
>
> |Method|Memory Consumption|GPU Utilization|
> |-|-|-|
> |BP| 3324 MB|0.36|
> |BSD|5034 MB|0.37|
>
> For a network with $L$ layers, $T$ timesteps, a batch size of $B$, and an average layer width of $D$, the complexities are:
>
> *   BPTT: The complexity is $O(L \cdot T \cdot B \cdot D^2)$, derived from the forward pass and a symmetric backward pass through the unrolled computation graph.
> *   BSD: The complexity is $O(L \cdot T \cdot B \cdot D^2 + L \cdot B^2 \cdot D)$. The first term, $O(L \cdot T \cdot B \cdot D^2)$, accounts for the propagation through the two pathways. The second term, $O(L \cdot B^2 \cdot D)$, arises from computing the $B \times B$ affinity matrix for the layer-wise ReCo loss, making our training complexity more sensitive to batch size.
>
> In summary, BSD has a higher training cost due to its dual-pathway architecture. However, its inference cost is the same as standard BP-trained SNNs.

---

> > ### Author Response · Authors · 2025-11-20
> >
> > ## 5. In the code, clear instructions for reproducing the experimental results are not given.
> > We sincerely apologize for the lack of clarity in the documentation accompanying the supplementary code. We appreciate you bringing this to our attention.
> >
> > As stated in our REPRODUCIBILITY STATEMENT, we promise to publicly release the full, well-commented source code upon acceptance. The code will be implemented in PyTorch to ensure it is accessible and compatible with current frameworks, and we will ensure the instructions are clear and easy to follow.
> >
> > Appendix E.2 offers comprehensive implementation details, covering network architectures, hyperparameter settings, training procedures, and evaluation metrics for experiments.
> >
> > ## 6. In Section 4.2, the notation is difficult to follow and contains apparent errors.
> >
> >
> > First, we will clarify the specific symbols. For a single neuron, the scalar membrane potentials from the basal and apical compartments are denoted as $v$ and $\hat{v}$, respectively. The symbol $\hat{v}'$ refers to the scalar apical potential of a Type 2 neuron. While these scalar notations are used in the descriptive text to explain the concept at a single-neuron level, Equations 4 and 5 operate on entire layers. The symbols in those equations, such as $\mathbf{v}_i$ and $\mathbf{v}'_i$, represent vectors formed by collecting these scalar potentials from all neurons in a given layer.
> >
> > Regarding the symbol mix-up, we apologize for the typos in the explanatory text following the equations. Equation 4 itself is correct, but the text after it will be revised to state: 'where $\mathbf{v}_i$ denotes the somatic membrane potential of Type 1 neurons in layer $i$ arising from basal dendritic integration, and $\hat{\mathbf{v}}'_i$ denotes the somatic membrane potential of Type 2 neurons in layer $i$ arising from apical dendritic integration.'
> >
> > Finally, regarding the matrices of membrane voltages defined in Section 4.3, they were introduced conceptually to motivate the use of the batch-wise ReCo Loss. We described stacking the per-sample vectors $\mathbf{v}_{i,k}$ and $\hat{\mathbf{v}}_{i,k}$ into matrices $\mathbf{V}_i$ and $\mathbf{\hat{V}}_i$ to make the concept of a batch-wise affinity matrix clear. For mathematical precision in the subsequent equations (Eq. 6-8), we correctly revert to the per-sample vector notation. We will ensure all notation is corrected and clarified throughout the manuscript in the revised version.
> >
> > ## 7. The application of FFT decomposition to input images and membrane voltages is mentioned but not adequately explained.
> >
> > The primary motivation for using FFT decomposition in our generation tasks is to apply a more nuanced, adaptive loss function that treats different types of reconstruction errors differently. A single, uniform penalty for reconstruction error is often suboptimal. In natural images, low-frequency components encode the overall structure and smooth regions, while high-frequency components capture sharp edges and fine-grained textures. A uniform loss forces an undesirable trade-off: a weak penalty may fail to reconstruct sharp details, leading to blurry images, while a strong penalty may over-penalize minor, perceptually insignificant deviations in smooth areas, potentially amplifying noise and creating artifacts. By using FFT, we can disentangle these components and apply a differential penalty, which is crucial for balancing the reconstruction of the image's global structure with the preservation of its fine-grained details.
> >
> > The process is as follows:
> >
> > First, we apply a 2D Fast Fourier Transform to the spatial tensor (e.g., a batch of membrane voltage maps of shape $[B, C, H, W]$), converting it into its frequency-domain representation.
> >
> > Second, to isolate these components, we construct frequency-domain masks. A low-pass filter is created by defining a circular region centered at the zero-frequency origin of the spectrum. Frequencies falling inside this radius are designated as low-frequency components, which represent the global structure and smooth areas of the image. Conversely, all frequencies outside this radius are designated as high-frequency components, which correspond to edges and fine-grained textures.
> >
> > Finally, by element-wise multiplying the frequency-domain representation with these two masks, we separate it into two distinct tensors: one containing only low frequencies and another containing only high frequencies. This separation enables the adaptive loss computation mentioned. We calculate the ReCo loss independently on these two components but use a different penalty weight $\lambda$ for each. A smaller $\lambda$ is applied to the low-frequency components to maintain structural coherence, while a larger $\lambda$ is applied to the high-frequency components to more strongly enforce the preservation of sharp details and edge fidelity. This approach allows the model to better preserve fine details without sacrificing the image's structural coherence.

---

> ### Author Response · Authors · 2025-11-20
>
> ## 8. Font sizes in figures (especially Figure 1c and Figure 3) are excessively small and difficult to read while going through the main text.
> We apologize for the small font sizes in the figures. We thank the reviewer for pointing this out, and we will increase the font sizes in figures 1c and 3, to ensure they are clear and legible in the revised manuscript.
>
> ## 9. Possible explanation for the firing threshold of 0.6.
> We thank the reviewer for this insightful question regarding the optimal firing threshold.
>
> As empirically demonstrated in Figure 3(b), the network achieves its peak performance at a firing threshold of 0.6. We believe this reflects a critical trade-off between representational density and information flow.
>
> When the firing threshold is too low, a large proportion of neurons fire at each timestep, leading to dense and noisy neural activity. This can cause the spike patterns for different input samples to become less distinct, impairing the network's ability to learn discriminative features.
>
> Conversely, if the firing threshold is set too high, the network's activity becomes excessively sparse. In this regime, information may be lost because too few spikes are generated to encode and propagate sufficiently rich signals through the network. This makes the learning task excessively difficult, as the distillation process relies on a meaningful flow of features between the two pathways, and many neurons may become "silent," ceasing to contribute to learning.
>
> Therefore, the optimal threshold of 0.6 appears to strike a crucial balance, maintaining enough sparsity to form distinct representations while still permitting a robust flow of information necessary for effective learning.
>
> [1] Lillicrap, T. P., Santoro, A., Marris, L., Akerman, C. J., & Hinton, G. E. (2020). Backpropagation and the brain. Nature Reviews Neuroscience, 21(6), 335–346.
>
> [2] Bengio, Y., Lee, D. H., Bornschein, J., Mesnard, T., & Lin, Z. (2015). Towards biologically plausible deep learning. arXiv preprint arXiv:1502.04156.

---

### Author Response · Authors · 2025-11-20
**Global Response**

We express our gratitude to all the reviewers for their valuable insights! We are happy to hear that you acknowledged our contributions. We appreciate all reviewers for their comments, highlighting the strengths of our work for a summary:
- **Solid biological plausibility**: BSD rigorously satisfies all five proposed biological criteria, resolving conflicts found in existing methods and aligning strictly with neurobiological principles (Reviewers 5Dww, xq86, SDT5, E6E3).

- **Novel and elegant mechanism**: The bidirectional distillation mechanism offers a unique and elegant approach that closely mirrors the brain's integration of bottom-up perception and top-down recall (Reviewers 5Dww, SDT5, E6E3).

- **Competitive performance**: Our algorithm achieves accuracy comparable to backpropagation on complex benchmarks, proving its practicality without compromising biological fidelity (Reviewers 5Dww, xq86, E6E3).

- **Broad versatility**: Extensive experiments across diverse network architectures (MLPs, CNNs, RNNs) and tasks (classification, generation, regression) verify the algorithm's strong generalization ability (Reviewers 5Dww, xq86, SDT5, MvQY, E6E3).

A major concern is the computational overhead, specifically regarding GPU memory consumption. In response, we have conducted profiling experiments and included the full results in the revised appendix. We clarify that while BSD incurs higher memory costs during training due to the necessity of maintaining parameters and activations for both the feedforward and backward pathways, its inference resource consumption is identical to that of a standard BP-trained SNN. This is because inference for classification exclusively utilizes the feedforward pathway. The detailed comparison across different batch sizes is shown below:

**Inference Phase Memory Consumption:**
| Batch Size | Method | Memory Consumption (MB) |
|:----------:|:------:|:-----------------------:|
| 32         | BP     | 344                     |
| 32         | BSD    | 344                     |
| 64         | BP     | 944                     |
| 64         | BSD    | 944                     |
| 128        | BP     | 1826                    |
| 128        | BSD    | 1826                    |

**Training Phase Memory Consumption:**
| Batch Size | Method | Memory Consumption (MB) |
|:----------:|:------:|:-----------------------:|
| 32         | BP     | 950                     |
| 32         | BSD    | 1154                    |
| 64         | BP     | 1844                    |
| 64         | BSD    | 2852                    |
| 128        | BP     | 3324                    |
| 128        | BSD    | 5034                    |

Another concern involves the sensitivity of the BSD algorithm to batch size, given the reliance on the contrastive ReCo loss. To address this, we performed experiments across multiple datasets, the results of which have been added to the revised manuscript. The experiments demonstrate a consistent trend where model performance improves as the batch size increases—aligning with the intuition that a larger set of negative samples refines the representation space—and eventually stabilizes, confirming that BSD is robust once a sufficient batch size (e.g., 128) is reached. The result is shown below:

| Batch Size | SVHN    | CIFAR-10 | CIFAR-100 |
|:----------:|:-------:|:--------:|:---------:|
| 16         | 71.55%  | 68.96%   | 34.99%    |
| 32         | 78.18%  | 78.07%   | 42.53%    |
| 64         | 90.97%  | 81.54%   | 49.15%    |
| 128        | 90.81%  | 84.53%   | 53.11%    |
| 256        | 88.97%  | 83.85%   | 53.49%    |

We have addressed the concerns and polished our paper in the revised version (blue font) to make our work better understandable to the wider community. Specifically:
1. We have optimized the figures in the manuscript, specifically increasing font sizes and refining descriptions to ensure they are clear and legible.
2. We have corrected typos and further clarified the mathematical notations and equations throughout the paper to enhance precision and readability.
3. We have expanded the Related Work section to include a brief discussion of additional biologically plausible learning algorithms suggested by the reviewers.
4. We have enriched the Appendix with additional experimental results and detailed analyses, including the studies on batch size, robustness, large-scale datasets, and computational resource consumption.

Finally, we sincerely appreciate the reviewers' recognition of our work's potential to inspire further research in biologically plausible learning and neuromorphic computing. Our study contributes not only by presenting a rigorous, biologically grounded framework but also by showing that such fidelity can yield performance competitive with BP across diverse tasks. We believe that this work is worth publishing to stimulate further discussion and pave the way for next-generation biologically inspired intelligence.

---

### Meta-Review · Area_Chair_6viz · 2026-01-07

**Summary:**

The paper presents a learning framework, BSD, with solid biological plausibility. Its core mechanism, bidirectional distillation between bottom-up perception and top-down memory recall, is novel and elegant. Across multiple tasks, BSD demonstrates competitive performance comparable to backpropagation while satisfying key biological constraints such as local synaptic plasticity and unsigned error signals. Although reviewers raised valid concerns regarding the locality of the ReCo loss, experimental scale, and reproducibility, the authors have effectively addressed them in their response with additional analyses, including training/inference memory usage and batch size robustness experiments.

**Reviewer Concerns:**

The reviewers' common concerns lie in the biological plausibility of the ReCo loss, limited experimental scale, ambiguity in methodological details, and reproducibility.

I believe the authors have now provided an effective and comprehensive improvement to the paper. They have addressed the major concerns by adding quantitative analyses of training and inference memory usage, demonstrating robustness across varying batch sizes on multiple datasets, clarifying notation and figure presentation, and refining their discussion of biological assumptions.

**Reviewer Scores:**

I think E6E3 SDT5 would likely raise their scores after seeing the authors’ responses.

---

### Decision · Program_Chairs · 2026-01-26

Accept (Poster)